# COMPUTATIONAL LIMITS OF LOW-RANK ADAPTATION (LoRA) FINE-TUNING FOR TRANSFORMER MODELS*

**Jerry Yao-Chieh Hu**[†]    **Maojiang Su**[‡]    **En-Jui Kuo**[♭]    **Zhao Song**[§]    **Han Liu**[†]

[†]Northwestern University    [‡]University of Science and Technology of China
[♭]National Yang Ming Chiao Tung University    [§]Simons Institute, UC Berkeley

jhu@u.northwestern.edu, sumaojiang@mail.ustc.edu.cn,
kuoenjui@nycu.edu.tw, magic.linuxkde@gmail.com, hanliu@northwestern.edu

## ABSTRACT

We study the computational limits of Low-Rank Adaptation (LoRA) for fine-tuning transformer-based models using fine-grained complexity theory. Our key observation is that the existence of low-rank decompositions within the gradient computation of LoRA adaptation leads to possible algorithmic speedup. This allows us to (i) identify a phase transition behavior of efficiency assuming the Strong Exponential Time Hypothesis (SETH), and (ii) prove the existence of almost linear algorithms by controlling the LoRA update computation term by term. For the former, we identify a sharp transition in the efficiency of all possible rank-$r$ LoRA update algorithms for transformers, based on specific norms resulting from the multiplications of the input sequence $X$, pretrained weights $W^\star$, and adapter matrices $\alpha BA/r$. Specifically, we derive a shared upper bound threshold for such norms, and show that efficient (sub-quadratic) approximation algorithms of LoRA exist only below this threshold. For the latter, we prove the existence of almost linear approximation algorithms for LoRA adaptation by utilizing the hierarchical low-rank structures of LoRA gradients and approximating the gradients with a series of chained low-rank approximations. To showcase our theory, we consider two practical scenarios: partial (e.g., only $W_V$ and $W_Q$) and full adaptations (e.g., $W_Q$, $W_V$, and $W_K$) of weights in attention heads.

## 1 INTRODUCTION

We investigate the computational limits of finetuning large transformer-based pretrained model with **Lo**w-**R**ank **A**daptation (**LoRA**). This analysis is of practical importance in the era of Large Foundation Models (Bommasani et al., 2021). Large foundation models are gigantic transformer-based architectures, pretrained on vast datasets, are pivotal across multiple fields, including natural language processing (Achiam et al., 2023; Touvron et al., 2023b;a; Brown et al., 2020; Floridi and Chiriatti, 2020), finance (Yang et al., 2023; Wu et al., 2023), genomics (Nguyen et al., 2024; Zhou et al., 2025; 2024; 2023; Ji et al., 2021), medical science (Thirunavukarasu et al., 2023; Singhal et al., 2023; Moor et al., 2023) and more. They are powerful but very expensive to pretrain. Therefore, most practitioners rely on finetuing methods to adapt these models for their specific needs (Zheng et al., 2024; Ding et al., 2022). LoRA (Mao et al., 2025; Hu et al., 2021) is the most prevalent fine-tuning method due to its parameter efficiency due to the low-rank adaptation of model weights. However, even with LoRA, updating the partial weights of pretrained transformer-based models using gradient methods remains costly. Notably, the naive backward pass in transformer architectures retains the same quadratic-in-sequence-length computational time complexity as its forward pass (see Appendix F for discussions and a proof). This work provides a timely theoretical analysis of LoRA's computational limits, aiming to advance efficient finetuning of large foundation models.

The hardness of LoRA finetuning transformer-based foundation model ties to both forward and backward passes. To analyze, it suffices to focus on just transformer attention heads due to their dominating quadratic time complexity in both passes. We first make the following observation:

---

*Code is available on OpenReview; full version and future updates are on arXiv.

The hardness of LoRA's forward pass is trivially characterized by (Alman and Song, 2023).

To see this, let $X \in \mathbb{R}^{L \times d}$ be input with length $L$, and $W_K, W_Q, W_V \in \mathbb{R}^{d \times d}$ be attention weights, and $Q = XW_V \in \mathbb{R}^{L \times d}, K = XW_K \in \mathbb{R}^{L \times d}, V = XV \in \mathbb{R}^{L \times d}$. The Attention Mechanism is

$$Z = \mathrm{Softmax}\left(QK^\mathsf{T}\beta\right)V = D^{-1}\exp\left(XW_QW_K^\mathsf{T}X^\mathsf{T}\beta\right)XW_V, \qquad (1.1)$$

with the inverse temperature $\beta > 0$ and $D := \mathrm{diag}\left(\exp\left(XW_QW_K^\mathsf{T}X^\mathsf{T}\beta\right)\mathbb{1}_L\right)$. Here, $\exp(\cdot)$ is entry-wise exponential function, $\mathrm{diag}(\cdot)$ converts a vector into a diagonal matrix with the entries of the vector, and $\mathbb{1}_L$ is the length-$L$ all ones vector. LoRA finetuning is given as

**Definition 1.1** (LoRA (Hu et al., 2021)). Let $W \in \mathbb{R}^{b \times a}$ be any weight matrix in a pretrained model $F$, LoRA fine-tunes $F$ through updating $W$ with a low-rank decomposition $W = W^\star + \frac{\alpha}{r}BA$. Here, $W^\star$ is the frozen pretrained weight. Only $B \in \mathbb{R}^{b \times r}$ and $A \in \mathbb{R}^{r \times a}$ are learnable (being update via gradient descent) with rank $r < \min(a, b)$ and tunable hyperparameter $\alpha \in \mathbb{R}$.

Under the Strong Exponential Time Hypothesis (Hypothesis 1), Alman and Song (2023) state:

**Lemma 1.1** (Informal, (Alman and Song, 2023)). Fast (sub-quadratic) forward pass of transformer only exist when entries of $K, Q, V$ are bounded by a constant $B = \Theta(\sqrt{\log L})$.

It is easy to see that Lemma 1.1 is transferable to LoRA inference according to Definition 1.1. However, we still need the hardness of backward pass to fully characterize LoRA for transformers. The analysis of the backpropagation (backward pass) is less straightforward. It involves managing the computation of numerous gradients for attention scores, with the number of chain-rule terms scaling quadratically in $L$ and the numbers of LoRA weights. While it is tempting to design algorithms to circumvent this $\Omega(L^2)$ computation time, to the best of our knowledge, there are no formal results to support and characterize such algorithms. To address this gap, we pose the following questions and provide a fundamental theory to fully characterize the complexity of LoRA for transformer models:

**Question 1.** Is it possible to improve the $\Omega(L^2)$ time with a bounded approximation error?

**Question 2.** More aggressively, is it possible to do such gradient computations in almost linear time?

To address these questions, we explore approximate LoRA gradient computations with precision guarantees. We first layout the objective of finetuning transformer-based pretrained models.

**Definition 1.2** (LoRA Loss for Adapting $W_K, W_Q, W_V$ of an Attention Head). Let $\mathcal{D} = \{X_i, Y_i\}_{i=1}^N$ be a dataset of size $N$ with $X_i \in \mathbb{R}^{L \times d}$ being the input and $Y_i \in \mathbb{R}^{L \times d}$ being the label. Fine-tuning a (self-)attention with LoRA with $\ell_2$ loss on dataset $\mathcal{D}$ is formulated as

$$\min_{\substack{B_K, B_Q, B_V \in \mathbb{R}^{d \times r}, \\ A_K, A_Q, A_V \in \mathbb{R}^{r \times d}}} \mathcal{L}\left(W_K = W_K^\star + \frac{\alpha}{r}B_KA_K, W_Q = W_Q^\star + \frac{\alpha}{r}B_QA_Q, W_V = W_V^\star + \frac{\alpha}{r}B_VA_V\right)$$

$$:= \frac{1}{2N}\sum_{i=1}^N \left\| D^{-1}\exp\left\{X_iW_QW_K^\mathsf{T}X_i^\mathsf{T}\beta\right\}X_iW_V - Y_i \right\|_F^2. \qquad (1.2)$$

Here $D := \mathrm{diag}\left(\exp\left\{XW_QW_K^\mathsf{T}X^\mathsf{T}\beta\right\}\mathbb{1}_n\right) \in \mathbb{R}^{L \times L}$.

We study the following approximation problem. Let $\underline{Z} := \mathrm{vec}(Z) \in \mathbb{R}^{ab}$ for any matrix $Z \in \mathbb{R}^{a \times b}$.

**Problem 1** (Approximate LoRA Gradient Computation ($\mathsf{ALoRAGC}(L, d, r, \epsilon)$)). Assume all numerical values in $\log(L)$ bits encoding. Let $\mathcal{L}$ follow Definition 1.2. The problem of approximating gradient computation of optimizing (1.2) is to find six surrogate gradient matrices $\{\widetilde{G}_\mu^{(A)} \in \mathbb{R}^{d \times r}, \widetilde{G}_\mu^{(B)} \in \mathbb{R}^{r \times d}\}_{\mu=K,Q,V}$ such that $\max\left(\left\{\left\|\widetilde{G}_\mu^{(B)} - \frac{\partial \mathcal{L}}{\partial B_\mu}\right\|_\infty, \left\|\widetilde{G}_\mu^{(A)} - \frac{\partial \mathcal{L}}{\partial A_\mu}\right\|_\infty\right\}_{\mu=K,Q,V}\right) \le \epsilon$, for some $\epsilon > 0$, where $\|Z\|_\infty := \max_{i,j}|Z_{ij}|$.

**Remark 1.1.** Any method or algorithm that aims to compute LoRA gradients beyond vanilla computation of (1.2) falls within the scope of this problem. Examples include using sampling strategies to avoid full LoRA gradient computation (Pan et al., 2024) or employing model quantization

for efficiency via low-precision gradient computation (Li et al., 2024; Dettmers et al., 2024). Common among these approaches is the need to compute surrogate LoRA gradients with reduced computational cost. We abstract this key subroutine and consider the fundamental algorithmic Problem 1.

In this work, we aim to investigate the computational limits of all possible efficient algorithms of ALoRAGC($L, d, r, \epsilon$) under realistic setting $\epsilon = 1/\text{poly}(L)$.

**Contributions.** Our contributions are 2-fold:

- **Norm-Based Phase Transition of Efficiency (Theorem 4.1).** We answer Question 1 by identifying a phase transition behavior on the norm of input, pretrained and adaptor weights, assuming the Strong Exponential Time Hypothesis (SETH). Specifically, we identify an inefficiency threshold for these norms such that, only below which, adapting transformer-based models with LoRA in $L^{2-o(1)}$ (sub-quadratic) time is possible.

> **Theorem 1.1** (Informal Version of Theorem 4.1). Without appropriately normalized inputs $X$, pretrained attention weights $W_K^\star, W_Q^\star, W_V^\star$, and LoRA matrices $\{\alpha A_\mu B_\mu / r\}_{\mu = K, Q, V}$, there is no algorithm running in subquadratic time $O(L^{2-\delta})$ for any constant $\delta > 0$ to solve ALoRAGC.

- **Existence of Almost Linear Time LoRA Algorithms.** We answer Question 2 by proving that precision-guaranteed approximation to Problem 1 is achievable in *almost linear time* via hierarchical low-rank decomposition of LoRA gradients. To showcase our theory, we analyze two practical scenarios highlighted in (Hu et al., 2021): *partial* adaptations (e.g., only $W_V$ and $W_Q$ in Section 3), and *full* adaptations (e.g., $W_K, W_Q, W_V$ in Appendix A) of weights in attention heads.

> **Theorem 1.2** (Informal Version of Theorems 3.1 and A.1). Given appropriately normalized inputs $X$, pretrained attention weights $W_K^\star, W_Q^\star, W_V^\star$, and LoRA matrices $\{\alpha A_\mu B_\mu / r\}_{\mu = K, Q, V}$, there exists an algorithm that solves ALoRAGC in almost linear time $O(L^{1+o(1)})$.

On the theoretical front, we characterize the computational feasibility of LoRA by showing the existence of precision-guaranteed, efficient (subquadratic or almost linear time) LoRA methods and identifying their necessary conditions. On the practical front, these conditions serve as valuable guidelines for implementations (please see Remark 5.2 for discussions and Appendix G for numerical justifications). Importantly, our theory only requires one assumption on numerical value encoding (e.g., in $\log L$ bits with $L$ being the sequence length). Such an assumption is minimal and realistic. No assumptions are made about the data or model, making our results widely applicable.

**Organization.** Section 2 includes preliminaries and problem setup. Section 3 presents analysis of LoRA adaptation on only $W_Q, W_K$. Appendix A presents analysis of LoRA adaptation on all $W_Q, W_K, W_V$. Section 4 characterizes the computational limits of all possible efficient algorithms for LoRA. Section 5 includes concluding remarks. We defer discussions of related works to Appendix B.

**Notations.** We denote (column) vectors by lower case letters, and matrices by upper case letters. Let $\mathbb{1}_L$ denote the length-$L$ all ones vector. We write $\langle a, b \rangle := a^\top b$ as the inner product for vectors $a, b$. Let $a[i]$ denotes the $i$-th component of vector $a$. Let $A[i, j]$ and $A_{ij}$ denotes the $(i, j)$-th entry of matrix $A$. For any matrix $A$, let $A[i, \cdot]$ and $A[\cdot, j]$ be the $i$-th row and $j$-th column of $A$, respectively. For $u, v \in \mathbb{R}^d$, we denote their Hadamard product as $u \odot v := (u_1 v_1, \ldots, u_d v_d)^\top$. The index set $\{1, \cdots, I\}$ is denoted by $[I]$, where $I \in \mathbb{N}_+$. For any $z \in \mathbb{R}^d$, we denote $\exp(z) \in \mathbb{R}^d$ whose $i$-th entry is $\exp(z_i)$. Let $\|A\|_\infty := \max_{i,j} |A_{ij}|$ for any matrix $A$. Let $\|\cdot\|_F$ denote the squared Frobenius norm, i.e., $\|A\|_F := (\sum_{i,j} A_{ij}^2)^{1/2}$.

## 2 PRELIMINARIES AND PROBLEM SETUP

This section presents the ideas we build on.

**Tensor Trick for Computing Gradients.** The tensor trick (Diao et al., 2019; 2018) is an instrument to compute complicated gradients in a clean and tractable fashion. As we shall see below, the purpose of the tensor trick is to convert matrix multiplication into vector form, making the gradient w.r.t. the matrix more tractable. For this, we introduce vectorization and its inverse operation, matrixization.

**Definition 2.1** (Vectorization). For any matrix $X \in \mathbb{R}^{L \times d}$, we define $\underline{X} := \text{vec}(X) \in \mathbb{R}^{Ld}$ such that $X_{i,j} = \underline{X}_{(i-1)d+j}$ for all $i \in [L]$ and $j \in [d]$.

**Definition 2.2** (Matrixization). For any vector $\underline{X} \in \mathbb{R}^{Ld}$, we define $\text{mat}(\underline{X}) = X$ such that $X_{i,j} = \text{mat}(\underline{X}) := \underline{X}_{(i-1)d+j}$ for all $i \in [L]$ and $j \in [d]$, namely $\text{mat}(\cdot) = \text{vec}^{-1}(\cdot)$.

Next, we introduce necessary tensor terminologies.

**Definition 2.3** (Kronecker Product). Let $A \in \mathbb{R}^{L_a \times d_a}$ and $B \in \mathbb{R}^{L_b \times d_b}$. We define the Kronecker product of $A$ and $B$ as $A \otimes B \in \mathbb{R}^{L_a L_b \times d_a d_b}$ such that $(A \otimes B)_{(i_a-1)L_b+i_b,(j_a-1)d_b+j_b}$, is equal to $A_{i_a,j_a} B_{i_b,j_b}$ with $i_a \in [L_a], j_a \in [d_a], i_b \in [L_b], j_b \in [d_b]$.

**Definition 2.4** (Sub-Block of a Tensor). For any $A \in \mathbb{R}^{L_a \times d_a}$ and $B \in \mathbb{R}^{L_b \times d_b}$, let $\mathsf{A} := A \otimes B \in \mathbb{R}^{L_a L_b \times d_a d_b}$. For any $\underline{j} \in [L_a]$, we define $\mathsf{A}_{\underline{j}} \in \mathbb{R}^{L_b \times d_a d_b}$ be the $\underline{j}$-th $L_b \times d_a d_b$ sub-block of $\mathsf{A}$.

Definition 2.3 creates a large matrix from two smaller matrices, preserving the structure and properties of the original matrices. Definition 2.4 provides a refined identification of specific entry-wise multiplications between the two *Kronecker-produced* matrices. Together, they makes the gradient w.r.t. the matrix more tractable: for instance, the gradient of below vectorized LoRA loss (2.1).

**Lemma 2.1** (Tensor Trick (Diao et al., 2019; 2018)). For any $A \in \mathbb{R}^{L_a \times d_a}$, $B \in \mathbb{R}^{L_b \times d_b}$ and $X \in \mathbb{R}^{d_a \times d_b}$, it holds $\text{vec}(AXB^\mathsf{T}) = (A \otimes B)\underline{X} \in \mathbb{R}^{L_a L_b}$.

To showcase the tensor trick for LoRA, let's consider a (single data point) simplified (1.2)

$$\mathcal{L}_0 := \| \underbrace{D^{-1}}_{\in \mathbb{R}^{L \times L}} \underbrace{\exp\{XWX^\mathsf{T}\beta\}}_{\in \mathbb{R}^{L \times L}} \underbrace{X}_{\in \mathbb{R}^{L \times d}} \underbrace{W_V}_{d \times d} - \underbrace{Y}_{\in \mathbb{R}^{L \times d}} \|_F^2, \quad \text{with } W := W_Q W_K^\mathsf{T} \in \mathbb{R}^{d \times d}.$$

By Definition 2.3 and Definition 2.4, we identify $D_{\underline{j},\underline{j}} := \left\langle \exp\left(\mathsf{A}_{\underline{j}} \underline{W}\right), \mathbb{1}_L \right\rangle \in \mathbb{R}$ for all $\underline{j} \in [L]$, with $\mathsf{A} := X \otimes X \in \mathbb{R}^{L^2 \times d^2}$ and $\underline{W} \in \mathbb{R}^{d^2}$. Therefore, for each $\underline{j} \in [L]$ and $\underline{i} \in [d]$, it holds

$$\mathcal{L}_0 = \sum_{\underline{j}=1}^{L} \sum_{\underline{i}=1}^{d} \frac{1}{2} \left( \left\langle D_{\underline{j},\underline{j}}^{-1} \exp\left(\mathsf{A}_{\underline{j}} \underline{W}\right), XW_V[\cdot, \underline{i}] \right\rangle - Y_{\underline{j},\underline{i}} \right)^2. \tag{2.1}$$

Gao et al. (2023a;b) show that (2.1) provides term-by-term tractability for gradient computation of $\mathcal{L}_0$. Specifically, it allow us to convert the attention score $D^{-1} \exp(XWX^\mathsf{T})$ into its vectorized form $(D \otimes I_L)^{-1} \exp(\mathsf{A} \underline{W}) \in \mathbb{R}^{L^2}$ and split the vectorized form into $L$ terms of size $L$. This provides a systematic way to manage the chain-rule terms in the gradient computation of losses like $\mathcal{L}_0$, and opens the door to more general analytical feasibility for deep transformer-based models.

**Problem Setup: Which Attention Weights in Transformer Should We Apply LoRA to?** Following (Hu et al., 2021), we consider only adapting the attention weights for downstream tasks. This consideration is sufficient to justify our techniques as the attention head dominates the time complexity of transformer-based foundation models. Namely, we consider updating (as in Definition 1.2)

$$W_Q = W_Q^\star + \frac{\alpha}{r} B_Q A_Q, \quad W_K = W_K^\star + \frac{\alpha}{r} B_K A_K, \quad W_V = W_V^\star + \frac{\alpha}{r} B_V A_V.$$

Furthermore, for completeness, we consider two de facto scenarios as in (Hu et al., 2021, Sec. 7.1):

(C1) **Special Case.** Adapting only $W_Q$ and $W_V$ for best performance under fixed parameter budge.

(C2) **General Case.** Adapting $W_K, W_Q, W_V$ for best performance.

We analyze (C1) **Special Case** in Section 3 and (C2) **General Case** in Appendix A.

To consider the problem of adapting attention head, we first generalize Definition 1.2 to the following generic attention with triplet input sequences. For reasons, this allows our results to be applicable. Moreover, this helps us to focus on parts dominating the efficiency of gradient computation.

**Definition 2.5** (Learning Generic Attention). Let $\mathcal{D} = \{(X_i^{(K)}, X_i^{(Q)}, X_i^{(V)}), Y_i\}_{i=1}^N$ be a dataset of size $N$ with the triplet $X_i^{(K)}, X_i^{(Q)}, X_i^{(V)} \in \mathbb{R}^{L \times d}$ being the input and $Y_i \in \mathbb{R}^{L \times d}$ being the label. The problem of learning a generic attention with $\ell_2$ loss from dataset $\mathcal{D}$ is formulated as

$$\min_{W_K, W_Q, W_V \in \mathbb{R}^{d \times d}} \frac{1}{N} \sum_{i=1}^N \mathcal{L}(W_K, W_Q, W_V)$$

$$:= \min_{W_K, W_Q, W_V \in \mathbb{R}^{d \times d}} \frac{1}{2N} \sum_{i=1}^N \left\| D^{-1} \exp\left\{ X_i^{(Q)} W_Q W_K^\mathsf{T} \left( X_i^{(K)} \right)^\mathsf{T} \beta \right\} X_i^{(V)} W_V - Y_i \right\|_F^2.$$

Here $D := \mathrm{diag}\left( \exp\left\{ X_i^{(Q)} W_Q W_K^\mathsf{T} \left( X_i^{(K)} \right)^\mathsf{T} \beta \right\} \mathbb{1}_n \right) \in \mathbb{R}^{L \times L}$.

**Remark 2.1.** Definition 2.5 is generic. If $X_i^{(K)} = X_i^{(V)} \neq X_i^{(Q)} \in \mathbb{R}^{L \times d}$, Definition 2.5 reduces to cross-attention. If $X_i^{(K)} = X_i^{(Q)} = X_i^{(V)} \in \mathbb{R}^{L \times d}$, Definition 2.5 reduces to self-attention.

# 3 SPECIAL CASE: LoRA ADAPTATION ON ONLY $W_Q$ AND $W_V$

Formally, we formulate the *partial* adaptation (C1) of an attention head as the following LoRA loss.

**Definition 3.1** (Adapting $W_Q$, $W_V$ of Generic Attention with LoRA). Let $\mathcal{D} = \{\left( X_i^{(K)}, X_i^{(Q)}, X_i^{(V)} \right), Y_i\}_{i=1}^N$ be a dataset of size $N$ with the triplet $X_i^{(K)}, X_i^{(Q)}, X_i^{(V)} \in \mathbb{R}^{L \times d}$ being the input and $Y_i \in \mathbb{R}^{L \times d}$ being the label. The problem of fine-tuning $W_Q$, $W_V$ a generic attention with LoRA with $\ell_2$ loss from dataset $\mathcal{D}$ is formulated as

$$\min_{\substack{B_Q, B_V \in \mathbb{R}^{d \times r} \\ A_Q, A_V \in \mathbb{R}^{r \times d}}} \mathcal{L}\left( W_K^\star, W_Q = W_Q^\star + \frac{\alpha}{r} B_Q A_Q, W_V = W_V^\star + \frac{\alpha}{r} B_V A_V \right) \tag{3.1}$$

$$:= \min_{\substack{B_Q, B_V \in \mathbb{R}^{d \times r} \\ A_Q, A_V \in \mathbb{R}^{r \times d}}} \frac{1}{2N} \sum_{i=1}^N \left\| \underbrace{D^{-1} \exp\left\{ X_i^{(Q)} W_Q (W_K^\star)^\mathsf{T} \left( X_i^{(K)} \right)^\mathsf{T} \beta \right\}}_{(I)} \underbrace{X_i^{(V)} W_V}_{(II)} - Y_i \right\|_F^2.$$

Here $D := \mathrm{diag}\left( \exp\left\{ X_i^{(Q)} W_Q (W_K^\star)^\mathsf{T} \left( X_i^{(K)} \right)^\mathsf{T} \beta \right\} \mathbb{1}_n \right) \in \mathbb{R}^{L \times L}$.

In this work, we are interested in the efficiency of optimizing (3.1) with gradient descent. For simplicity of our analysis, we employ the following four simplifications:

(S1) Since (II) ($V$ multiplication) is linear in weight while (I) ($K$-$Q$ multiplication) is exponential in weights, we only need to focus on the gradient of $K$-$Q$ multiplication. Therefore, for efficiency analysis of gradient, it is equivalent to analyze a reduced problem with fixed $W_V$.

(S2) To further simplify, we introduce $C_i^{(1)}, C_i^{(2)}, C_i^{(3)} \in \mathbb{R}^{L \times d}$ via

$$\underbrace{X_i^{(Q)} \frac{\alpha}{r}}_{:= C_i^{(1)} \in \mathbb{R}^{L \times d}} \left( \frac{r}{\alpha} W_Q^\star + B_Q A_Q \right) \underbrace{(W_K^\star)^\mathsf{T} \left( X_i^{(K)} \right)^\mathsf{T}}_{:= \left( C_i^{(2)} \right)^\mathsf{T} \in \mathbb{R}^{d \times L}} := C_i^{(1)} B_Q A_Q \left( C_i^{(2)} \right)^\mathsf{T}, \quad X_i^{(V)} W_V^\star := C_i^{(3)}.$$

$$\tag{3.2}$$

Notably, $C_i^{(1)}, C_i^{(2)}, C_i^{(3)}$ are constants with respect to adapting (3.1) with gradient updates.

(S3) **Trivial Reduction.** To prove the hardness of Problem 1 for both full gradient descent and stochastic mini-batch gradient descent, it suffices to consider adapting on a single data point.

(S4) We set $\beta = 1$ without loss of generality. Note that $\beta$ and $\alpha/r$ do not impact the running time of gradient computation since they are just rescaling factors.

Thus, we deduce Definition 3.1 to

$$
\min_{\substack{B_Q \in \mathbb{R}^{d \times r} \\ A_Q \in \mathbb{R}^{r \times d}}} \mathcal{L}(B_Q, A_Q) = \min_{\substack{B_Q \in \mathbb{R}^{d \times r} \\ A_Q \in \mathbb{R}^{r \times d}}} \frac{1}{2} \left\| D^{-1} \exp\left\{ C^{(1)} \left( \overline{W}_Q^\star + B_Q A_Q \right) \left( C^{(2)} \right)^\mathsf{T} \right\} C^{(3)} - Y \right\|_F^2,
$$
$$(3.3)$$

where $\overline{W}_Q^\star := r W_Q^\star / \alpha$ and $D = \mathrm{diag}\left( \exp\left\{ C^{(1)} \left( \overline{W}_Q^\star + B_Q A_Q \right) \left( C^{(2)} \right)^\mathsf{T} \right\} \mathbb{1}_L \right) \in \mathbb{R}^{L \times L}$.

We introduce the next problem to characterize all possible (efficient or not) gradient computation of optimizing (3.3). Let $Y[i, \cdot]$ and $Y[\cdot, j]$ be the $i$-th row and $j$-th column of $Y$, respectively.

> **Problem 2** (Approximate LoRA Gradient Computation $\mathsf{ALoRAGC}(L, d, r, \epsilon)$). Given $C_i^{(1)}, C_i^{(2)}, C_i^{(3)}, Y_i \in \mathbb{R}^{L \times d}$. Let $\epsilon > 0$. Assume all numerical values are in $\log(L)$-bits encoding. Let $\mathcal{L}$ follows (3.3). The problem of approximating gradient computation of optimizing (3.3) is to find two matrices $\widetilde{G}_Q^{(A)} \in \mathbb{R}^{d \times r}$ and $\widetilde{G}_Q^{(B)} \in \mathbb{R}^{r \times d}$ such that
>
> $$
> \max \left( \| \widetilde{G}_Q^{(B)} - \frac{\partial \mathcal{L}}{\partial \underline{B}_Q} \|_\infty, \| \widetilde{G}_Q^{(A)} - \frac{\partial \mathcal{L}}{\partial \underline{A}_Q} \|_\infty \right) \leq \epsilon.
> $$

The explicit gradient of LoRA loss (3.3) is too complicated to characterize Problem 2. To combat this, we employ the tensor trick. Let $W := \overline{W}_Q^\star + B_Q A_Q \in \mathbb{R}^{d \times d}$ such that $\mathrm{vec}(W) = \underline{W} \in \mathbb{R}^{d^2}$.

> **Definition 3.2** (Vectorized Attention Score). Let $\mathsf{C} := C^{(1)} \otimes C^{(2)}$ such that $\mathsf{C}_{\underline{j}} \in \mathbb{R}^{L \times d^2}$ for all $\underline{j} \in [L]$. For every $\underline{j} \in [L]$, we define $u(\underline{W})_{\underline{j}} : \mathbb{R}^{d^2} \to \mathbb{R}^L$ as: $u(\underline{W})_{\underline{j}} := \exp\left( \mathsf{C}_{\underline{j}} \underline{W} \right) \in \mathbb{R}^L$.

Definition 3.2 decomposes the complicated matrix $\exp\left( C^{(1)} (\overline{W}_Q^\star + B_Q A_Q)(C_i^{(2)})^\mathsf{T} \right)$ in loss (3.3) into $L$ vectors. Importantly, since the weight $W$ is vectorized into $\underline{W}$, such a vectorized representation allows more tractable gradient computation by its term-by-term identifiability.

> **Definition 3.3** (Attention Score Normalization). Let $\mathsf{C} := C^{(1)} \otimes C^{(2)}$ such that $\mathsf{C}_{\underline{j}} \in \mathbb{R}^{L \times d^2}$ for all $\underline{j} \in [L]$. For every $\underline{j} \in [L]$, we define $\alpha(x)_{\underline{j}} : \mathbb{R}^{d^2} \to \mathbb{R}$ as: $\alpha(\underline{W})_{\underline{j}} := \left\langle \exp\left( \mathsf{C}_{\underline{j}} \underline{W} \right), \mathbb{1}_L \right\rangle \in \mathbb{R}$.

Similarly, Definitions 3.2 and 3.3 provide analytical tractability of the matrix $D$ in loss (3.3).

> **Definition 3.4** (Vectorized, Normalized Attention Score). For a fixed $\underline{j} \in [L]$, we define $f(\underline{W})_{\underline{j}} : \mathbb{R}^{d^2} \to \mathbb{R}^L$ as: $f(\underline{W})_{\underline{j}} := \alpha(\underline{W})_{\underline{j}}^{-1} u(\underline{W})_{\underline{j}}$ such that $f(\underline{W}) \in \mathbb{R}^{L \times L}$ denotes the matrix whose $\underline{j}$-th row is $(f(\underline{W})_{\underline{j}})^\top$.

Definition 3.4 decomposes the complicated matrix multiplication $D^{-1} \exp\left( C^{(1)} (W_Q^\star + B_Q A_Q)(C^{(2)})^\mathsf{T} \right) C^{(3)}$ in loss (3.3) into $L$ terms. Note that the gradients w.r.t. $\underline{W}$ are still tractable due to simple chain rule (by design of $\alpha(\cdot)$ and $u(\cdot)$).

> **Definition 3.5** (Vectorized LoRA Loss (3.3)). For every $i \in [d]$, let $C^{(3)}[\cdot, i]$ follow (S2). For every $\underline{j} \in [L]$ and $i \in [d]$, we define $c(x)_{\underline{j}, i} : \mathbb{R}^{d^2} \times \mathbb{R}^{d^2} \to \mathbb{R}$ as: $c(\underline{W})_{\underline{j}, i} := \langle f(\underline{W})_{\underline{j}}, C^{(3)}[\cdot, i] \rangle - Y_{\underline{j}, i}$. Here $Y_{\underline{j}, i} = Y[\underline{j}, i]$ is the $(\underline{j}, i)$-th entry of $Y \in \mathbb{R}^{L \times d}$ for $\underline{j} \in [L], i \in [d]$.

From above definitions, we read out $c(\underline{W}) = f(\underline{W}) C^{(3)} - Y$ such that (3.3) becomes

$$
\mathcal{L}(\underline{W}) = \sum_{\underline{j}}^L \sum_{i=1}^d \mathcal{L}(\underline{W})_{\underline{j}, i} = \frac{1}{2} \sum_{\underline{j}}^L \sum_{i=1}^d c(\underline{W})_{\underline{j}, i}^2. \tag{3.4}
$$

(3.4) presents a decomposition of the LoRA loss (3.3) into $L \cdot d$ terms, each simple enough for tracking gradient computation. Now, we are ready to compute the gradient of the LoRA loss.

**Lemma 3.1** (Low-Rank Decomposition of LoRA Gradient). Let matrix $B_Q, A_Q$ and loss function $\mathcal{L}$ follow (3.3), $W := \overline{W}_Q^\star + B_Q A_Q$ and $\mathsf{C} := C^{(1)} \otimes C^{(2)}$. It holds

$$\frac{\mathrm{d}\mathcal{L}(\underline{W})}{\mathrm{d}\underline{W}} = \sum_{\underline{j}=1}^{L} \sum_{i=1}^{d} c(\underline{W})_{\underline{j},i} \mathsf{C}_{\underline{j}}^\top \underbrace{\left( \overbrace{\mathrm{diag}\left( f(\underline{W})_{\underline{j}} \right)}^{(II)} - \overbrace{f(\underline{W})_{\underline{j}} f(\underline{W})_{\underline{j}}^\top}^{(III)} \right)}_{(I)} C^{(3)}[\cdot, i]. \tag{3.5}$$

*Proof.* See Appendix C.1 for a detailed proof. □

**Remark 3.1** (Benefit from Tensor Trick: Fast Approximation). As we shall show in subsequent sections, Lemma 3.1 also enables the construction of fast approximation algorithms for (3.5) with precision guarantees due to its analytical feasibility. Surprisingly, it is even possible to compute (3.5) in almost linear time. To proceed, we further decompose (3.5) into its fundamental building blocks according to the chain-rule in the next lemma, and then conduct the approximation term-by-term.

**Remark 3.2** (LoRA Gradient Computation Takes Quadratic Time). Lemma 3.1 implies that LoRA's gradient computation takes quadratic time, similar to inference hardness result (Alman and Song, 2023). This is non-trivial yet not the main focus of this work. Please see Appendix F for details.

**Lemma 3.2** (Vectorized $\frac{\partial \mathcal{L}}{\partial A_Q}, \frac{\partial \mathcal{L}}{\partial B_Q}$). Let $q(\underline{W}) := C^{(3)}(c(\underline{W}))^\top \in \mathbb{R}^{L \times L}$. For every index $\underline{j} \in [L]$ , we define $p(\underline{W})_{\underline{j}} \in \mathbb{R}^L$ as $p(\underline{W})_{\underline{j}} := \left( \mathrm{diag}\left( f(\underline{W})_{\underline{j}} \right) - f(\underline{W})_{\underline{j}} f(\underline{W})_{\underline{j}}^\top \right) q(\underline{W})$. Then it holds

$$\frac{\partial \mathcal{L}}{\partial \underline{A}_Q} = \mathrm{vec}\left( B_Q^\top \left( C^{(1)} \right)^\top p(\underline{W}) C^{(2)} \right), \quad \frac{\partial \mathcal{L}}{\partial \underline{B}_Q} = \mathrm{vec}\left( \left( C^{(1)} \right)^\top p(\underline{W}) A_Q C^{(2)} \right). \tag{3.6}$$

*Proof.* See Appendix C.2 for a detailed proof. □

Lemma 3.2 states that the chain rule terms for characterizing Problem 2 are tied to $p(\cdot)$. Therefore, to characterize $\widetilde{G}_Q^{(A)}, \widetilde{G}_Q^{(B)}$ (i.e., the approximations of $G_Q^{(A)}, G_Q^{(B)}$), we need to approximate the functions $f(\cdot), q(\cdot), c(\cdot)$, and hence $p(\cdot)$ with precision guarantees. To do so, it is convenient to consider the following decomposition of $p(\cdot)$.

**Definition 3.6** (Decomposition of $p(\cdot)$). For every $\underline{j} \in [L]$, we define $p_1(\underline{W})_{\underline{j}}, p_2(\underline{W})_{\underline{j}} \in \mathbb{R}^L$ as

$$p_1(\underline{W})_{\underline{j}} := \mathrm{diag}\left( f(\underline{W})_{\underline{j}} \right) q(\underline{W})_{\underline{j}} \quad \text{and} \quad p_2(\underline{W})_{\underline{j}} := f(\underline{W})_{\underline{j}} f(\underline{W})_{\underline{j}}^\top q(\underline{W})_{\underline{j}},$$

such that $p(\underline{W}) = p_1(\underline{W}) - p_2(\underline{W})$.

**Overview of Our Proof Strategy.** Definition 3.6 motivates the following strategy: term-by-term approximation for precision-guaranteed, almost linear time algorithms to compute (3.6) (Problem 2).

**Step 1.** Prove the existence of almost linear approximation algorithms for $f(\cdot), q(\cdot), c(\cdot)$ via low-rank approximation: Lemma 3.3, Lemma 3.5 and Lemma 3.4.

**Step 2.** Prove the existence of almost linear approximation algorithms for $p_1(\cdot), p_2(\cdot)$ and hence $p(\cdot)$ via the low-rank-preserving property of the multiplication between $f(\cdot)$ and $q(\cdot)$: Lemma 3.6 and Lemma 3.7.

**Step 3.** Prove existence of almost linear approximation algorithms for the LoRA adapter gradients (i.e., $\frac{\partial \mathcal{L}}{\partial \underline{A}_Q}$ and $\frac{\partial \mathcal{L}}{\partial \underline{B}_Q}$ in (3.6)) with results from **Step 1 & 2**: Theorem 3.1.

**Step 1.** We start with low-rank approximations for $f(\cdot), q(\cdot), c(\cdot)$.

**Lemma 3.3** (Approximate $f(\cdot)$, Modified from (Alman and Song, 2023)). Let $\Gamma = o(\sqrt{\log L})$ and $k_1 = L^{o(1)}$. Let $C^{(1)}, C^{(2)} \in \mathbb{R}^{L \times d}$, $W \in \mathbb{R}^{d \times d}$, and $f(\underline{W}) = D^{-1} \exp\left( C^{(1)} W \left( C^{(2)} \right)^\top \right)$ with $D = \mathrm{diag}\left( \exp\left( C^{(1)} W \left( C^{(2)} \right)^\top \right) \mathbb{1}_L \right)$ follows Definitions 3.2 to 3.5. If $\max\left( \left\| C^{(1)} W \right\|_\infty \le \right.$

$\Gamma, \|C^{(2)}\|_\infty) \leq \Gamma$, then there exist two matrices $U_1, V_1 \in \mathbb{R}^{L \times k_1}$ such that $\|U_1 V_1^\top - f(\underline{W})\|_\infty \leq \epsilon/\mathrm{poly}(L)$. In addition, it takes $L^{1+o(1)}$ time to construct $U_1$ and $V_1$.

*Proof.* This lemma is an application of (Alman and Song, 2023, Theorem 3.8). □

**Lemma 3.4** (Approximate $c(\cdot)$)**.** Assume all numerical values are in $O(\log L)$ bits. Let $d = O(\log L)$ and $c(\underline{W}) \in \mathbb{R}^{L \times d}$ follows Definition 3.5. There exist two matrices $U_1, V_1 \in \mathbb{R}^{L \times k_1}$ such that

$$\left\| U_1 V_1^\top C^{(3)} - Y - c(\underline{W}) \right\|_\infty \leq \epsilon/\mathrm{poly}(L).$$

*Proof.* See Appendix C.3 for a detailed proof. □

**Lemma 3.5** (Approximate $q(\cdot)$)**.** Let $k_2 = L^{o(1)}$, $c(W) \in \mathbb{R}^{L \times d}$ follows Definition 3.5 and let $q(\underline{W}) := C^{(3)} (c(\underline{W}))^\top \in \mathbb{R}^{L \times L}$ follows Lemma 3.2. There exist two matrices $U_2, V_2 \in \mathbb{R}^{L \times k_2}$ such that $\|U_2 V_2^\top - q(\underline{W})\|_\infty \leq \epsilon/\mathrm{poly}(L)$. In addition, it takes $L^{1+o(1)}$ time to construct $U_2, V_2$.

*Proof.* See Appendix C.4 for a detailed proof. □

**Step 2.** Now, we use above lemmas to construct low-rank approximations for $p_1(\cdot), p_2(\cdot), p(\cdot)$.

**Lemma 3.6** (Approximate $p_1(\cdot)$)**.** Let $k_1, k_2, k_3 = L^{o(1)}$. Suppose $U_1, V_1 \in \mathbb{R}^{L \times k_1}$ approximates $f(\underline{W}) \in \mathbb{R}^{L \times L}$ such that $\|U_1 V_1^\top - f(\underline{W})\|_\infty \leq \epsilon/\mathrm{poly}(L)$, and $U_2, V_2 \in \mathbb{R}^{L \times k_2}$ approximates the $q(\underline{W}) \in \mathbb{R}^{L \times L}$ such that $\|U_2 V_2^\top - q(\underline{W})\|_\infty \leq \epsilon/\mathrm{poly}(L)$. Then there exist two matrices $U_3, V_3 \in \mathbb{R}^{L \times k_3}$ such that

$$\left\| U_3 V_3^\top - p_1(\underline{W}) \right\|_\infty \leq \epsilon/\mathrm{poly}(L).$$

In addition, it takes $L^{1+o(1)}$ time to construct $U_3, V_3$.

*Proof Sketch.* By tensor formulation, we construct $U_3, V_3$ as tensor products of $U_1, V_1$ and $U_2, V_2$, respectively, while preserving their low-rank structure. Then, we show the low-rank approximation of $p_1(\cdot)$ with bounded error by Lemma 3.3 and Lemma 3.5. See Appendix C.5 for a detailed proof. □

**Lemma 3.7** (Approximate $p_2(\cdot)$)**.** Let $k_1, k_2, k_4 = L^{o(1)}$. Let $p_2(W) \in \mathbb{R}^{L \times L}$ follow Definition 3.6 such that its $\underline{j}$-th column is $p_2(\underline{W})_{\underline{j}} = f(\underline{W})_{\underline{j}} f(\underline{W})_{\underline{j}}^\top q(\underline{W})_{\underline{j}}$ for each $\underline{j} \in [L]$. Suppose $U_1, V_1 \in \mathbb{R}^{L \times k_1}$ approximates the f(X) such that $\|U_1 V_1^\top - f(\underline{W})\|_\infty \leq \epsilon/\mathrm{poly}(L)$, and $U_2, V_2 \in \mathbb{R}^{L \times k_2}$ approximates the $q(\underline{W}) \in \mathbb{R}^{L \times L}$ such that $\|U_2 V_2^\top - q(\underline{W})\|_\infty \leq \epsilon/\mathrm{poly}(L)$. Then there exist matrices $U_4, V_4 \in \mathbb{R}^{L \times k_4}$ such that

$$\left\| U_4 V_4^\top - p_2(\underline{W}) \right\|_\infty \leq \epsilon/\mathrm{poly}(L)$$

In addition, it takes $L^{1+o(1)}$ time to construct $U_4, V_4$.

*Proof Sketch.* By considering the following decomposition through tensor formulation

$$p_2(\underline{W})_{\underline{j}} := \overbrace{f(\underline{W})_{\underline{j}} \underbrace{f(\underline{W})_{\underline{j}}^\top q(\underline{W})_{\underline{j}}}_{(I)}}^{(II)},$$

we approximate the $p_2(\cdot)$ part by part. Specifically, for (I), we show its low-rank approximation by observing the low-rank-preserving property of the multiplication between $f(\cdot)$ and $q(\cdot)$ (from Lemma 3.3 and Lemma 3.5). For (II), we show its low-rank approximation by the low-rank structure of $f(\cdot)$ and (I). See Appendix C.6 for a detailed proof. □

**Step 3.** Combining above, we arrive our main result: almost linear algorithm for Problem 2.

---

**Theorem 3.1** (Main Result: Existence of Almost Linear Time ALoRAGC). Suppose all numerical values are in $O(\log L)$-bits encoding. Recall that $W = \overline{W}_Q^\star + B_Q A_Q \in \mathbb{R}^{d \times d}$ with $\overline{W}_Q^\star := r W_Q^\star / \alpha$. Let $C^{(1)} = X^{(Q)} \frac{\alpha}{r}, C^{(2)} = X^{(K)} W_K$ follows (3.2). If $\|C^{(1)} W\|_\infty \le \Gamma$ and $\|C^{(2)}\|_\infty \le \Gamma$, where $\Gamma = o(\sqrt{\log L})$, then there exists a $L^{1+o(1)}$ time algorithm to solve ALoRAGC $\left(L, d = O(\log L), r = L^{o(1)}, \epsilon = 1/\text{poly}(L)\right)$ (i.e., Problem 2). In particular, this algorithm outputs gradient matrices $\widetilde{G}_Q^{(A)} \in \mathbb{R}^{d \times r}, \widetilde{G}_Q^{(B)} \in \mathbb{R}^{r \times d}$ such that

$$\|\frac{\partial \mathcal{L}}{\partial \underline{A}_Q} - \widetilde{G}_Q^{(A)}\|_\infty \le 1/\text{poly}(L), \quad \text{and} \quad \|\frac{\partial \mathcal{L}}{\partial \underline{B}_Q} - \widetilde{G}_Q^{(B)}\|_\infty \le 1/\text{poly}(L).$$

---

*Proof Sketch.* By Lemma 3.2, we have $\partial \mathcal{L}/\partial \underline{A}_Q = \text{vec}(B_Q^\top (C^{(1)})^\top p(\underline{W}) C^{(2)})$, and $\partial \mathcal{L}/\partial \underline{B}_Q = \text{vec}((C^{(1)})^\top p(\underline{W}) A_Q C^{(2)})$. By Lemma 3.2 and Definition 3.6, we have $p(\underline{W}) = p_1(\underline{W}) - p_2(\underline{W})$. Firstly, we notice that the *exact* computation of $B_Q^\top (C^{(1)})$ and $A_Q C^{(2)}$ takes only $L^{1+o(1)}$ time, by $A_Q \in \mathbb{R}^{r \times d}$, $B_Q \in \mathbb{R}^{d \times r}$, $C^{(1)}$, $C^{(2)} \in \mathbb{R}^{L \times d}$. Thus, to show the existence of $L^{1+o(1)}$ time algorithms for Problem 2, we prove fast low-rank approximations for $B_Q^\top (C^{(1)})^\top p_1(\underline{W}) C^{(2)}$ and $(C^{(1)})^\top p_1(\underline{W}) A_Q C^{(2)}$ by Lemma 3.6. The fast low-rank approximations for $-B_Q^\top (C^{(1)})^\top p_2(\underline{W}) C^{(2)}$ and $-(C^{(1)})^\top p_2(\underline{W}) A_Q C^{(2)}$ follow trivially. See Appendix C.7 for a detailed proof. □

**General Case: Full LoRA Adaptation on $W_K, W_Q, W_V$.** We defer the analysis of full LoRA on transformer ((C2) **General Case:** adapting both $W_K, W_Q, W_V$) to Appendix A due to page limit. Importantly, we also prove the existence of an almost linear-time LoRA (Theorem A.1). In addition, we derive the norm bound conditions required for it to hold.

## 4 NORM-BASED PHASE TRANSITION IN EFFICIENCY

In this section, we characterize the computational limits of all possible efficient algorithms of ALoRAGC, via fine-grained reduction under the Strong Exponential Time Hypothesis (SETH).

**Strong Exponential Time Hypothesis (SETH).** Impagliazzo and Paturi (2001) introduce the Strong Exponential Time Hypothesis (SETH) as a stronger form of the P $\neq$ NP conjecture. It suggests that our current best SAT algorithms are optimal and is a popular conjecture for proving fine-grained lower bounds for a wide variety of algorithmic problems (Williams, 2018b; 2013; Cygan et al., 2016).

---

**Hypothesis 1** (SETH). For every $\epsilon > 0$, there is a positive integer $k \ge 3$ such that $k$-SAT on formulas with $n$ variables cannot be solved in $\mathcal{O}(2^{(1-\epsilon)n})$ time, even by a randomized algorithm.

---

Our primary technique involves casting the ALoRAGC problem (Problem 1) as a fine-grained reduction under SETH, from the hardness result of fast attention approximation algorithm (Alman and Song, 2023). For simplicity of analysis, we consider the special case (C1).

---

**Theorem 4.1** (Inefficient Threshold). Let $\kappa : \mathbb{N} \to \mathbb{N}$ by any function with $\kappa(L) = \omega(1)$ and $\kappa(L) = o(\log L)$. Let $\Gamma = O(\sqrt{\log L} \cdot \kappa(L))$. Assuming Hypothesis 1, there is no algorithm running in time $O(L^{2-\delta})$ for any constant $\delta > 0$ for ALoRAGC$(L, d = O(\log L), r < d, \epsilon)$, i.e., Problem 2, subject to (3.3), even in the case where the input and weight matrices satisfy $\|X^{(K)} W_K^\star\|_\infty \le \Gamma$, $\|\alpha X_i^{(Q)} B_Q A_Q / r\|_\infty \le \Gamma$, $Y = 0$ and $\epsilon = O((\log L)^{-4})$.

---

*Proof Sketch.* Firstly, we recall the hardness of sub-quadratic **Att**ention **G**radient **C**omputation approximation, i.e., AttLGC from (Alman and Song, 2024a) (defined in Definition E.1). This serves as a reference point for the complexity we anticipate for ALoRAGC defined in Problem 2. We then proceed with a reduction from problem AttLGC to problem ALoRAGC. Essentially, by showing that AttLGC is at least as hard as ALoRAGC, and then showing how to solve AttLGC using a solution to ALoRAGC, we establish the hardness of ALoRAGC. See for Appendix E for a detailed proof. □

**Remark 4.1.** Theorem 4.1 suggests an efficiency threshold for $\Gamma$. Only below this threshold are efficient algorithms for ALoRAGC possible. This is a $\Gamma$-based phase transition behavior in efficiency.

**Remark 4.2.** In Theorem 4.1, we show that even the simplest single-data-point case with $Y = 0$ is hard. Hence, our result also applies to the special case (C1) (i.e., Problem 2) and general case (C2) (i.e., Problem 3). Specifically, it is evident that computing the gradient for multiple data points (whether the full gradient or a stochastic mini-batch gradient) is *at least* as hard as for a single data point. The hardness follows trivially.

## 5 DISCUSSION AND CONCLUDING REMARKS

We study the computational limits of the Low-Rank Adaptation (LoRA) for transformer-based model finetuning using fine-grained complexity theory (i.e., under Hypothesis 1). Our main contribution is the proof of the existence of almost linear approximation algorithms for LoRA adaptation on transformer-based models. We accomplish this by utilizing the hierarchical low-rank structures of LoRA gradients (Lemmas 3.3 to 3.5) and approximating the gradients with a series of chained low-rank approximations (Lemmas 3.6 and 3.7). To showcase our theory, we establish such almost linear approximation for both partial (Theorem 3.1) and full LoRA adaptions (Theorem A.1) of attention weights. In addition, we identify a phase transition behavior in the efficiency of all possible variants of LoRA (Theorem 4.1) by adjusting the norm upper-bound $\Gamma$ of input, pretrained, and adaptor weights. Specifically, we establish an "inefficiency threshold" for $\Gamma$, only below which adapting transformer-based models with LoRA in $L^{2-o(1)}$ (sub-quadratic) time is possible.

**Remark 5.1** (General Case: Full LoRA Adaptation on $W_K, W_Q, W_V$). We defer the analysis of full LoRA on transformer (adapting both $W_K, W_Q, W_V$ matrices) to Appendix A due to page limit.

**Remark 5.2** (Insights for Practitionars: Necessary Conditions for Efficient and Robust LoRA). This work is about LoRA on transformer models. Therefore, the computational bottleneck is by design $\mathcal{O}(L^2)$ (see Appendix F for discussions and a proof.) In this regard, our work provides in-depth analysis to address this $\mathcal{O}(L^2)$ bottleneck and provides useful insights and guidance for designing efficient LoRA algorithms and methods with precision guarantees:

- **Theorem 4.1: Necessary Conditions for Subqudratic Time LoRA.** Proper normalization of the composed norms, e.g., $\|X^{(K)}W_K^\star\| \leq \Gamma$ and $\|\alpha X_i^{(Q)}B_Q A_Q/r\| \leq \Gamma$ with $\Gamma = \mathcal{O}(\sqrt{\log L} \cdot \kappa(L))$.

- **Theorems 3.1 and A.1: Necessary Conditions for Almost Linear Time LoRA.** Proper normalization of the composed norms, e.g.,

  - For partial LoRA on $W_Q, W_V$ (Theorem 3.1): $\left\|\frac{\alpha}{r}X^{(Q)}W\right\|_\infty \leq \Gamma$ and $\left\|X^{(K)}W_K^\star\right\|_\infty \leq \Gamma$ with $\Gamma = o(\sqrt{\log L})$.
  - For full LoRA on $W_K, W_Q, W_V$ (Theorem A.1): $\left\|X^{(Q)}\left(W_Q^\star + \frac{\alpha}{r}B_Q A_Q\right)W_K\right\|_\infty \leq \Gamma$, $\left\|X^{(K)}\right\| \leq \Gamma$, $\left\|X^{(Q)}W_Q\right\| \leq \Gamma$, and $\left\|X^{(K)}\left(W_K^\star + \frac{\alpha}{r}B_K A_K\right)\right\|_\infty \leq \Gamma$ with $\Gamma = o(\sqrt{\log L})$.

Suitable normalization of the composed norms can be implemented using pre-activation layer normalization (Xiong et al., 2020; Wang et al., 2019) to control $\|X\|$, or outlier-removing attention activation functions (Hu et al., 2024a) to control $\{\|W_\mu\|, \|A_\mu\|, \|B_\mu\|\}_{\mu=K,Q}$. On one hand, our findings provide formal justifications for these methods. On the other hand, these necessary conditions also motivate the design of future efficient methods with minimal model and data assumptions.

**Remark 5.3** (Self- and Cross-Attention). We emphasize that all these results hold for not only self-attention but also cross-attention due to our generic problem setting (Definition 2.5 and Remark 2.1).

**Proof-of-Concept Experiments.** We provide numerical results to justify our theory in Appendix G.

**Limitations.** We identify necessary conditions for fast LoRA methods, not sufficient conditions. Therefore, our results do not lead to direct implementations. This limitation is inherent to hardness results (Toolkit, 2013). However, as discussed above, we expect our findings to provide valuable insights for future efficient LoRA implementations in both forward and backward computations.

**Impact Statement.** This theoretical work aims to elucidate the foundations of large transformer-based foundation models and is not expected to have negative social impacts.

**Related Works.** We defer the discussion of related works to Appendix B due to page limit.

## ACKNOWLEDGMENTS

JH thanks Mimi Gallagher, Sara Sanchez, Dino Feng, and Andrew Chen for enlightening discussions; Yen-Ju Lu, Shang Wu, Robin Luo, and Jiahao Yu for collaborations on related topics; Shang Wu and authors of (Wu et al., 2024c) for assistance with numerical experiments; and the Red Maple Family for their support. The authors also thank the anonymous reviewers and program chairs for their constructive comments.

JH is partially supported by the Walter P. Murphy Fellowship. HL is partially supported by NIH R01LM1372201, AbbVie and Dolby. EJK thanks the National Center for Theoretical Sciences of Taiwan for funding (112-2124-M-002-003). This research was supported in part through the computational resources and staff contributions provided for the Quest high performance computing facility at Northwestern University which is jointly supported by the Office of the Provost, the Office for Research, and Northwestern University Information Technology. The content is solely the responsibility of the authors and does not necessarily represent the official views of the funding agencies.

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

# Appendix

## A GENERAL CASE: FULL LoRA ADAPTATION ON $W_K$, $W_Q$ AND $W_V$

Similarly, we formulate the full adaptation (C2) of an attention head as the following LoRA loss.

**Definition A.1** (Adapting $W_K$, $W_Q$, $W_V$ of Generic Attention with LoRA). Let $\mathcal{D} = \{(X_i^{(K)}, X_i^{(Q)}, X_i^{(V)}), Y_i\}_{i=1}^N$ be a dataset of size $N$ with the triplet $X_i^{(K)}, X_i^{(Q)}, X_i^{(V)} \in \mathbb{R}^{L \times d}$ being the input and $Y_i \in \mathbb{R}^{L \times d}$ being the label. The problem of fine-tuning a generic attention with LoRA with $\ell_2$ loss from dataset $\mathcal{D}$ is formulated as

$$\min_{\substack{B_K, B_Q, B_V \in \mathbb{R}^{d \times r}, \\ A_K, A_Q, A_V \in \mathbb{R}^{r \times d}}} \mathcal{L}(W_K = W_K^\star + \frac{\alpha}{r} B_K A_K, W_Q = W_Q^\star + \frac{\alpha}{r} B_Q A_Q, W_V = W_V^\star + \frac{\alpha}{r} B_V A_V)$$

$$:= \frac{1}{2N} \sum_{i=1}^N \left\| D^{-1} \exp\left\{ X_i^{(Q)} W_Q W_K^\mathsf{T} X_i^{(K)} \beta \right\} X_i^{(V)} W_V - Y_i \right\|_F^2.$$

Here $D := \mathrm{diag}(\exp\{X^{(Q)} W_Q W_K^\mathsf{T} X^{(K)} \beta\} \mathbb{1}_n) \in \mathbb{R}^{L \times L}$.

By simplifications (S1), (S3) and (S4), we fix $W_V$, set $\beta = \alpha/r = 1$ and consider LoRA adaptation on a single data point. Akin to simplification (S2), we introduce $C_K^{(1)}, C_K^{(2)}, C_Q^{(1)}, C_Q^{(2)}, C^{(3)} \in \mathbb{R}^{L \times d}$:

$$C_K^{(1)} := X^{(Q)} \left( W_Q^\star + \frac{\alpha}{r} B_Q A_Q \right), \quad C_K^{(2)} := X^{(K)}, \tag{A.1}$$

$$C_Q^{(1)} := X^{(Q)}, \quad C_Q^{(2)} := X^{(K)} (W_K^\star + B_K A_K), \quad \text{and} \quad C^{(3)} := X^{(V)} W_V^\star.$$

**Remark A.1.** $C_K^{(1)}, C_K^{(2)}, C^{(3)}$ are constants with respect to adapting $B_K, A_K$ with gradient updates. $C_Q^{(1)}, C_Q^{(2)}, C^{(3)}$ are constants with respect to adapting $B_Q, A_Q$ with gradient updates.

Therefore, the full LoRA adaptation loss in Definition A.1 becomes

$$\min_{\substack{B_K, B_Q \subset \mathbb{R}^{d \times r} \\ A_K, A_Q \subset \mathbb{R}^{r \times d}}} \left\| D^{-1} \exp\left\{ X^{(Q)} \left( W_Q^\star + B_Q A_Q \right) \left( W_K^\star + B_K A_K \right)^\top \left( X^{(K)} \right)^\top \right\} X^{(V)} W_V^\star - Y \right\|_F^2,$$

$$\tag{A.2}$$

where $D = \mathrm{diag}\left( \exp\left( C_K^{(1)} (W_K^\star + B_K A_K)^\top (C_K^{(2)})^\top \right) \mathbb{1}_L \right) = \mathrm{diag}\left( \exp\left( C_Q^{(1)} (W_Q^\star + B_Q A_Q)(C_Q^{(2)})^\top \right) \mathbb{1}_L \right) \in \mathbb{R}^{L \times L}$.

Similar to Section 3, we introduce the following problem to characterize all possible gradient computation of (A.2), and arrive similar results as Section 3: almost linear algorithm for Problem 3.

**Problem 3** (Approximate LoRA Gradient Computation (ALoRAGC$(L, d, r, \epsilon)$)). Assume all numerical values be in $\log(L)$ bits encoding. Let $\mathcal{L}$ follow (A.2), $\epsilon > 0$, and $\|Z\|_\infty := \max_{i,j} |Z_{ij}|$. The problem of approximating gradient computation of optimizing (A.2) is to find four surrogate gradient matrices $\{\widetilde{G}_\mu^{(A)} \in \mathbb{R}^{d \times r}, \widetilde{G}_\mu^{(B)} \in \mathbb{R}^{r \times d}\}_{\mu = K, Q}$ such that

$$\max \left( \left\{ \left\| \widetilde{\underline{G}}_\mu^{(B)} - \frac{\partial \mathcal{L}}{\partial \underline{B}_Q} \right\|_\infty, \left\| \widetilde{\underline{G}}_\mu^{(A)} - \frac{\partial \mathcal{L}}{\partial \underline{A}_Q} \right\|_\infty \right\}_{\mu = K, Q} \right) \leq \epsilon.$$

**Theorem A.1** (Main Result: Existence of Almost Linear Time ALoRAGC). Let $\Gamma = o(\sqrt{\log L})$. Suppose all numerical values are in $O(\log L)$-bits encoding. For $\mu = Q, K$, let $W_\mu = W_\mu^\star + B_\mu A_\mu \in \mathbb{R}^{d \times d}$. If $\left\| C_\mu^{(1)} W_\mu \right\|_\infty \leq \Gamma$ and $\left\| C_\mu^{(2)} \right\|_\infty \leq \Gamma$ for both $\mu = Q, K$, then there exists a $L^{1+o(1)}$ time algorithm to solve ALoRAGC$(L, d = O(\log L), r = L^{o(1)}, \epsilon = 1/\mathrm{poly}(L))$ (i.e., Problem 3) up to $1/\mathrm{poly}(L)$ accuracy. In particular, this algorithm outputs gradient matrices $\{\widetilde{G}_\mu^{(A)} \in \mathbb{R}^{d \times r}, \widetilde{G}_\mu^{(B)} \in \mathbb{R}^{r \times d}\}_{\mu = K, Q}$ such that

$$\max \left( \left\{ \left\| \frac{\partial \mathcal{L}}{\partial \underline{B}_\mu} - \widetilde{\underline{G}}_\mu^{(A)} \right\|_\infty, \left\| \frac{\partial \mathcal{L}}{\partial \underline{A}_\mu} - \widetilde{\underline{G}}_\mu^{(A)} \right\|_\infty \right\}_{\mu = K, Q} \right) \leq 1/\mathrm{poly}(L).$$

*Proof.* See Appendix D for a detailed proof. □

## B  RELATED WORKS

**Fine-Grained Complexity.**  The Strong Exponential Time Hypothesis (SETH) is a conjecture in computational complexity theory that posits solving the Boolean satisfiability problem (SAT) for $n$ variables requires time $2^n$ in the worst case, up to sub-exponential factors (Impagliazzo and Paturi, 2001). It extends the Exponential Time Hypothesis (ETH) by suggesting that no algorithm can solve $k$-SAT in $O(2^{(1-\epsilon)n})$ time for any $\epsilon > 0$ (Calabro et al., 2009). SETH has significant implications for the hardness of various computational problems, as proving or disproving it would greatly enhance our understanding of computational limits (Williams, 2018b; 2013).

In essence, SETH is a stronger form of the $P \neq NP$ conjecture, suggesting that our current best SAT algorithms are optimal. It states as follows:

**Hypothesis 2** (SETH).  For every $\epsilon > 0$, there is a positive integer $k \geq 3$ such that $k$-SAT on formulas with $n$ variables cannot be solved in $\mathcal{O}(2^{(1-\epsilon)n})$ time, even by a randomized algorithm.

SETH is widely used for establishing fine-grained lower bounds for various algorithmic challenges, including $k$-Hitting Set and $k$-NAE-SAT (Williams, 2018b; Cygan et al., 2016). This conjecture is crucial in deriving conditional lower bounds for many significant problems that otherwise have polynomial-time solutions in diverse fields such as pattern matching (Chen and Williams, 2019; Bringman and Künnemann, 2018; Bringmann et al., 2017; Bringmann and Mulzer, 2016; Backurs and Indyk, 2016; Bringmann, 2014; Abboud et al., 2014), graph theory (Dalirrooyfard et al., 2022; Chan et al., 2022; Abboud et al., 2018; Gao et al., 2018; Krauthgamer and Trabelsi, 2018; Roditty and Vassilevska Williams, 2013), and computational geometry (Karthik and Manurangsi, 2020; Williams, 2018a; Rubinstein, 2018; Chen, 2018; Buchin et al., 2016).

Based on this conjecture, our study employs fine-grained reductions under SETH to explore the computational limits of Low-Rank Adaptation (LoRA). Previous research in fine-grained reductions includes the work by Backurs et al. (2017), who examine the computational complexity of various Empirical Risk Minimization problems, such as kernel SVMs and kernel ridge. Alman et al. (2020) investigate the effectiveness of spectral graph theory on geometric graphs within the constraints of SETH. Aggarwal and Alman (2022) address the computational limitations of Batch Gaussian Kernel Density Estimation. Expanding on these studies, Gu et al. (2024a;b); Alman and Song (2024b; 2023) explore transformer attention and introduced a tensor generalization. Alman and Yu (2024) establish the fundamental limitations on subquadratic alternatives to softmax transformers. Hu et al. (2024c) show that efficient dense associative memory a.k.a. modern Hopfield models and corresponding networks also need bounded query and key patterns for sub-quadratic time complexity. Compared to existing works, this work is, to the best of our knowledge, the first analysis of computational limits for parameter-efficient fine-tuning of large foundation models (Hu et al., 2021).

**Low-Rank Adaptation (LoRA).**  In this paper, we focus on LoRA (Hu et al., 2021), a method that leverages low-rank matrices to approximate updates to the weights of neural models. Various extensions of LoRA have been proposed to address different challenges in model training and deployment. For instance, DoRA (Liu et al., 2024) focus on enhanced parameter efficiency. QLoRA (Dettmers et al., 2024), LoftQ (Li et al., 2024), QA-LoRA (Xu et al., 2024b), and LQ-LoRA (Guo et al., 2024) focus on both memory and parameter efficiency in model compression and quantization. Additionally, DyLoRA (Li et al., 2020), AdaLoRA (Zhang et al., 2023), and SoRA (Ding et al., 2023) focus on dynamically determining the optimal rank $r$ for LoRA implementations. LoRAHub (Huang et al., 2023) focus on multi-task finetuning. LoRA+ (Hayou et al., 2024) focus on efficient feature learning. Despite the methodological and empirical successes, the theoretical side is relatively underdeveloped. While Zeng and Lee (2024) explore the expressiveness of LoRA from a universal-approximation perspective, and Hayou et al. (2024) investigate the optimal adapter learning rate with respect to large model width, to the best of our knowledge, no existing analysis focuses on the computational limits of LoRA. Therefore, this work provides a timely theoretical analysis of LoRA's computational limits, aiming to advance efficient finetuning of large foundation models in terms of both parameter usage and computational time.

**Outliers in Attention Heads.**  Our results indicate that outliers (e.g., large $\|XW^\star\|$ and $\|XW^\star + \alpha XBA/r\|$) in attention heads hamper LoRA efficiency and performance. This outlier effect is well-known in pretraining large foundation models for its negative impact on models' quantization

performance (Sun et al., 2024). For pretraining, prior works identify the existence of no-op tokens as the main source: tokens with small value vectors tend to receive significantly large attention weights (Hu et al., 2024a; Bondarenko et al., 2023). Specifically, Hu et al. (2024a) interpret this outlier effect as inefficient *rare* memory retrieval from the associative memory/modern Hopfield model perspective (Wu et al., 2024a;b; Xu et al., 2024a; Hu et al., 2025; 2024b;c; 2023) and propose the outlier-efficient Hopfield layer for transformer-based large models, demonstrating strong empirical performance and theoretical guarantees. The advantages of controlling outliers in the attention heads of transformer-based large foundation models are also emphasized in various theoretical studies (Gu et al., 2024a;b; Alman and Song, 2024a;b; 2023; Gao et al., 2023a). Yet, to the best of our knowledge, there is no existing work on outliers in LoRA fine-tuning. This is the first work establishing that the LoRA adaptor weights might lead to performance and efficiency degradation due to their additive nature: $\|XW^\star + \alpha XBA/r\|$.

## C  PROOFS OF SECTION 3

### C.1  PROOF OF LEMMA 3.1

*Proof of Lemma 3.1.*  With LoRA loss (3.3), we have

$$\frac{\mathrm{d}\mathcal{L}(\underline{W})}{\mathrm{d}\underline{W}} = \sum_{\underline{j}=1}^{L} \sum_{i=1}^{d} \frac{\mathrm{d}}{\mathrm{d}\underline{W}_i} \left( \frac{1}{2} c(\underline{W})_{\underline{j},i}^2 \right).$$

Note that for each $\underline{j} \in [L]$ and $i \in [d]$,

$$\frac{\mathrm{d}}{\mathrm{d}\underline{W}_i} \left( \frac{1}{2} c(\underline{W})_{\underline{j},i}^2 \right) \qquad\qquad\qquad\qquad\qquad\qquad (\text{By (3.3)})$$

$$= c(\underline{W})_{\underline{j},i} \frac{\mathrm{d}\left\langle f(\underline{W})_{\underline{j}}, C^{(3)}[\cdot,i] \right\rangle}{\mathrm{d}\underline{W}_i} \qquad\qquad\qquad (\text{By Definition 3.5})$$

$$= c(\underline{W})_{\underline{j},i} \left\langle \frac{\mathrm{d}f(\underline{W})_{\underline{j}}}{\mathrm{d}\underline{W}_i}, C^{(3)}[\cdot,i] \right\rangle$$

$$= c(\underline{W})_{\underline{j},i} \left\langle \frac{\mathrm{d}\left( \alpha^{-1}(\underline{W})_{\underline{j}} u(\underline{W})_{\underline{j}} \right)}{\mathrm{d}\underline{W}_i}, C^{(3)}[\cdot,i] \right\rangle \qquad (\text{By Definition 3.4})$$

$$= c(\underline{W})_{\underline{j},i} \left\langle \alpha(\underline{W})_{\underline{j}}^{-1} \underbrace{\frac{\mathrm{d}u(\underline{W})_{\underline{j}}}{\mathrm{d}\underline{W}_i}}_{(I)} - \alpha(\underline{W})_{\underline{j}}^{-2} \underbrace{\frac{\mathrm{d}\alpha(\underline{W})_{\underline{j}}}{\mathrm{d}\underline{W}_i}}_{(II)} u(\underline{W})_{\underline{j}}, C^{(3)}[\cdot,i] \right\rangle.$$

$$(\text{By product rule and then chain rule})$$

- **Part (I).** We have

$$\frac{\mathrm{d}u(\underline{W})_{\underline{j}}}{\mathrm{d}\underline{W}_i} = \frac{\mathrm{d}\exp\left( \mathsf{C}_{\underline{j}}\underline{W} \right)}{\mathrm{d}\underline{W}_i} \qquad\qquad\qquad (\text{By Definition 3.2})$$

$$= \exp\left( \mathsf{C}_{\underline{j}}\underline{W} \right) \odot \frac{\mathrm{d}\mathsf{C}_{\underline{j}}\underline{W}}{\mathrm{d}\underline{W}_i}$$

$$= \mathsf{C}_{\underline{j}}[\cdot,i] \odot u(\underline{W})_{\underline{j}}. \qquad (\text{By } \frac{\mathrm{d}\left( \mathsf{C}_{\underline{j}}\underline{W} \right)}{\mathrm{d}\underline{W}_i} = \frac{\mathrm{d}\mathsf{C}_{\underline{j}}\underline{W}}{\mathrm{d}\underline{W}_i} = \mathsf{C}_{\underline{j}} \cdot \frac{\mathrm{d}\underline{W}}{\mathrm{d}\underline{W}_i} = \mathsf{C}_{\underline{j}} \cdot e_i = \left( \mathsf{C}_{\underline{j}} \right)[\cdot,i])$$

- **Part (II).** We have

$$\frac{\mathrm{d}\alpha(\underline{W})_{\underline{j}}}{\mathrm{d}\underline{W}_i} = \frac{\mathrm{d}\left\langle u(\underline{W})_{\underline{j}}, \mathbb{1}_L \right\rangle}{\mathrm{d}\underline{W}_i} \qquad\qquad\qquad (\text{By Definition 3.3})$$

$$= \left\langle \mathsf{C}_{\underline{j}}[\cdot,i] \odot u(\underline{W})_{\underline{j}}, \mathbb{1}_L \right\rangle \qquad\qquad (\text{By Definition 3.2})$$

$$= \left\langle \mathsf{C}_{\underline{j}}[\cdot,i], u(\underline{W})_{\underline{j}} \right\rangle. \qquad\qquad (\text{By element-wise product identity})$$

Combining (I) and (II), we get

$$\frac{\mathrm{d}}{\mathrm{d}\underline{W}_i} \left( \frac{1}{2} c(\underline{W})_{\underline{j},i}^2 \right)$$

$$= c(\underline{W})_{\underline{j},i} \left[ \left\langle C^{(3)}[\cdot,i], \mathsf{C}_{\underline{j}}[\cdot,i] \odot f(\underline{W})_{\underline{j}} \right\rangle - \left\langle C^{(3)}[\cdot,i], f(\underline{W})_{\underline{j}} \right\rangle \cdot \left\langle \mathsf{C}_{\underline{j}}[\cdot,i], f(\underline{W})_{\underline{j}} \right\rangle \right]$$

$$= c(\underline{W})_{\underline{j},i} \mathsf{C}_{\underline{j}}^{\top} \left( \mathrm{diag}\left( f(\underline{W})_{\underline{j}} \right) - f(\underline{W})_{\underline{j}} f(\underline{W})_{\underline{j}}^{\top} \right) C^{(3)}[\cdot,i].$$

This completes the proof. $\qquad\qquad\qquad\qquad\qquad\qquad\qquad\qquad\qquad\qquad\qquad \square$

## C.2 PROOF OF LEMMA 3.2

First, we present a helper lemma.

**Lemma C.1.** For any $a \in \mathbb{R}$, let $\mathrm{diag}_d(a) \in \mathbb{R}^{d \times d}$ be a $d \times d$ diagonal matrix with all entries equal to $a$. Let $J_B, J_A \in \mathbb{R}^{d^2 \times rd}$ be two matrices such that $\underline{W} = \overline{W}_Q^\star + J_B \underline{A}_Q$, and $\underline{W} = \overline{W}_Q^\star + J_A \underline{B}_Q$ via

$$
J_B = \begin{pmatrix} B_Q & & & \\ & B_Q & & \\ & & \ddots & \\ & & & B_Q \end{pmatrix}, J_A = \begin{pmatrix} \mathrm{diag}_d\left(A_Q[1,1]\right) & \cdots & \mathrm{diag}_d\left(A_Q[r,1]\right) \\ \mathrm{diag}_d\left(A_Q[1,2]\right) & \cdots & \mathrm{diag}_d\left(A_Q[r,2]\right) \\ \vdots & & \vdots \\ \mathrm{diag}_d\left(A_Q[1,d]\right) & \cdots & \mathrm{diag}_d\left(A_Q[r,d]\right) \end{pmatrix}
$$

The derivatives of loss function (3.3) w.r.t. $A_Q, B_Q$ are therefore

$$
\frac{\partial \mathcal{L}}{\partial \underline{A}_Q} = \sum_{\underline{j}=1}^{L} \sum_{i=1}^{d} J_B^\top c(\underline{W})_{\underline{j},i} \mathsf{C}_{\underline{j}}^\top \left( \mathrm{diag}\left( f(\underline{W})_{\underline{j}} \right) - f(\underline{W})_{\underline{j}} f(\underline{W})_{\underline{j}}^\top \right) C^{(3)}[\cdot, i],
$$

$$
\frac{\partial \mathcal{L}}{\partial \underline{B}_Q} = \sum_{\underline{j}=1}^{L} \sum_{i=1}^{d} J_A^\top c(\underline{W})_{\underline{j},i} \mathsf{C}_{\underline{j}}^\top \left( \mathrm{diag}\left( f(\underline{W})_{\underline{j}} \right) - f(\underline{W})_{\underline{j}} f(\underline{W})_{\underline{j}}^\top \right) C^{(3)}[\cdot, i].
$$

*Proof.* The proof follows standard chain-rule and Lemma 3.1. □

Then, we prove Lemma 3.2.

*Proof of Lemma 3.2.* From Lemma C.1, we have

$$
\frac{\partial \mathcal{L}}{\partial \underline{A}_Q} = \sum_{\underline{j}=1}^{L} \sum_{i=1}^{d} J_B^\top c(\underline{W})_{\underline{j},i} \mathsf{C}_{\underline{j}}^\top \left( \mathrm{diag}\left( f(\underline{W})_{\underline{j}} \right) - f(\underline{W})_{\underline{j}} f(\underline{W})_{\underline{j}}^\top \right) C^{(3)}[\cdot, i]
$$

$$
= \sum_{\underline{j}=1}^{L} J_B^\top \mathsf{C}_{\underline{j}}^\top \left( \mathrm{diag}\left( f(\underline{W})_{\underline{j}} \right) - f(\underline{W})_{\underline{j}} f(\underline{W})_{\underline{j}}^\top \right) q(\underline{W})_{\underline{j}}
$$

$$
\left( \text{By } q(\underline{W}) := C^{(3)} \left( c(\underline{W}) \right)^\top \in \mathbb{R}^{L \times L} \right)
$$

$$
= \sum_{\underline{j}=1}^{L} J_B^\top \mathsf{C}_{\underline{j}}^\top p(\underline{W})_{\underline{j}} \qquad\qquad (\text{By Definition 3.6})
$$

$$
= \mathrm{vec}\left( B_Q^\top \left( C^{(1)} \right)^\top p(\underline{W}) C^{(2)} \right).
$$

Similarly,

$$
\frac{\partial \mathcal{L}}{\partial \underline{B}_Q} = \sum_{j=1}^{L} \sum_{i=1}^{d} J_A^\top c(\underline{W})_{\underline{j},i} \mathsf{C}_{\underline{j}}^\top \left( \mathrm{diag}\left( f(\underline{W})_{\underline{j}} \right) - f(\underline{W})_{\underline{j}} f(\underline{W})_{\underline{j}}^\top \right) C^{(3)}[\cdot, i]
$$

$$
= \sum_{\underline{j}=1}^{L} J_A^\top \mathsf{C}_{\underline{j}}^\top \left( \mathrm{diag}\left( f(\underline{W})_{\underline{j}} \right) - f(\underline{W})_{\underline{j}} f(\underline{W})_{\underline{j}}^\top \right) q(\underline{W})_{\underline{j}}
$$

$$
\left( \text{By } q(\underline{W}) := C^{(3)} \left( c(\underline{W}) \right)^\top \in \mathbb{R}^{L \times L} \right)
$$

$$
= \sum_{\underline{j}=1}^{L} J_A^\top \mathsf{C}_{\underline{j}}^\top p(\underline{W})_{\underline{j}} \qquad\qquad (\text{By Definition 3.6})
$$

$$
= \mathrm{vec}\left( \left( C^{(1)} \right)^\top p(\underline{W}) A_Q C^{(2)} \right).
$$

$$
\left( \text{By } J_B^\top C_{\underline{j}}^\top = \left( C^{(1)} B_Q \otimes C^{(2)} \right)^\top, \text{ and } J_A^\top C_{\underline{j}}^\top = \left( C^{(1)} \otimes A_Q C^{(2)} \right)^\top \right)
$$

This completes the proof. □

### C.3 PROOF OF LEMMA 3.4

*Proof of Lemma 3.4.* Our proof is built on (Alman and Song, 2023, Lemma D.2). By definitions,

$$
\left\| U_1 V_1^\top C^{(3)} - Y - c(\underline{W}) \right\|_\infty
$$
$$
= \left\| U_1 V_1^\top C^{(3)} - Y - f(\underline{W}) C^{(3)} + Y \right\|_\infty \qquad \text{(By } c(\underline{W}) = f(\underline{W}) C^{(3)} - Y)
$$
$$
= \left\| \left( U_1 V_1^\top - f(\underline{W}) \right) C^{(3)} \right\|_\infty
$$
$$
\leq \epsilon/\text{poly}(L). \qquad \text{(By (Alman and Song, 2023, Lemma D.2))}
$$

This completes the proof. $\qquad\square$

### C.4 PROOF OF LEMMA 3.5

*Proof of Lemma 3.5.* Our proof is built on (Alman and Song, 2023, Lemma D.3).

Let $\widetilde{q}(\underline{W})$ denote an approximation to $q(\underline{W})$. By Lemma 3.4, $U_1 V_1^\top C^{(3)} - Y$ approximates $c(\underline{W})$ with a controllable error.

Then, by setting

$$
\widetilde{q}(\underline{W}) = C^{(3)} \left( U_1 V_1^\top C^{(3)} - Y \right)^\top,
$$

we turn $\widetilde{q}(\underline{W})$ into some low-rank representation

$$
\widetilde{q}(\underline{W}) = C^{(3)} \left( C^{(3)} \right)^\top V_1 U_1^\top - C^{(3)} Y^\top.
$$

By $k_1, d = L^{o(1)}$, it is obvious that computing $\underbrace{\left( C^{(3)} \right)^\top}_{d \times L} \underbrace{V_1}_{L \times k_1} \underbrace{U_1^\top}_{k_1 \times L}$ only takes $L^{1+o(1)}$ time.

Then we can explicitly construct $U_2, V_2 \in \mathbb{R}^{L \times k_2}$ in $L^{1+o(1)}$ time as follows:

$$
U_2 := \underbrace{\left( C^{(3)} \quad -C^{(3)} \right)}_{L \times (d+d)} \in \mathbb{R}^{L \times k_2}, \quad V_2 := \underbrace{\left( U_1 V_1^\top C^{(3)} \quad Y \right)}_{L \times (d+d)} \in \mathbb{R}^{L \times k_2},
$$

with $k_2 = 2d = L^{o(1)}$ by $d = O(\log L)$. This leads to

$$
\widetilde{q}(\underline{W}) = \left( C^{(3)} \quad -C^{(3)} \right) \left( \begin{matrix} \left( C^{(3)} \right)^\top V_1 U_1^\top \\ Y^\top \end{matrix} \right) = U_2 V_2^\top.
$$

Therefore, for controlling the approximation error, it holds

$$
\| \widetilde{q}(\underline{W}) - q(\underline{W}) \|_\infty = \left\| C^{(3)} \left( U_1 V_1^\top C^{(3)} - Y \right)^\top - C^{(3)} Y^\top \right\|_\infty
$$
$$
\leq d \left\| C^{(3)} \right\|_\infty \left\| U_1 V_1^\top C^{(3)} - Y - c(\underline{W}) \right\|_\infty
$$
$$
\leq \epsilon/\text{poly}(L). \qquad \text{(By Lemma 3.4)}
$$

Thus, we complete the proof. $\qquad\square$

## C.5   PROOF OF LEMMA 3.6

*Proof of Lemma 3.6.* We proceed the proof by constructing low-rank approximation of $p_1(\cdot)$ with decomposing $p_1(\cdot)$ into $f(\cdot)$ and $q(\cdot)$ through tensor formulation, and then approximating $p_1$ part by part.

We denote $\oslash$ for *column-wise* Kronecker product such that $A \oslash B := [A[\cdot, 1] \otimes B[\cdot, 1] \mid \ldots \mid A[\cdot, k_1] \otimes B[\cdot, k_1]] \in \mathbb{R}^{L \times k_1 k_2}$ for $A \in \mathbb{R}^{L \times k_1}, B \in \mathbb{R}^{L \times k_2}$.

Let $\widetilde{f}(\underline{W}) := U_1 V_1^{\mathsf{T}}$ and $\widetilde{q}(\underline{W}) := U_2 V_2^{\mathsf{T}}$ denote matrix-multiplication approximations to $f(\underline{W})$ and $q(\underline{W})$, respectively.

For the case of presentation, let $U_3 = \overbrace{U_1}^{L \times k_1} \oslash \overbrace{U_2}^{L \times k_2}$ and $V_3 = \overbrace{V_1}^{L \times k_1} \oslash \overbrace{V_2}^{L \times k_2}$. It holds

$$
\begin{aligned}
&\left\| U_3 V_3^\top - p_1(\underline{W}) \right\|_\infty \\
=\ & \left\| U_3 V_3^\top - f(\underline{W}) \odot q(\underline{W}) \right\|_\infty && \big(\text{By } p_1(\underline{W}) = f(\underline{W}) \odot q(\underline{W})\big) \\
=\ & \left\| (U_1 \oslash U_2)(V_1 \oslash V_2)^\top - f(\underline{W}) \odot q(\underline{W}) \right\|_\infty \\
=\ & \left\| (U_1 V_1^\top) \odot (U_2 V_2^\top) - f(\underline{W}) \odot q(\underline{W}) \right\|_\infty \\
=\ & \| \widetilde{f}(\underline{W}) \odot \widetilde{q}(\underline{W}) - f(\underline{W}) \odot q(\underline{W}) \|_\infty \\
\leq\ & \| \widetilde{f}(\underline{W}) \odot \widetilde{q}(\underline{W}) - \widetilde{f}(\underline{W}) \odot q(\underline{W}) \|_\infty + \| \widetilde{f}(\underline{W}) \odot q(\underline{W}) - f(\underline{W}) \odot q(\underline{W}) \|_\infty \\
&&& \big(\text{By triangle inequality}\big) \\
\leq\ & \epsilon/\mathrm{poly}(L). && \big(\text{By Lemma 3.3 and Lemma 3.5}\big)
\end{aligned}
$$

Computationally, by $k_1, k_2 = L^{o(1)}$, computing $U_3$ and $V_3$ takes $L^{1+o(1)}$ time.

This completes the proof. $\qquad\qquad\square$

## C.6 Proof of Lemma 3.7

*Proof of Lemma 3.7.* By considering the following decomposition through tensor formulation

$$p_2(\underline{W})_{\underline{j}} := \overbrace{f(\underline{W})_{\underline{j}} \underbrace{f(\underline{W})_{\underline{j}}^\top q(\underline{W})_{\underline{j}}}_{(I)}}^{(II)},$$

we approximate the $p_2(\cdot)$ part by part. Specifically, for (I), we show its low-rank approximation by observing the low-rank-preserving property of the multiplication between $f(\cdot)$ and $q(\cdot)$ (from Lemma 3.3 and Lemma 3.5). For (II), we show its low-rank approximation by the low-rank structure of $f(\cdot)$ and (I).

**Part (I).** We define a function $r(\underline{W}) : \mathbb{R}^{d^2} \to \mathbb{R}^L$ such that the $\underline{j}$-th component $r(\underline{W})_{\underline{j}} := \left( f(\underline{W})_{\underline{j}} \right)^\top q(\underline{W})_{\underline{j}}$ for all $\underline{j} \in [L]$. Let $\widetilde{r}(\underline{W})$ denote the approximation of $r(\underline{W})$ via decomposing into $f(\cdot)$ and $q(\cdot)$:

$$\widetilde{r}(\underline{W})_{\underline{j}} := \left\langle \widetilde{f}(\underline{W})_{\underline{j}}, \widetilde{q}(\underline{W})_{\underline{j}} \right\rangle = \left( U_1 V_1^\top \right) [\underline{j}, \cdot] \cdot \left[ \left( U_2 V_2^\top \right) [\underline{j}, \cdot] \right]^\top$$

$$= U_1[\underline{j}, \cdot] \underbrace{V_1^\top}_{k_1 \times L} \underbrace{V_2}_{L \times k_2} \left( U_2[\underline{j}, \cdot] \right)^\top, \tag{C.1}$$

for all $\underline{j} \in [L]$. This allows us to write $p_2(\underline{W}) = f(\underline{W}) \operatorname{diag}(r(\underline{W}))$ with $\operatorname{diag}(\widetilde{r}(\underline{W}))$ denoting a diagonal matrix with diagonal entries being components of $\widetilde{r}(\underline{W})$.

**Part (II).** With $r(\cdot)$, we approximate $p_2(\cdot)$ with $\widetilde{p}_2(\underline{W}) = \widetilde{f}(\underline{W}) \operatorname{diag}(\widetilde{r}(\underline{W}))$ as follows.

Since $\widetilde{f}(\underline{W})$ has low rank representation, and $\operatorname{diag}(\widetilde{r}(\underline{W}))$ is a diagonal matrix, $\widetilde{p}_2(\cdot)$ has low-rank representation by definition. Thus, we set $\widetilde{p}_2(\underline{W}) = U_4 V_4^\top$ with $U_4 = U_1$ and $V_4 = \operatorname{diag}(\widetilde{r}(\underline{W}))V_1$. Then, we bound the approximation error

$$\left\| U_4 V_4^\top - p_2(\underline{W}) \right\|_\infty$$
$$= \left\| \widetilde{p}_2(\underline{W}) - p_2(\underline{W}) \right\|_\infty$$
$$= \max_{\underline{j} \in [L]} \left\| \widetilde{f}(\underline{W})_{\underline{j}} \widetilde{r}(\underline{W})_{\underline{j}} - f(\underline{W})_{\underline{j}} r(\underline{W})_{\underline{j}} \right\|_\infty$$
$$\leq \max_{\underline{j} \in [L]} \left[ \left\| \widetilde{f}(\underline{W})_{\underline{j}} \widetilde{r}(\underline{W})_{\underline{j}} - f(\underline{W})_{\underline{j}} r(\underline{W})_{\underline{j}} \right\|_\infty + \left\| \widetilde{f}(\underline{W})_{\underline{j}} \widetilde{r}(\underline{W})_{\underline{j}} - f(\underline{W})_{\underline{j}} r(\underline{W})_{\underline{j}} \right\|_\infty \right]$$
$$\text{(By triangle inequality)}$$
$$\leq \epsilon / \operatorname{poly}(L).$$

Computationally, computing $V_1^\top V_2$ takes $L^{1+o(1)}$ time by $k_1, k_2 = L^{o(1)}$.

Once we have $V_1^\top V_2$ precomputed, (C.1) only takes $O(k_1 k_2)$ time for each $\underline{j} \in [L]$. Thus, the total time is $O(L k_1 k_2) = L^{1+o(1)}$. Since $U_1$ and $V_1$ takes $L^{1+o(1)}$ time to construct and $V_4 = \underbrace{\operatorname{diag}(\widetilde{r}(\underline{W}))}_{L \times L} \underbrace{V_1}_{L \times k_1}$ also takes $L^{1+o(1)}$ time, $U_4$ and $V_4$ takes $L^{1+o(1)}$ time to construct.

This completes the proof. $\square$

## C.7 PROOF OF THEOREM 3.1

*Proof of Theorem 3.1.* By the definitions of matrices $p(\underline{W})$ (Lemma 3.2), $p_1(\underline{W})$ and $p_2(\underline{W})$ (Definition 3.6), we have $p(\underline{W}) = p_1(\underline{W}) - p_2(\underline{W})$.

By Lemma 3.2, we have

$$\frac{\partial \mathcal{L}}{\partial \underline{A}_Q} = \mathrm{vec}\left(B_Q^\top \left(C^{(1)}\right)^\top p(\underline{W})C^{(2)}\right), \quad \frac{\partial \mathcal{L}}{\partial \underline{B}_Q} = \mathrm{vec}\left(\left(C^{(1)}\right)^\top p(\underline{W})A_Q C^{(2)}\right). \quad \text{(C.2)}$$

Firstly, we note that the *exact* computation of $B_Q^\top \left(C^{(1)}\right)$ and $A_Q C^{(2)}$ takes $L^{1+o(1)}$ time, by $A_Q \in \mathbb{R}^{r \times d}, B_Q \in \mathbb{R}^{d \times r}, C^{(1)}, C^{(2)} \in \mathbb{R}^{L \times d}$. Therefore, to show the existence of $L^{1+o(1)}$ algorithms for Problem 2, we prove fast low-rank approximations for $B_Q^\top \left(C^{(1)}\right)^\top p_1(\underline{W})C^{(2)}$ and $\left(C^{(1)}\right)^\top p_1(\underline{W})A_Q C^{(2)}$ as follows. The fast low-rank approximations for $-B_Q^\top \left(C^{(1)}\right)^\top p_2(\underline{W})C^{(2)}$ and $-\left(C^{(1)}\right)^\top p_2(\underline{W})A_Q C^{(2)}$ trivially follow.

**Fast Approximation for $B_Q^\top \left(C^{(1)}\right)^\top p_1(\underline{W})C^{(2)}$.** Using $\widetilde{p}_1(\underline{W}), \widetilde{p}_2(\underline{W})$ as the approximations to $p_1(\underline{W}), p_2(\underline{W})$, by Lemma 3.6, it takes $L^{1+o(1)}$ time to construct $U_3, V_3 \in \mathbb{R}^{L \times k_3}$ subject to

$$B_Q^\top \left(C^{(1)}\right)^\top \widetilde{p}_1(\underline{W})C^{(2)} = B_Q^\top \left(C^{(1)}\right)^\top U_3 V_3^\top C^{(2)}.$$

Then we compute $\overbrace{B_Q^\top}^{r \times d} \overbrace{\left(C^{(1)}\right)^\top}^{d \times L} \overbrace{U_3}^{L \times k_3}, \overbrace{V_3^\top}^{k_3 \times L} \overbrace{C^{(2)}}^{L \times d}$. By $r, d, k_1, k_3 = L^{o(1)}$, this takes $L^{1+o(1)}$ time.

Finally we compute $\overbrace{\left(B_Q^\top \left(C^{(1)}\right)^\top U_3\right)}^{r \times k_3} \overbrace{\left(V_3^\top C^{(2)}\right)}^{k_3 \times d}$. By $r, d, k_1, k_3 = L^{o(1)}$, this takes $L^{1+o(1)}$ time. So, overall running time is still $L^{1+o(1)}$.

**Fast Approximation for $\left(C^{(1)}\right)^\top p_1(\underline{W})A_Q C^{(2)}$.** Similarly, computing $\left(C^{(1)}\right)^\top p_1(\underline{W})A_Q C^{(2)}$ takes $L^{1+o(1)}$ time.

**Fast Approximation for (C.2).** Notably, above results hold for both $p_2(x)$ and $p_1(x)$. Therefore, computing $B_Q^\top \left(C^{(1)}\right)^\top p(\underline{W})C^{(2)}, \left(C^{(1)}\right)^\top p(\underline{W})A_Q C^{(2)}$ also takes $L^{1+o(1)}$ time.

**Approximation Error.** We have

$$\left\|\frac{\partial \mathcal{L}}{\partial \underline{A}_Q} - \widetilde{G}_Q^{(A)}\right\|_\infty$$

$$= \left\|\mathrm{vec}\left(B_Q^\top \left(C^{(1)}\right)^\top p(\underline{W})C^{(2)}\right) - \mathrm{vec}\left(B_Q^\top \left(C^{(1)}\right)^\top \widetilde{p}(\underline{W})C^{(2)}\right)\right\|_\infty \quad \text{(By Lemma 3.2)}$$

$$= \left\|\left(B_Q^\top \left(C^{(1)}\right)^\top p(\underline{W})C^{(2)}\right) - \left(B_Q^\top \left(C^{(1)}\right)^\top \widetilde{p}(\underline{W})C^{(2)}\right)\right\|_\infty$$

$$\text{(By definition, } \|A\|_\infty := \max_{i,j} |A_{ij}| \text{ for any matrix } A)$$

$$\leq \left\|\left(B_Q^\top \left(C^{(1)}\right)^\top (p_1(\underline{W}) - \widetilde{p}_1(\underline{W}))C^{(2)}\right)\right\|_\infty + \left\|\left(B_Q^\top \left(C^{(1)}\right)^\top (p_2(\underline{W}) - \widetilde{p}_2(\underline{W}))C^{(2)}\right)\right\|_\infty$$

$$\text{(By Definition 3.6 and triangle inequality)}$$

$$\leq \|B_Q\|_\infty \left\|C^{(1)}\right\|_\infty \left\|C^{(2)}\right\|_\infty \left(\|(p_1(\underline{W}) - \widetilde{p}_1(\underline{W}))\|_\infty + \|(p_2(\underline{W}) - \widetilde{p}_2(\underline{W}))\|_\infty\right)$$

$$\text{(By the sub-multiplicative property of } \infty\text{-norm)}$$

$$\leq \epsilon/\mathrm{poly}(L). \quad \text{(By Lemma 3.6 and Lemma 3.7)}$$

Similarly, it holds

$$
\left\| \frac{\partial \mathcal{L}}{\partial \underline{B}_Q} - \widetilde{G}_Q^{(B)} \right\|_\infty
$$

$$
= \left\| \mathrm{vec}\left( \left( C^{(1)} \right)^\top p(\underline{W}) A_Q C^{(2)} \right) - \mathrm{vec}\left( B_Q^\top \left( C^{(1)} \right)^\top \widetilde{p}(\underline{W}) A_Q C^{(2)} \right) \right\|_\infty
$$

$$
= \left\| \left( \left( C^{(1)} \right)^\top p(\underline{W}) A_Q C^{(2)} \right) - \left( \left( C^{(1)} \right)^\top \widetilde{p}(\underline{W}) A_Q C^{(2)} \right) \right\|_\infty
$$

$$
\leq \left\| \left( \left( C^{(1)} \right)^\top (p_1(\underline{W}) - \widetilde{p}_1(\underline{W})) A_Q C^{(2)} \right) \right\|_\infty + \left\| \left( \left( C^{(1)} \right)^\top (p_2(\underline{W}) - \widetilde{p}_2(\underline{W})) A_Q C^{(2)} \right) \right\|_\infty
$$

$$
\leq \|A_Q\|_\infty \left\| C^{(1)} \right\|_\infty \left\| C^{(2)} \right\|_\infty \left( \|(p_1(\underline{W}) - \widetilde{p}_1(\underline{W}))\|_\infty + \|(p_2(\underline{W}) - \widetilde{p}_2(\underline{W}))\|_\infty \right)
$$

$$
\leq \epsilon/\mathrm{poly}(L).
$$

Setting $\epsilon = 1/\mathrm{poly}(L)$, we complete the proof. $\qquad\square$

## D  PROOF OF THEOREM A.1

We prepare the proof with the following definitions and lemmas.

Similar to Section 3, we introduce the $u(\cdot), \alpha(\cdot), f(\cdot), c(\cdot)$ notations. Notably, we introduce them for both $K$ and $Q$ because there are two sets of adaptors: $B_K, A_K$ and $B_Q, A_Q$.

**Definition D.1** ($u(\cdot)$). Let $\mathsf{C}^K := C_K^{(1)} \otimes C_K^{(2)}$, and $\mathsf{C}^Q := C_Q^{(1)} \otimes C_Q^{(2)}$. Recall that $\mathsf{C}_{\underline{j}}^K, \mathsf{C}_{\underline{j}}^Q \in \mathbb{R}^{L \times d^2}$ are sub-block matrices of $\mathsf{C}^K, \mathsf{C}^Q$. For every $\underline{j} \in [L]$, we define two functions $u_K(\underline{W})_{\underline{j}}, u_Q(\underline{W})_{\underline{j}}$ : $\mathbb{R}^{d^2} \to \mathbb{R}^L$: $u_K(\underline{W})_{\underline{j}} := \exp\left(\mathsf{C}_{\underline{j}}^K \underline{W}\right) \in \mathbb{R}^L$ and $u_Q(\underline{W})_{\underline{j}} := \exp\left(\mathsf{C}_{\underline{j}}^Q \underline{W}\right) \in \mathbb{R}^L$.

**Definition D.2** ($\alpha(\cdot)$). Let $\mathsf{C}^K := C_K^{(1)} \otimes C_K^{(2)}$, and $\mathsf{C}^Q := C_Q^{(1)} \otimes C_Q^{(2)}$. Recall that $\mathsf{C}_{\underline{j}}^K, \mathsf{C}_{\underline{j}}^Q \in \mathbb{R}^{L \times d^2}$ are sub-block matrices of $\mathsf{C}^K, \mathsf{C}^Q$. For every index $\underline{j} \in [L]$, we define two functions $\alpha_Q(\underline{W})_{\underline{j}}, \alpha_K(\underline{W})_{\underline{j}}$ : $\mathbb{R}^{d^2} \to \mathbb{R}$: $\alpha_Q(\underline{W})_{\underline{j}} := \langle \exp\left(\mathsf{C}_{\underline{j}}^Q \underline{W}\right), \mathbb{1}_L \rangle \in \mathbb{R}$ and $\alpha_K(\underline{W})_{\underline{j}} := \langle \exp\left(\mathsf{C}_{\underline{j}}^K \underline{W}\right), \mathbb{1}_L \rangle \in \mathbb{R}$.

**Definition D.3** ($f(\cdot)$). Let $\alpha_Q(\underline{W})_{\underline{j}}, \alpha_K(\underline{W})_{\underline{j}} \in \mathbb{R}$ follow Definition D.2, and $u_K(\underline{W})_{\underline{j}}, u_Q(\underline{W})_{\underline{j}} \in \mathbb{R}^L$ follow Definition D.1. For any $\underline{j} \in [L]$, we define two functions $f_Q(\underline{W})_{\underline{j}}, f_K(\underline{W})_{\underline{j}} : \mathbb{R}^{d^2} \to \mathbb{R}^L$ as

$$f_Q(\underline{W})_{\underline{j}} := \underbrace{\alpha_Q(\underline{W})_{\underline{j}}^{-1}}_{\text{scalar}} \underbrace{u_Q(\underline{W})_{\underline{j}}}_{L \times 1}, \quad f_K(\underline{W})_{\underline{j}} := \underbrace{\alpha_K(\underline{W})_{\underline{j}}^{-1}}_{\text{scalar}} \underbrace{u_K(\underline{W})_{\underline{j}}}_{L \times 1},$$

such that $f_Q(\underline{W}), f_K(\underline{W}) \in \mathbb{R}^{L \times L}$ denote the matrices whose $\underline{j}$-th rows are $f_Q(\underline{W})_{\underline{j}}^\top, f_K(\underline{W})_{\underline{j}}^\top$.

**Definition D.4** ($c(\cdot)$). For every $\underline{j} \in [L]$, let $f_Q(\underline{W})_{\underline{j}}, f_K(\underline{W})_{\underline{j}} : \mathbb{R}^{d^2} \to \mathbb{R}^L$ follow Definition D.3. For every $i \in [d]$, let $C^{(3)}[\cdot, i] \in \mathbb{R}^L$ follow (A.1). For each $\underline{j} \in [L]$ and $i \in [d]$, we define two functions $c_Q(\underline{W})_{\underline{j},i}, c_K(\underline{W})_{\underline{j},i} : \mathbb{R}^{d^2} \times \mathbb{R}^{d^2} \to \mathbb{R}$ as

$$c_Q(\underline{W})_{\underline{j},i} := \langle f_Q(\underline{W})_{\underline{j}}, C^{(3)}[\cdot, i] \rangle - Y_{\underline{j},i}, \quad c_K(\underline{W})_{\underline{j},i} := \langle f_K(\underline{W})_{\underline{j}}, C^{(3)}[\cdot, i] \rangle - Y_{\underline{j},i}.$$

Here $Y_{\underline{j},i}$ is the $(\underline{j}, i)$-th coordinate/location of $Y \in \mathbb{R}^{L \times d}$ for $\underline{j} \in [L], i \in [d]$.

These give

$$\underbrace{c_Q(\underline{W})}_{L \times d} = \underbrace{f_Q(\underline{W})}_{L \times L} \underbrace{C^{(3)}}_{L \times d} - \underbrace{Y}_{L \times d}, \quad \text{and} \quad \underbrace{c_K(\underline{W})}_{L \times d} = \underbrace{f_K(\underline{W})}_{L \times L} \underbrace{C^{(3)}}_{L \times d} - \underbrace{Y}_{L \times d}.$$

**Definition D.5.** For every $\underline{j} \in [L]$ and every $i \in [d]$, let $\mathcal{L}_Q(\underline{W})_{\underline{j},i} := c_Q(\underline{W})_{\underline{j},i}^2/2$, and $\mathcal{L}_K(\underline{W})_{\underline{j},i} := c_K(\underline{W})_{\underline{j},i}^2/2$.

Let matrix $W_Q = W_Q^\star + B_Q A_Q \cdot W_K = W_K^\star + B_K A_K$ and loss function $\mathcal{L}$ be (A.2). From above definitions, it holds $\mathcal{L}(A_K, B_K, A_Q, B_Q) = \mathcal{L}(\underline{W}_Q, \underline{W}_K)$ and the adaptation gradients of $\mathcal{L}$ (A.2) become

$$\frac{\partial \mathcal{L}\left(\underline{W}_Q, \underline{W}_K\right)}{\partial \underline{W}_Q} = \frac{\partial}{\partial \underline{W}_Q} \sum_{\underline{j}}^{L} \sum_{i=1}^{d} \mathcal{L}_Q(\underline{W}_Q)_{\underline{j},i} = \frac{\partial}{\partial \underline{W}_Q} \frac{1}{2} \sum_{\underline{j}}^{L} \sum_{i=1}^{d} c_Q(\underline{W}_Q)_{\underline{j},i}^2, \tag{D.1}$$

and

$$\frac{\partial \mathcal{L}\left(\underline{W}_Q, \underline{W}_K\right)}{\partial \underline{W}_K^\top} = \frac{\partial}{\partial \underline{W}_K^\top} \sum_{\underline{j}}^{L} \sum_{i=1}^{d} \mathcal{L}_K(\underline{W}_K^\top)_{\underline{j},i} = \frac{\partial}{\partial \underline{W}_K^\top} \frac{1}{2} \sum_{\underline{j}}^{L} \sum_{i=1}^{d} c_K(\underline{W}_K^\top)_{\underline{j},i}^2. \tag{D.2}$$

(D.1) and (D.2) present a decomposition of the gradients of LoRA loss $\mathcal{L}$ (A.2) aspect to $\underline{W}_Q$ and $\underline{W}_K^\top$ into $L \cdot d$ terms, each simple enough for tracking gradient computation.

Now, we are ready to compute the gradients of the LoRA loss aspect to $\underline{W}_Q$ and $\underline{W}_K^\top$ as follows.

**Lemma D.1** (Low-Rank Decomposition of LoRA Gradients). Let $\mathsf{C}_K := C_K^{(1)} \otimes C_K^{(2)}, \mathsf{C}_Q := C_Q^{(1)} \otimes C_Q^{(2)}$. Let fine-tuning weights be $W_Q = W_Q^\star + B_Q A_Q$ and $W_K = W_K^\star + B_K A_K$, and the loss function $\mathcal{L}$ follow Definition D.5. It holds

$$\frac{\partial \mathcal{L}(W_Q, W_K)}{\partial \underline{W}_Q} = \sum_{j=1}^{L} \sum_{i=1}^{d} c_Q (W_Q)_{j,i} \left( \mathsf{C}_{\underline{j}}^Q \right)^\top \left( \mathrm{diag} \left( f_Q (W_Q)_{\underline{j}} \right) - f_Q (W_Q)_{\underline{j}} f_Q (W_Q)_{\underline{j}}^\top \right) C^{(3)}[\cdot, i],$$

$$\frac{\partial \mathcal{L}(W_Q, W_K)}{\partial \underline{W}_K^\top} = \sum_{j=1}^{L} \sum_{i=1}^{d} c_K \left( W_K^\top \right)_{\underline{j},i} \left( \mathsf{C}_{\underline{j}}^K \right)^\top \left( \mathrm{diag} \left( f_K \left( W_K^\top \right)_{\underline{j}} \right) - f_K \left( W_K^\top \right)_{\underline{j}} f_K \left( W_K^\top \right)_{\underline{j}}^\top \right) C^{(3)}[\cdot, i].$$

*Proof.* This lemma is a generalization of Lemma 3.1. □

Next, we introduce the $q(\cdot)$ and $p(\cdot)$ notations. Again, there are two sets corresponding to the two sets of adaptors.

**Definition D.6.** Let $q_K(\underline{W}) := C^{(3)} (c_K(\underline{W}))^\top \in \mathbb{R}^{L \times L}, q_Q(\underline{W}) := C^{(3)} (c_Q(\underline{W}))^\top \in \mathbb{R}^{L \times L}$.

**Definition D.7.** For every index $\underline{j} \in [L]$, we define $p_Q(\underline{W})_{\underline{j}}, p_Q(\underline{W})_{\underline{j}} \in \mathbb{R}^L$ as

$$p_Q(\underline{W})_{\underline{j}} := \left( \mathrm{diag} \left( f_Q (\underline{W})_{\underline{j}} \right) - f_Q (\underline{W})_{\underline{j}} f_Q (\underline{W})_{\underline{j}}^\top \right) q_Q(\underline{W})_{\underline{j}},$$

$$p_K(\underline{W})_{\underline{j}} := \left( \mathrm{diag} \left( f_K (\underline{W})_{\underline{j}} \right) - f_K (\underline{W})_{\underline{j}} f_K (\underline{W})_{\underline{j}}^\top \right) q_K(\underline{W})_{\underline{j}}.$$

Lemma D.1 presents the Low-Rank Decomposition of LoRA Gradients. Before using the chain rule to compute the gradients of the loss $\mathcal{L}$ (A.2) with respect to $A_Q, A_K, B_Q, B_K$, we need to define a matrix $T$ to handle the transpose term $\underline{W}_K^\top$.

**Lemma D.2** (Sparse Matrix $T$). For any matrix $W \in \mathbb{R}^{m \times n}$, there exists a matrix $T(m,n) \in \mathbb{R}^{mn \times mn}$ such that $\underline{W}^\top = T(m,n)(\underline{W})$. The matrix $T(m,n)$ is sparse. Namely, for any $i \in [mn]$, there exist $1 \leq p \leq m$ and $1 \leq k \leq n$ such that $i = (p-1)n + k$. Then, for any $i, j \in [mn]$,

$$T(m,n)[i,j] := \begin{cases} 1, & \text{if } j = (k-1)m + p, \\ 0, & \text{otherwise.} \end{cases}$$

*Proof.* For any $1 \leq p \leq m$ and $1 \leq k \leq n$, consider the position of $W[p,k]$ in $\underline{W}$ and $\underline{W}^\top$.

In $\underline{W}, W[p,k] = \underline{W}[(k-1)m + p]$.

In $\underline{W}^\top, W[p,k] = \underline{W}^\top[(p-1)n + k]$.

Thus,

$$\underline{W}^\top[i] = T(m,n)[i,\cdot]\underline{W}$$
$$= T(m,n)[i,j] \cdot \underline{W}[j].$$

This completes the proof. □

Now, we are ready to compute the gradients of the LoRA loss $\mathcal{L}$ (A.2) with respect to $A_Q, A_K, B_Q, B_K$ using the chain rule as follows.

**Lemma D.3.** For any $a \in \mathbb{R}$, let $\operatorname{diag}_d(a) \in \mathbb{R}^{d \times d}$ be a $d \times d$ diagonal matrix with all entries equal to $a$. Recall $W_Q = W_Q^\star + B_Q A_Q$ and $W_K = W_K^\star + B_K A_K$. Let $J_{B_K}, J_{A_K} \in \mathbb{R}^{d^2 \times rd}$ be two matrices such that $\underline{W}_Q = \underline{W}_Q^\star + J_{B_Q}\underline{A}_Q$ and $\underline{W}_Q = \underline{W}_Q^\star + J_{A_Q}\underline{B}_Q$ via

$$
J_{B_K} = \begin{pmatrix} B_K & & & \\ & B_K & & \\ & & \ddots & \\ & & & B_K \end{pmatrix}, J_{A_Q} = \begin{pmatrix} \operatorname{diag}_d\left(A_K[1,1]\right) & \cdots & \operatorname{diag}_d\left(A_K[r,1]\right) \\ \operatorname{diag}_d\left(A_K[1,2]\right) & \cdots & \operatorname{diag}_d\left(A_K[r,2]\right) \\ \vdots & & \vdots \\ \operatorname{diag}_d\left(A_K[1,d]\right) & \cdots & \operatorname{diag}_d\left(A_K[r,d]\right) \end{pmatrix}.
$$

Let $J_{B_K}, J_{A_K}$ be two matrices such that $\underline{W}_K = \underline{W}_K^\star + J_{B_K}\underline{A}_K$ and $\underline{W}_K = W_K^\star + J_{A_K}\underline{B}_K$ via

$$
J_{B_Q} = \begin{pmatrix} B_Q & & & \\ & B_Q & & \\ & & \ddots & \\ & & & B_Q \end{pmatrix}, J_{A_Q} = \begin{pmatrix} \operatorname{diag}_d\left(A_Q[1,1]\right) & \cdots & \operatorname{diag}_d\left(A_Q[r,1]\right) \\ \operatorname{diag}_d\left(A_Q[1,2]\right) & \cdots & \operatorname{diag}_d\left(A_Q[r,2]\right) \\ \vdots & & \vdots \\ \operatorname{diag}_d\left(A_Q[1,d]\right) & \cdots & \operatorname{diag}_d\left(A_Q[r,d]\right) \end{pmatrix}.
$$

Then the derivatives of loss function $\mathcal{L}$ (A.2) respect to $\underline{A}_Q, \underline{B}_Q, \underline{A}_K, \underline{B}_K$ are

$$
\frac{\partial \mathcal{L}}{\partial \underline{A}_Q} = \sum_{j=1}^L \sum_{i=1}^d \left(J_{B_Q}\right)^\top c_Q \left(\underline{W}_Q\right)_{j,i} \left(\mathsf{C}_{\underline{j}}^Q\right)^\top \left(\operatorname{diag}\left(f_Q\left(\underline{W}_Q\right)_{\underline{j}}\right) - f_Q\left(\underline{W}_Q\right)_{\underline{j}} f_Q\left(\underline{W}_Q\right)_{\underline{j}}^\top\right) C^{(3)}[\cdot, i],
$$

$$
\frac{\partial \mathcal{L}}{\partial \underline{B}_Q} = \sum_{j=1}^L \sum_{i=1}^d \left(J_{A_Q}\right)^\top c_Q \left(\underline{W}_Q\right)_{j,i} \left(\mathsf{C}_{\underline{j}}^Q\right)^\top \left(\operatorname{diag}\left(f_Q\left(\underline{W}_Q\right)_{\underline{j}}\right) - f_Q\left(\underline{W}_Q\right)_{\underline{j}} f_Q\left(\underline{W}_Q\right)_{\underline{j}}^\top\right) C^{(3)}[\cdot, i],
$$

$$
\frac{\partial \mathcal{L}}{\partial \underline{A}_K} = \sum_{j=1}^L \sum_{i=1}^d \left(T\left(d^2, d^2\right) J_{B_K}\right)^\top c_K \left(\underline{W}_K^\top\right)_{j,i} \left(\mathsf{C}_{\underline{j}}^K\right)^\top \left(\operatorname{diag}\left(f_K\left(\underline{W}_K^\top\right)_{\underline{j}}\right) - f_K\left(\underline{W}_K^\top\right)_{\underline{j}} f_K\left(\underline{W}_K^\top\right)_{\underline{j}}^\top\right) C^{(3)}[\cdot, i],
$$

$$
\frac{\partial \mathcal{L}}{\partial \underline{B}_K} = \sum_{j=1}^L \sum_{i=1}^d \left(T\left(d^2, d^2\right) J_{A_K}\right)^\top c_K \left(\underline{W}_K^\top\right)_{j,i} \left(\mathsf{C}_{\underline{j}}^K\right)^\top \left(\operatorname{diag}\left(f_K\left(\underline{W}_K^\top\right)_{\underline{j}}\right) - f_K\left(\underline{W}_K^\top\right)_{\underline{j}} f_K\left(\underline{W}_K^\top\right)_{\underline{j}}^\top\right) C^{(3)}[\cdot, i].
$$

*Proof.* $\frac{\partial \mathcal{L}}{\partial \underline{A}_Q}$ and $\frac{\partial \mathcal{L}}{\partial \underline{B}_Q}$ follow Lemma C.1 directly.

For $\frac{\partial \mathcal{L}}{\partial \underline{A}_K}$ and $\frac{\partial \mathcal{L}}{\partial \underline{B}_K}$, we have:

$$
\begin{aligned}
\underline{W}_K^\top &= T(d^2, d^2)\underline{W}_K \\
&= T(d^2, d^2)\left(\underline{W}_K^\star + J_{B_K}\underline{A}_K\right) \\
&= T(d^2, d^2)\left(\underline{W}_K^\star + J_{A_K}\underline{B}_K\right).
\end{aligned}
$$

Therefore,

$$
\begin{aligned}
\frac{\partial \mathcal{L}}{\partial \underline{A}_K} &= \frac{\partial \underline{W}_K^\top}{\partial \underline{A}_K} \frac{\partial \mathcal{L}(\underline{W}_Q, \underline{W}_K)}{\partial \underline{W}_K^\top} \\
&= T(d^2, d^2)J_{B_K} \frac{\partial \mathcal{L}(\underline{W}_Q, \underline{W}_K)}{\partial \underline{W}_K^\top}.
\end{aligned}
$$

Similarly,

$$
\begin{aligned}
\frac{\partial \mathcal{L}}{\partial \underline{B}_K} &= \frac{\partial \underline{W}_K^\top}{\partial \underline{B}_K} \frac{\partial \mathcal{L}(\underline{W}_Q, \underline{W}_K)}{\partial \underline{W}_K^\top} \\
&= T(d^2, d^2)J_{A_K} \frac{\partial \mathcal{L}(\underline{W}_Q, \underline{W}_K)}{\partial \underline{W}_K^\top}.
\end{aligned}
$$

Thus, we complete the proof by following the conclusions of Lemma D.1. $\qquad \square$

Next, we simplify the derivatives with $p(\cdot)$ notation.

**Lemma D.4.** Let $q_Q, q_K \in \mathbb{R}^{L \times L}$ as defined in Definition D.6. Let $p_Q, p_K$ as defined in Definition D.7. Then it holds

$$\frac{\partial \mathcal{L}}{\partial \underline{A}_Q} = \mathrm{vec}\left( B_Q^\top \left( C_Q^{(1)} \right)^\top p_Q(\underline{W}_Q) C_Q^{(2)} \right),$$

$$\frac{\partial \mathcal{L}}{\partial \underline{B}_Q} = \mathrm{vec}\left( \left( C_Q^{(1)} \right)^\top p_Q(\underline{W}_Q) A_Q C_Q^{(2)} \right),$$

$$\frac{\partial \mathcal{L}}{\partial \underline{A}_K} = T\left(d^2, d^2\right)^\top \mathrm{vec}\left( B_K^\top \left( C_K^{(1)} \right)^\top p_K\left( \underline{W}_K^\top \right) C_K^{(2)} \right),$$

$$\frac{\partial \mathcal{L}}{\partial \underline{B}_K} = T\left(d^2, d^2\right)^\top \mathrm{vec}\left( \left( C_K^{(1)} \right)^\top p_K\left( \underline{W}_K^\top \right) A_K C_K^{(2)} \right).$$

*Proof.* For $\frac{\partial \mathcal{L}}{\partial \underline{A}_Q}$ and $\frac{\partial \mathcal{L}}{\partial \underline{B}_Q}$, we follow the proof of Theorem 3.1.

For $\frac{\partial \mathcal{L}}{\partial \underline{A}_K}$, we have

$$\frac{\partial \mathcal{L}}{\partial \underline{A}_K}$$

$$= \sum_{\underline{j}=1}^{L} \sum_{i=1}^{d} \left( T\left(d^2, d^2\right) J_{B_K} \right)^\top c_K \left( \underline{W}_K^\top \right)_{\underline{j}, i} \left( C_{\underline{j}}^K \right)^\top \left( \mathrm{diag}\left( f_K\left( \underline{W}_K^\top \right)_{\underline{j}} \right) - f_K\left( \underline{W}_K^\top \right)_{\underline{j}} f_K\left( \underline{W}_K^\top \right)_{\underline{j}}^\top \right) C^{(3)}[\cdot, i]$$

(By Lemma D.3)

$$= \sum_{\underline{j}=1}^{L} \left( T\left(d^2, d^2\right) J_{B_K} \right)^\top \left( C_{\underline{j}}^K \right)^\top \left( \mathrm{diag}\left( f_K\left( \underline{W}_K^\top \right)_{\underline{j}} \right) - f_K\left( \underline{W}_K^\top \right)_{\underline{j}} f_K\left( \underline{W}_K^\top \right)_{\underline{j}}^\top \right) q_K\left( \underline{W}_K^\top \right)_{\underline{j}}$$

(By Definition D.6)

$$= T\left(d^2, d^2\right)^\top \sum_{\underline{j}=1}^{L} J_{B_K}^\top \left( C_{\underline{j}}^K \right)^\top p_K\left( \underline{W}_K^\top \right)_{\underline{j}}$$

(By Definition D.7)

$$= T\left(d^2, d^2\right)^\top \mathrm{vec}\left( B_K^\top \left( C_K^{(1)} \right)^\top p_K\left( \underline{W}_K^\top \right) C_K^{(2)} \right).$$

(By Lemma 2.1)

Similarly, for $\frac{\partial \mathcal{L}}{\partial \underline{B}_K}$, it holds

$$\frac{\partial \mathcal{L}}{\partial \underline{B}_K}$$

$$= \sum_{\underline{j}=1}^{L} \sum_{i=1}^{d} \left( T(d^2, d^2) J_{A_K} \right)^\top c_K \left( \underline{W}_K^\top \right)_{\underline{j}, i} \left( C_{\underline{j}}^K \right)^\top \left( \mathrm{diag}\left( f_K\left( \underline{W}_K^\top \right)_{\underline{j}} \right) - f_K\left( \underline{W}_K^\top \right)_{\underline{j}} f_K\left( \underline{W}_K^\top \right)_{\underline{j}}^\top \right) C^{(3)}[\cdot, i]$$

$$= \sum_{\underline{j}=1}^{L} \left( T\left(d^2, d^2\right) J_{A_K} \right)^\top \left( C_{\underline{j}}^K \right)^\top \left( \mathrm{diag}\left( f_K\left( \underline{W}_K^\top \right)_{\underline{j}} \right) - f_K\left( \underline{W}_K^\top \right)_{\underline{j}} f_K\left( \underline{W}_K^\top \right)_{\underline{j}}^\top \right) q_K\left( \underline{W}_K^\top \right)_{\underline{j}}$$

$$= T\left(d^2, d^2\right)^\top \sum_{\underline{j}=1}^{L} J_{A_K}^\top \left( C_{\underline{j}}^K \right)^\top q_K\left( \underline{W}_K^\top \right)_{\underline{j}}$$

$$= T\left(d^2, d^2\right)^\top \mathrm{vec}\left( \left( C_K^{(1)} \right)^\top p_K\left( \underline{W}_K^\top \right) A_K C_K^{(2)} \right).$$

This completes the proof. $\square$

Similarly, Lemma D.4 states that the chain rule terms for characterizing Problem 3 are tied to $p_Q(\cdot)$ and $p_K Q(\cdot)$. Therefore, to characterize $\widetilde{G}_Q^{(A)}, \widetilde{G}_Q^{(B)}, \widetilde{G}_K^{(A)},$ and $\widetilde{G}_K^{(B)}$ (i.e., the approximations of

$G_Q^{(A)}$, $G_Q^{(B)}$, $G_K^{(A)}$, and $G_K^{(B)}$), for $\mu = Q, K$, we need to approximate the functions $f_\mu(\cdot)$, $q_\mu(\cdot)$, $c_\mu(\cdot)$, and thus $p_\mu(\cdot)$ with precision guarantees. To do so, it is convenient to consider the following decomposition of $p_\mu(\cdot)$ for $\mu = Q, K$.

**Definition D.8.** For every index $j \in [L]$, we define $p_1^K(\underline{W})_j, p_2^K(\underline{W})_j \in \mathbb{R}^L$ as

$$p_1^Q(\underline{W})_j := \mathrm{diag}\left(f_Q\left(\underline{W}\right)_j\right) q_Q(\underline{W})_j, \quad p_2^Q(\underline{W})_j := f_Q\left(\underline{W}\right)_j f_Q\left(\underline{W}\right)_j^\top q_Q(\underline{W})_j,$$

$$p_1^K(\underline{W})_j := \mathrm{diag}\left(f_K\left(\underline{W}\right)_j\right) q_K(\underline{W})_j, \quad p_2^K(\underline{W})_j := f_K\left(\underline{W}\right)_j f_K\left(\underline{W}\right)_j^\top q_K(\underline{W})_j.$$

such that $p_Q(\underline{W}) = p_1^Q(\underline{W}) - p_2^Q(\underline{W}), p_Q(\underline{W}) = p_1^Q(\underline{W}) - p_2^Q(\underline{W})$.

**Overview of Our Proof Strategy.** Similar to Section 3, we adopt the following strategy: term-by-term approximation for precision-guaranteed, almost linear time algorithms to compute LoRA gradients in Problem 3. For all $\mu = Q, K$, we do the following.

**Step 1.** Prove the existence of almost linear approximation algorithms for $f_\mu(\cdot)$, $q_\mu(\cdot)$, and $c_\mu(\cdot)$ via low-rank approximation (Lemma D.5, Lemma D.7, and Lemma D.6).

**Step 2.** Prove the existence of almost linear approximation algorithms for $p_1^\mu(\cdot)$, $p_2^\mu(\cdot)$, and thus $p_\mu(\cdot)$ via the low-rank-preserving property of the multiplication between $f_\mu(\cdot)$ and $q_\mu(\cdot)$ (Lemma D.8 and Lemma D.9).

**Step 3.** Prove the existence of almost linear approximation algorithms for the LoRA adapter gradients (i.e., $\frac{\partial \mathcal{L}}{\partial \underline{A}_Q}$, $\frac{\partial \mathcal{L}}{\partial \underline{A}_K}$, $\frac{\partial \mathcal{L}}{\partial \underline{B}_Q}$, and $\frac{\partial \mathcal{L}}{\partial \underline{B}_K}$ in Lemma D.4) using the results from **Step 1** and **Step 2** (Theorem A.1).

**Step 1.** We start with low-rank approximations for $f_\mu(\cdot), q_\mu(\cdot), c_\mu(\cdot)$.

**Lemma D.5** (Approximate $f_Q(\cdot), f_K(\cdot)$). Let $\Gamma = o(\sqrt{\log L})$, for $\mu = Q, K$, suppose $C_\mu^{(1)}, C_\mu^{(2)} \in \mathbb{R}^{L \times d}$, $W \in \mathbb{R}^{d \times d}$, and $f_\mu(\underline{W}) = D^{-1} \exp\left(C_\mu^{(1)} W \left(C_\mu^{(2)}\right)^\top\right)$ with $D$ following (A.2). There exists a $k_1 = L^{o(1)}$ such that if $\left\|C_\mu^{(1)} W\right\|_\infty \leq \Gamma$ and $\left\|C_\mu^{(2)}\right\|_\infty \leq \Gamma$, then there exist four matrices $U_1^Q, V_1^Q, U_1^K, V_1^K \in \mathbb{R}^{L \times k_1}$ such that

$$\left\|U_1^Q(V_1^Q)^\top - f_Q(\underline{W})\right\|_\infty \leq \epsilon/\mathrm{poly}(L),$$
$$\left\|U_1^K(V_1^K)^\top - f_K(\underline{W})\right\|_\infty \leq \epsilon/\mathrm{poly}(L).$$

In addition, it takes $L^{1+o(1)}$ time to construct $U_1^Q, V_1^Q, U_1^K, V_1^K$.

*Proof.* This follows the proof of Lemma 3.3 □

**Lemma D.6** (Approximate $c_Q(\cdot), c_K(\cdot)$). Assume all numerical values are in $O(\log L)$ bits. Let $d = O(\log L)$ and $c_Q(\underline{W}), c_K(\underline{W}) \in \mathbb{R}^{L \times d}$ follows Definition D.4. Then there exist four matrices $U_1^Q, V_1^Q, U_1^K, V_1^K \in \mathbb{R}^{L \times k_1}$ such that

$$\left\|U_1^Q(V_1^Q)^\top C^{(3)} - Y - c_Q(\underline{W})\right\|_\infty \leq \epsilon/\mathrm{poly}(L),$$
$$\left\|U_1^K(V_1^K)^\top C^{(3)} - Y - c_K(\underline{W})\right\|_\infty \leq \epsilon/\mathrm{poly}(L).$$

*Proof.* This follows the proof of Lemma 3.4 □

**Lemma D.7** (Approximate $q_Q(\cdot), q_K(\cdot)$). Let $k_2 = L^{o(1)}$, $c_Q(W), c_K(W) \in \mathbb{R}^{L \times d}$ follows Definition D.4 and let $q_K(\underline{W}) := C^{(3)} (c_K(\underline{W}))^\top \in \mathbb{R}^{L \times L}$, $q_Q(\underline{W}) := C^{(3)} (c_Q(\underline{W}))^\top \in \mathbb{R}^{L \times L}$.

(follows Definition D.6). Then there exist four matrices $U_2^Q, V_2^Q, U_2^K, V_2^K \in \mathbb{R}^{L \times k_2}$ such that

$$\left\| U_2^Q (V_2^Q)^\top - q_Q(\underline{W}) \right\|_\infty \le \epsilon/\text{poly}(L),$$

$$\left\| U_2^K (V_2^K)^\top - q_K(\underline{W}) \right\|_\infty \le \epsilon/\text{poly}(L).$$

In addition, it takes $L^{1+o(1)}$ time to construct $U_2^Q, V_2^Q, U_2^K, V_2^K$.

*Proof.* This follows the proof of Lemma 3.5 □

**Step 2.** Now, we use above lemmas to construct low-rank approximations for $p_1^\mu(\cdot), p_2^\mu(\cdot), p_\mu(\cdot)$.

**Lemma D.8** (Approximate $p_1^Q(\cdot), p_1^K(\cdot)$). Let $k_1, k_2, k_3 = L^{o(1)}$. For $\mu = K, Q$, suppose $U_1^\mu, V_1^\mu \in \mathbb{R}^{L \times k_1}$ approximate $f_\mu(\underline{W}) \in \mathbb{R}^{L \times L}$ such that $\left\| U_1^\mu (V_1^\mu)^\top - f_\mu(\underline{W}) \right\|_\infty \le \epsilon/\text{poly}(L)$, and $U_2^\mu, V_2^\mu \in \mathbb{R}^{L \times k_2}$ approximate the $q_\mu(\underline{W}) \in \mathbb{R}^{L \times L}$ such that $\left\| U_2^\mu (V_2^\mu)^\top - q_\mu(\underline{W}) \right\|_\infty \le \epsilon/\text{poly}(L)$. Then there exist two matrices $U_3^\mu, V_3^\mu \in \mathbb{R}^{L \times k_3}$ such that

$$\left\| U_3^\mu (V_3^\mu)^\top - p_1^\mu(\underline{W}) \right\|_\infty \le \epsilon/\text{poly}(L), \quad \text{for } \mu = K, Q.$$

In addition, it takes $L^{1+o(1)}$ time to construct $U_3^Q, V_3^Q, U_3^K, V_3^K$.

*Proof.* This follows the proof of Lemma 3.6 □

**Lemma D.9** (Approximate $p_2^Q(\cdot), p_2^K(\cdot)$). Let $k_1, k_2, k_4 = L^{o(1)}$. Let $p_2^Q(\underline{W}), p_2^K(\underline{W}) \in \mathbb{R}^{L \times L}$ such that its $j$-th column is $p_2(\underline{W})_j = f(\underline{W})_j f(\underline{W})_j^\top q(\underline{W})_j$ follow Definition D.8, for each $j \in [L]$. For $\mu = K, Q$, suppose $U_1^\mu, V_1^\mu \in \mathbb{R}^{L \times k_1}$ approximates the $f_\mu(\underline{W})$ such that $\left\| U_1^\mu (V_1^\mu)^\top - f_\mu(\underline{W}) \right\|_\infty \le \epsilon/\text{poly}(L)$, and $U_2^\mu, V_2^\mu \in \mathbb{R}^{L \times k_2}$ approximates the $q_\mu(\underline{W}) \in \mathbb{R}^{L \times L}$ such that $\left\| U_2^\mu (V_2^\mu)^\top - q_\mu(\underline{W}) \right\|_\infty \le \epsilon/\text{poly}(L)$. Then there exist matrices $U_4^\mu, V_4^\mu \in \mathbb{R}^{L \times k_4}$ such that

$$\left\| U_4^\mu (V_4^\mu)^\top - p_2^\mu(\underline{W}) \right\|_\infty \le \epsilon/\text{poly}(L), \quad \text{for } \mu = K, Q.$$

In addition, it takes $L^{1+o(1)}$ time to construct $U_4^Q, V_4^Q, U_4^K, V_4^K$.

*Proof.* This follows the proof of Lemma 3.7 □

**Step 3.** Combining above, we arrive our main result: almost linear algorithm for Problem 3.

**Theorem D.1** (Main Result: Existence of almost Linear Time ALoRAGC). Let $\Gamma = o(\sqrt{\log L})$. Suppose all numerical values are in $O(\log L)$-bits encoding. Then there exists a $L^{1+o(1)}$ time algorithm to solve ALoRAGC $(L, d = O(\log L), r = L^{o(1)}, \epsilon = 1/\text{poly}(L)$ (i.e Problem 3) up to $1/\text{poly}(L)$ accuracy. In particular, this algorithm outputs gradient matrices $\{\widetilde{G}_\mu^{(A)} \in \mathbb{R}^{d \times r}, \widetilde{G}_\mu^{(B)} \in \mathbb{R}^{r \times d}\}_{\mu=K,Q}$ such that

$$\max \left( \left\| \frac{\partial \mathcal{L}}{\partial \underline{B}_\mu} - \widetilde{G}_\mu^{(B)} \right\|_\infty, \left\| \frac{\partial \mathcal{L}}{\partial \underline{A}_\mu} - \widetilde{G}_\mu^{(A)} \right\|_\infty \right) \le 1/\text{poly}(L), \quad \text{for } \mu = K, Q.$$

*Proof of Theorem A.1.* By the definitions of matrices $p_1^K(\underline{W}), p_1^Q(\underline{W}), p_2^K(\underline{W}), p_2^Q(\underline{W})$ in Definition D.8 and $p_K(\underline{W}), p_Q(\underline{W})$ in Definition D.7. It is straightforward that

$$p_K(\underline{W}) = p_1^K(\underline{W}) - p_2^K(\underline{W}), \quad \text{and} \quad p_Q(\underline{W}) = p_1^Q(\underline{W}) - p_2^Q(\underline{W}).$$

According to Lemma D.4, we have

$$\frac{\partial \mathcal{L}}{\partial \underline{A}_Q} = \text{vec}\left(B_Q^\top \left(C_Q^{(1)}\right)^\top p_Q\left(\underline{W}_Q\right) C_Q^{(2)}\right)$$

$$\frac{\partial \mathcal{L}}{\partial \underline{B}_Q} = \text{vec}\left(\left(C_Q^{(1)}\right)^\top p_Q\left(\underline{W}_Q\right) A_Q C_Q^{(2)}\right)$$

$$\frac{\partial \mathcal{L}}{\partial \underline{A}_K} = T\left(d^2, d^2\right)^\top \text{vec}\left(B_K^\top \left(C_K^{(1)}\right)^\top p_K\left(\underline{W}_K^\top\right) C_K^{(2)}\right)$$

$$\frac{\partial \mathcal{L}}{\partial \underline{B}_K} = T\left(d^2, d^2\right)^\top \text{vec}\left(\left(C_K^{(1)}\right)^\top p_K\left(\underline{W}_K^\top\right) A_K C_K^{(2)}\right).$$

Next, we compute the time complexity of approximating these gradients to $1/\text{poly}(L)$ precision.

For $\frac{\partial \mathcal{L}}{\partial \underline{A}_Q}$ and $\frac{\partial \mathcal{L}}{\partial \underline{B}_Q}$, we follow the proof of Theorem 3.1. Specifically, it takes $L^{1+o(1)}$ time to approximate these gradients to $1/\text{poly}(L)$ precision.

For $\frac{\partial \mathcal{L}}{\partial \underline{A}_K}$ and $\frac{\partial \mathcal{L}}{\partial \underline{B}_K}$, we first note that $\left(T\left(d^2, d^2\right)\right)^\top$ is a constant matrix. In addition, due to Theorem 3.1, $\text{vec}\left(B_K^\top \left(C_K^{(1)}\right)^\top p_K\left(\underline{W}_K^\top\right) C_K^{(2)}\right)$ and $\text{vec}\left(\left(C_K^{(1)}\right)^\top p_K\left(\underline{W}_K^\top\right) A_K C_K^{(2)}\right)$, which are similar to $\frac{\partial \mathcal{L}}{\partial \underline{A}_Q}$ and $\frac{\partial \mathcal{L}}{\partial \underline{B}_Q}$, take $L^{1+o(1)}$ time to approximate to $1/\text{poly}(L)$ precision.

Therefore, to show the existence of $L^{1+o(1)}$ algorithms for Problem 3, we prove exact computation for $T\left(d^2, d^2\right)^\top \text{vec}\left(B_K^\top \left(C_K^{(1)}\right)^\top p_K\left(\underline{W}_K^\top\right) C_K^{(2)}\right)$ and $T\left(d^2, d^2\right)^\top \text{vec}\left(\left(C_K^{(1)}\right)^\top p_K\left(\underline{W}_K^\top\right) A_K C_K^{(2)}\right)$ takes $o(L^{1+o(1)})$ time as follows.

**Exact Computation for** $T\left(d^2, d^2\right)^\top \text{vec}\left(B_K^\top \left(C_K^{(1)}\right)^\top p_K\left(\underline{W}_K^\top\right) C_K^{(2)}\right)$. Recall from Lemma D.2 that $T\left(d^2, d^2\right)^\top$ is a sparse matrix with only one non-zero entry in each row. Thus, for each row, the exact computation takes $O(1)$ time. Therefore, the total time is $O(d^2)$. Given that $d = o(\log L)$, the overall time is still $L^{1+o(1)}$.

**Exact Computation for** $T\left(d^2, d^2\right)^\top \text{vec}\left(\left(C_K^{(1)}\right)^\top p_K\left(\underline{W}_K^\top\right) A_K C_K^{(2)}\right)$. Similarly, computing $T\left(d^2, d^2\right)^\top \text{vec}\left(\left(C_K^{(1)}\right)^\top p_K\left(\underline{W}_K^\top\right) A_K C_K^{(2)}\right)$ takes $O(d^2)$ time. Therefore, the total time is $O(d^2)$. Given that $d = o(\log L)$, the overall time is still $L^{1+o(1)}$.

**Approximation Error.** For $\frac{\partial \mathcal{L}}{\partial \underline{A}_Q}$ and $\frac{\partial \mathcal{L}}{\partial \underline{B}_Q}$, we follow the proof of Theorem 3.1. For $\frac{\partial \mathcal{L}}{\partial \underline{A}_K}$,

$$\left\| \frac{\partial \mathcal{L}}{\partial \underline{A}_K} - \widetilde{G}_K^{(A)} \right\|_\infty$$

$$= \left\| T\left(d^2, d^2\right)^\top \text{vec}\left(B_K^\top \left(C_K^{(1)}\right)^\top p_K\left(\underline{W}_K^\top\right) C_K^{(2)}\right) - T\left(d^2, d^2\right)^\top \text{vec}\left(B_K^\top \left(C_K^{(1)}\right)^\top \widetilde{p}_K\left(\underline{W}_K^\top\right) C_K^{(2)}\right) \right\|_\infty$$

$$\leq \left\| T\left(d^2, d^2\right)^\top \right\|_\infty \left\| \left(B_K^\top \left(C_K^{(1)}\right)^\top p_K\left(\underline{W}_K^\top\right) C_K^{(2)}\right) - \left(B_K^\top \left(C_K^{(1)}\right)^\top \widetilde{p}_K\left(\underline{W}_K^\top\right) C_K^{(2)}\right) \right\|_\infty$$

$$\leq \left\| \left(B_K^\top \left(C_K^{(1)}\right)^\top \left(p_1^K\left(\underline{W}_K^\top\right) - \widetilde{p}_1^K\left(\underline{W}_K^\top\right)\right) C_K^{(2)}\right) \right\|_\infty + \left\| \left(B_K^\top \left(C_K^{(1)}\right)^\top \left(p_2^K\left(\underline{W}_K^\top\right) - \widetilde{p}_2^K\left(\underline{W}_K^\top\right)\right) C_K^{(2)}\right) \right\|_\infty$$

$$\leq \|B_K\|_\infty \left\|C_K^{(1)}\right\|_\infty \left\|C_K^{(2)}\right\|_\infty \left(\left\| \left(p_1^K\left(\underline{W}_K^\top\right) - \widetilde{p}_1^K\left(\underline{W}_K^\top\right)\right) \right\|_\infty + \left\| \left(p_2^K\left(\underline{W}_K^\top\right) - \widetilde{p}_2^K\left(\underline{W}_K^\top\right)\right) \right\|_\infty\right)$$

$$\leq \epsilon/\text{poly}(L),$$

where the first step follows from Lemma D.3, the second step follows from the definition $\|A\|_\infty := \max_{i,j} |A_{ij}|$ for any matrix $A$, the third step follows from Definition D.8 and the triangle inequality,

the fourth step follows from the sub-multiplicative property of the $\infty$-norm, and the last step follows from Lemma D.8 and Lemma D.9.

Similarly, for $\frac{\partial \mathcal{L}}{\partial \underline{B}_K}$, it holds

$$
\left\| \frac{\partial \mathcal{L}}{\partial \underline{B}_K} - \widetilde{G}_K^{(B)} \right\|_\infty
$$

$$
= \left\| T\left(d^2, d^2\right)^\top \mathrm{vec}\left(\left(C_K^{(1)}\right)^\top p_K\left(\underline{W}_K^\top\right) A_K C_K^{(2)}\right) - T\left(d^2, d^2\right)^\top \mathrm{vec}\left(\left(C_K^{(1)}\right)^\top \widetilde{p}_K\left(\underline{W}_K^\top\right) A_K C_K^{(2)}\right) \right\|_\infty
$$

$$
\leq \left\| \left(T\left(d^2, d^2\right)\right)^\top \right\|_\infty \left\| \left(\left(C_K^{(1)}\right)^\top p_K\left(\underline{W}_K^\top\right) A_K C_K^{(2)}\right) - \left(\left(C_K^{(1)}\right)^\top \widetilde{p}_K\left(\underline{W}_K^\top\right) A_K C_K^{(2)}\right) \right\|_\infty
$$

$$
\leq \left\| \left(\left(C_K^{(1)}\right)^\top \left(p_1^K\left(\underline{W}_K^\top\right) - \widetilde{p}_1^K\left(\underline{W}_K^\top\right)\right) A_K C_K^{(2)}\right) \right\|_\infty + \left\| \left(\left(C_K^{(1)}\right)^\top \left(p_2^K\left(\underline{W}_K^\top\right) - \widetilde{p}_2^K\left(\underline{W}_K^\top\right)\right) A_K C_K^{(2)}\right) \right\|_\infty
$$

$$
\leq \|A_K\|_\infty \left\| C_K^{(1)} \right\|_\infty \left\| C_K^{(2)} \right\|_\infty \left( \left\| \left(p_1^K\left(\underline{W}_K^\top\right) - \widetilde{p}_1^K\left(\underline{W}_K^\top\right)\right) \right\|_\infty + \left\| \left(p_2^K\left(\underline{W}_K^\top\right) - \widetilde{p}_2^K\left(\underline{W}_K^\top\right)\right) \right\|_\infty \right)
$$

$$
\leq \epsilon/\mathrm{poly}(L).
$$

Setting $\epsilon = 1/\mathrm{poly}(L)$, we complete the proof. $\qquad\square$

# E   PROOF OF THEOREM 4.1

We recall our definition of ALoRAGC$(L, d, r, \epsilon)$ for special case from Problem 2 subject to LoRA loss (3.3). We aim to make the reduction from AAttLGC$(L, r, \epsilon)$ (Alman and Song, 2024a, Definition 1.4) to our problem ALoRAGC$(L, d, r, \epsilon)$.

---

**Definition E.1** (Approximate Attention Loss Gradient Computation (AAttLGC$(L, r, \epsilon)$), Definition 1.4 of (Alman and Song, 2024a)). Given four $L \times r$ size matrices $A_1 \in \mathbb{R}^{L \times r}, A_2 \in \mathbb{R}^{L \times r}, A_3 \in \mathbb{R}^{L \times r}, E \in \mathbb{R}^{L \times r}$ and a square matrix $X \in \mathbb{R}^{r \times r}$ to be fixed matrices. Assume that $\|A_1 X\|_\infty \leq B, \|A_2\|_\infty \leq B$. Assume all numerical values are in $\log(L)$-bits encoding. Let $\mathcal{L}(X) := \frac{1}{2}\|D^{-1}\exp(A_1 X A_2^\top/r)A_3 - E\|_F^2$. which $D := \mathrm{diag}(\exp(A_1 X A_2^\top/r)\mathbb{1}_L)$. Let $\frac{\mathrm{d}\mathcal{L}(X)}{\mathrm{d}X}$ denote the gradient of loss function $\mathcal{L}$. The goal is to output a matrix $\widetilde{g} \in \mathbb{R}^{L \times L}$ such that

$$\|\widetilde{g} - \frac{\mathrm{d}\mathcal{L}(X)}{\mathrm{d}X}\|_\infty \leq \epsilon.$$

---

We recall the main hardness result of (Alman and Song, 2024a) which shows a lower bound of AAttLGC$(L, r, \epsilon)$ (Definition E.1) in the following particular case by assuming SETH.

---

**Lemma E.1** (Theorem 5.5 of (Alman and Song, 2024a)). Let $\kappa : \mathbb{N} \to \mathbb{N}$ by any function with $\kappa(L) = \omega(1)$ and $\kappa(L) = o(\log L)$. Assuming SETH, there is no algorithm running in time $O(L^{2-\delta})$ for any constant $\delta > 0$ for Approximate Attention Loss Gradient Computation AAttLGC$(L, r, \epsilon)$, even in the case where $r = O(\log L)$ and the input matrices satisfy $\|A_1\|_\infty, \|A_2\|_\infty, \|A_3\|_\infty \leq O(\sqrt{\log L} \cdot \kappa(L)) = B, E = 0, X = \lambda I_r$ for some scalar $\lambda \in [0, 1]$, and $\varepsilon = O(1/(\log L)^4)$.

---

Finally, we are ready for our main proof of Theorem 4.1.

*Proof.* Considering Problem 2, we start with the following $O(1)$ reduction. Given the instance of AAttLGC$(L, r, \epsilon)$ and $A_1 \in \mathbb{R}^{L \times r}, A_2 \in \mathbb{R}^{L \times r}, A_3 \in \mathbb{R}^{L \times r}, E = 0, B = O(\sqrt{\log L} \cdot \kappa(L))$. We then transfer this instance to the instance of ALoRAGC$(L, d, r, \epsilon)$ by making the following substitution:

$$C^{(1)}B_Q = A_1, C^{(2)} = \{\underbrace{A_2}_{L \times r}, \underbrace{0}_{L \times (d-r)}\}/r, C^{(3)} = \{\underbrace{A_3}_{L \times r}, \underbrace{0}_{L \times (d-r)}\}, A_Q = \{\underbrace{X}_{r \times r}, \underbrace{0}_{r \times (d-r)}\}, \Gamma = B.$$

Then we have $\|C^{(2)}\|_\infty, \|C^{(1)}B_Q A_Q\|_\infty, \|Y\|_\infty \leq \Gamma$ such that

$$A_1 X A_2^T/r = C^{(1)}B_Q A_Q \left(C^{(2)}\right)^\mathsf{T},$$

and hence

$$\exp(A_1 X A_2^T)/r = \exp\left(C^{(1)}B_Q A_Q \left(C^{(2)}\right)^\mathsf{T}\right).$$

This implies that the upper $L \times r$ subblock is exactly the same. (Here we can assume $E = Y = 0$.)

$$(D^{-1}\exp\left\{C^{(1)}B_Q A_Q(C^{(2)})^\top\right\}C^{(3)} - Y)|_{L \times r} = (D^{-1}\exp(A_1 X A_2^\top/r)A_3 - E)|_{L \times r}.$$

This follows that the derivative with respect to $X$ of the RHS is the same as the partial derivative with respect to $A_Q$ by embedding $X$ into a subblock of $A_Q$. Now, by letting $\widetilde{G}_A = \widetilde{g}$ in the AAttLGCC$(L, r, \epsilon)$, which finishes the reduction. This completes the proof. $\square$

# F  QUADRATIC TIME COMPLEXITY OF EXACT LORA GRADIENT COMPUTATION

Here, we make more comments on tensor-trick decomposed LoRA loss from Lemma 3.1:

$$
\frac{\mathrm{d}\mathcal{L}(\underline{W})}{\mathrm{d}\underline{W}} = \sum_{\underline{j}=1}^{L}\sum_{i=1}^{d} c(\underline{W})_{\underline{j},i} \mathsf{C}_{\underline{j}}^{\top} \underbrace{\Big( \overbrace{\mathrm{diag}\left(f(\underline{W})_{\underline{j}}\right)}^{(II)} - \overbrace{f(\underline{W})_{\underline{j}} f(\underline{W})_{\underline{j}}^{\top}}^{(III)} \Big)}_{(I)} C^{(3)}[\cdot,i]. \qquad \text{(i.e., (3.5))}
$$

**Remark F.1** (Benefit from Tensor Trick: Speedup Seemingly Cubic Time Exact Computation)**.** Lemma 3.1 highlights the benefits of the tensor trick and the potential for speeding up *exact* LoRA adaptation on transformer-based models. To be more specific, for any $\underline{j} \in [L]$, **Part-(I)** is an $L \times L$ matrix, thus requiring $\Theta(L^2)$ time to compute. Moreover, with a total of $L$ terms, the overall computation time amounts to $\Theta(L^3)$.

However, (3.5) decomposes **Part-(I)** into a *diagonal* **Part-(II)** and a *low-rank* **Part-(III)** (specifically, rank-1). This decomposition allows us to reduce the computation time of **Part-(I)** to $O(L)$ for each $\underline{j} \in [L]$, and of the entire $\mathrm{d}\mathcal{L}(\underline{W})/\mathrm{d}\underline{W}$ to $O(L^2)$. Our next theorem verifies this claim and shows such seemingly cubic time exact computation is in fact quadratic.

**Definition F.1.** Let $n_1, n_2, n_3$ denote any three positive integers. We use $\mathcal{T}_{\mathrm{mat}}(n_1, n_2, n_3)$ to denote the time of multiplying an $n_1 \times n_2$ matrix with another $n_2 \times n_3$.

**Theorem F.1** (Exact LoRA Gradient Computation Takes Quadratic Time)**.** Suppose the following objects are given and if following conditions hold,
- Let $C^{(1)}, C^{(2)}, C^{(3)} \in \mathbb{R}^{L \times d}$ be in (3.2). Let $B_Q \in \mathbb{R}^{d \times r}, A_Q \in \mathbb{R}^{r \times d}, W \in \mathbb{R}^{d \times d}$ be in (3.3).
- Let $f(\cdot), c(\cdot), p_1(\cdot), p_2(\cdot)$ follow from their definitions in Section 3.
- Let $\underline{G}_Q^{(A)} := \frac{\partial \mathcal{L}}{\partial \underline{A}_Q}$, $\underline{G}_Q^{(B)} := \frac{\partial \mathcal{L}}{\partial \underline{B}_Q}$ (Where $\mathcal{L}$ is defined in (3.3) ).

Then we can make *exact* computation of $\underline{G}_Q^{(A)}, \underline{G}_Q^{(B)}$ in $O(\mathcal{T}_{\mathrm{mat}}(d, L, L) + \mathcal{T}_{\mathrm{mat}}(d, d, L) + \mathcal{T}_{\mathrm{mat}}(d, d, r))$ time.

*Proof.* Due to Lemma 3.2, it holds

$$
\frac{\partial \mathcal{L}}{\partial \underline{A}_Q} = \mathrm{vec}\left( B_Q^{\top}\left(C^{(1)}\right)^{\top} p(\underline{W}) C^{(2)} \right), \quad \frac{\partial \mathcal{L}}{\partial \underline{B}_Q} = \mathrm{vec}\left( \left(C^{(1)}\right)^{\top} p(\underline{W}) A_Q C^{(2)} \right).
$$

Recall that the decomposition of $p(\underline{W}) = p_1(\underline{W}) - p_2(\underline{W})$. And according to Definition 3.6, for every index $\underline{j} \in [L]$,

$$
p_1(\underline{W})_{\underline{j}} := \mathrm{diag}\left( f\left(\underline{W}\right)_{\underline{j}} \right) q(\underline{W})_{\underline{j}}, \quad p_2(\underline{W})_{\underline{j}} := f\left(\underline{W}\right)_{\underline{j}} f\left(\underline{W}\right)_{\underline{j}}^{\top} q(\underline{W})_{\underline{j}},
$$

In addition, due to Lemma 3.2, $q(\underline{W})$ is defined as

$$
q(\underline{W}) := C^{(3)}\left(c(\underline{W})\right)^{\top} \in \mathbb{R}^{L \times L}.
$$

Therefore, we compute $f(\underline{W}), c(\underline{W}), p_1(\underline{W}), p_2(\underline{W})$ in order as follows. Then we combine them together to get total running time.

- **Step 1.** We compute $f(\underline{W})$.

  Note that

$$
f(\underline{W}) = D^{-1} \exp\left( \overbrace{C^{(1)}}^{L \times d} \overbrace{W}^{d \times d} \overbrace{(C^{(2)})^{\top}}^{d \times L} \right),
$$

  where

$$
D^{-1} = \mathrm{diag}(\exp\left( C^{(1)} W (C^{(2)})^{\top} \right) \mathbb{1}_L).
$$

We firstly compute $\exp\left(C^{(1)} W (C^{(2)})^\top\right) C^{(3)}$ which takes time of $\mathcal{T}_{\mathrm{mat}}(d, d, L) + \mathcal{T}_{\mathrm{mat}}(d, L, L)$.

Then, we can compute $D$ which takes $O(L^2)$ time.

Then, we can compute $f(\underline{W})$ which takes $O(L^2)$ time.

Thus, the overall time is

$$\mathcal{T}_{\mathrm{mat}}(d, d, L) + \mathcal{T}_{\mathrm{mat}}(d, L, L) + O(L^2) = O(\mathcal{T}_{\mathrm{mat}}(d, d, L) + \mathcal{T}_{\mathrm{mat}}(d, L, L)).$$

Therefore, the proof is completed.

- **Step 2.** We compute $c(\underline{W})$. Based on the Definition 3.5, which is

$$c(\underline{W}) = \overbrace{f(\underline{W})}^{L \times L} \overbrace{C^{(3)}}^{L \times d} - Y.$$

Computing $f(\underline{W}) C^{(3)}$ takes time of $\mathcal{T}_{\mathrm{mat}}(d, L, L)$ and computing $f(\underline{W}) C^{(3)} - Y$ takes time of $O(Ld)$. Thus, the overall time is $\mathcal{T}_{\mathrm{mat}}(d, L, L) + O(Ld) = O(\mathcal{T}_{\mathrm{mat}}(d, L, L))$.

- **Step 3.** We compute $q(\underline{W})$. Recall that

$$q(\underline{W}) := \overbrace{c(\underline{W})}^{L \times d} \overbrace{(C^{(3)})^\top}^{d \times L}.$$

Therefore, it takes time $O(\mathcal{T}_{\mathrm{mat}}(d, L, L))$.

- **Step 4.** We compute $p(\underline{W})$. Note that due to Definition 3.6, which is

$$p_1(\underline{W})_{\underline{j}} := \mathrm{diag}\left(f\,(\underline{W})_{\underline{j}}\right) q(\underline{W})_{\underline{j}}, \quad p_2(\underline{W})_{\underline{j}} := f\,(\underline{W})_{\underline{j}}\, f\,(\underline{W})_{\underline{j}}^\top\, q(\underline{W})_{\underline{j}},$$

such that $p(\underline{W}) = p_1(\underline{W}) - p_2(\underline{W})$.

Since $\mathrm{diag}(f(\underline{W})_{\underline{j}})$ is a diagonal matrix and $f(\underline{W})_{\underline{j}}(f(\underline{W})_{\underline{j}})^\top$ is a rank-one matrix, we know that $p(\underline{W})_{\underline{j}} \in \mathbb{R}^L$ can be computed in $O(L)$, for each $\underline{j} \in [L]$. Thus we can construct matrix $p(\underline{W}) \in \mathbb{R}^{L \times L}$ in $L \times O(L) = O(L^2)$ time in total.

- **Step 5.** Using Lemma 3.2, we know that

$$\frac{\partial \mathcal{L}}{\partial \underline{A}_Q} = \mathrm{vec}(\overbrace{B_Q^\top}^{r \times d} \overbrace{(C^{(1)})^\top}^{d \times L} \overbrace{p(\underline{W})}^{L \times L} \overbrace{C^{(2)}}^{L \times d}), \quad \frac{\partial \mathcal{L}}{\partial \underline{B}_Q} = \mathrm{vec}(\overbrace{(C^{(1)})^\top}^{d \times L} \overbrace{p(\underline{W})}^{L \times L} \overbrace{A_Q}^{L \times d} \overbrace{C^{(2)}}^{L \times d}).$$

Suppose $B_Q \in \mathbb{R}^{d \times r}, A_Q \in \mathbb{R}^{r \times d}, C^{(1)}, C^{(2)}, C^{(3)} \in \mathbb{R}^{L \times d}$ are given, then each of the gradients can be computed in time of $O(\mathcal{T}_{\mathrm{mat}}(d, L, L) + \mathcal{T}_{\mathrm{mat}}(d, d, L) + \mathcal{T}_{\mathrm{mat}}(d, d, r))$.

Thus, the overall running time for gradients computation is

$$O(\mathcal{T}_{\mathrm{mat}}(d, L, L) + \mathcal{T}_{\mathrm{mat}}(d, d, L) + \mathcal{T}_{\mathrm{mat}}(d, d, r)).$$

This completes the proof. $\square$

# G  PROOF-OF-CONCEPT EXPERIMENTS

Here we provide minimally sufficient numerical results to back up our theory. For generality, we consider the full LoRA fine-tuning on $W_K, W_Q, W_V$ as analyzed in Appendix A.

**Objective: Control Norms of Attention Heads' Pretrained Weights to Achieve Speedup.** We use the outlier-removing transformer architecture proposed by Hu et al. (2024a) to showcase the efficiency gains from controlling the norms of $\{\|W_\mu\|, \|A_\mu\|, \|B_\mu\|\}_{\mu=K,Q,V}$. This type of architectures bounds these norms by preventing extreme weight values inherited from the pretraining process.

**Fine-Tuning Task.** We perform cross-modality fine-tuning on 3 sizes of the Open Pretrained Transformer (OPT) models (Zhang et al., 2022): OPT125M, OPT350M and OPT1.3B. Specifically, we adapt OPT language models to speech data, creating a SpeechLM (Speech Language Model) with both text and speech modalities, following (Maiti et al., 2024; Wu et al., 2024c).

**Pretrianed Model Setup.** We test our theory on three OPT model sizes: OPT125M, OPT350M, and OPT1.3B.

Table 1: **Training Time (Per Epoch) Comparison between LoRA on "Standard vs. Outlier-Free" Transformers for 3 OPT Model Sizes.** We perform full LoRA fine-tuning on $W_K, W_Q, W_V$ of the attention heads in Open Pretrained Transformers (OPTs) (Zhang et al., 2022). Our results show that, with norm-bound control, Outlier-Free Transformers (Hu et al., 2024a) are 5.5% faster for OPT-125M, 13.1% faster for OPT-350M, and 33.3% faster for OPT-1.3B.

| Model | Standard Transformer | Outlier-Free Transformer |
|---|---|---|
| OPT-125M | 58 mins | 55 mins (-5.2%) |
| OPT-350M | 69 min | 61 min (-11.6%) |
| OPT-1.3B | 84 min | 63 min (-25.0%) |

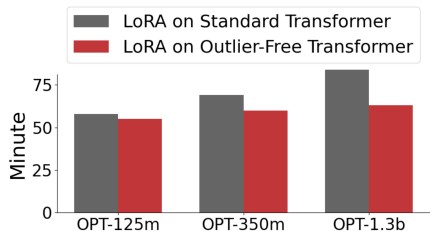

Figure 1

Each model size has two versions: one with standard transformers (Vaswani et al., 2017) and another with outlier-removing (outlier-free) transformers (Hu et al., 2024a). The training process for all OPT models follows (Hu et al., 2024a).

**LoRA Setup.** Following the original LoRA settings (Hu et al., 2021), we fine-tune the models using a rank of $r = 128$ and an alpha value of $\alpha = 256$.

**Data.** We use the LibriLight dataset (Kahn et al., 2020) for fine-tuning. LibriLight contains 60,000 hours of audiobook recordings from 7,000 speakers, totaling 12 million utterances.

**Computational Resource.** We conduct all experiments using 4 NVIDIA A100 GPU with 80GB of memory. Our code are based on standard PyTorch and the Hugging Face Transformer Library.

**Efficiency Results: Training Time Comparison.** To demonstrate the efficiency benefits of norm control suggested by Theorems 3.1, 4.1 and A.1, we compare the training speed of the two architectures. In Table 1 and Figure 1, we report the training time per epoch for both architectures across three model sizes. Our results indicate that the Outlier-Free Transformer is 5.5% faster for OPT-125M, 13.1% faster for OPT-350M, and 33.3% faster for OPT-1.3B.

These numerical results align with our theory: proper normalization of weights and inputs enhances LoRA training efficiency. Notably, we observe greater computational gains in larger models.

