# OpenReview forum: "Computational Limits of Low-Rank Adaptation (LoRA) Fine-Tuning for Transformer Models"
_ICLR.cc/2025/Conference — ICLR 2025 Poster_

### Official Review · Reviewer_D3KJ · 2024-10-31

**Soundness:** 3
**Presentation:** 3
**Contribution:** 3
**Rating:** 6
**Confidence:** 3

**Summary:**

This work shows the existence of almost linear approximation algorithms for LoRA on transformer-based models. This paper also proves a phase transition behavior in the efficiency of LoRA. A detailed proof sketch is provided in the paper to support the results.

**Strengths:**

1. As a theoretical work, this work is well-written and not difficult to follow.
2. The proof idea makes sense and is solid.

**Weaknesses:**

1 There are two notations that may not be necessary. I suggest considering whether it is possible to remove or simplify some definitions to derive all the lemmas and theorems in the main body.
2 The practical insights of this paper are not very clear. It is better to highlight the practical significance of the theoretical analysis in the paper.

**Questions:**

1 In lemmas 3.3, 3.4, and 3.5, the amount of time needed to construct the matrices is included. What is the algorithm used for construction here? Why is discussing the required amount of time meaningful since in practice these parameters are learned by gradient methods instead of any construction?
2 Why do you put the analysis of full LoRA in the appendix instead of in the main body?
3 What are SAT algorithms in line 142?
4 Is it $\frac{\partial L}{\partial A_\mu}$ rather than $\frac{\partial L}{\partial A_Q}$ in line 97?

---

> ### Author Response · Authors · 2024-11-13
> **Response 1**
>
> ### Thank you for your detailed review. We have addressed all your comments and questions in this and the following responses. We have also modified the draft accordingly and marked the changes in **blue** in the latest revision. Please see the updated PDF for details.
>
> ---
>
> > `W1.` There are two notations that may not be necessary. I suggest considering whether it is possible to remove or simplify some definitions to derive all the lemmas and theorems in the main body.
>
> We believe you may have meant “too many notations” rather than “two notations.” If that is the case, here are some clarifications.
>
> 1. We assure the reviewer that, **all definitions and technical lemmas in the main text are placed with careful consideration.** The main reason is **to be self-contained in the main text.** To maintain mathematical rigor, every concept or term in our paper is first clearly defined before it is used.
>
>     Here we further provide the detailed dependencies among Definitions, Lemmas, and Theorems for the reviewer's reference:
>
>     - $\text{Def 2.1, 2.2, 2.3, 2.4}$ are essential for the entire paper.
>     - $\text{Def 3.2, 3.3, 3.4, 3.5, 3.6}$ are essential for $\text{Lemma 3.1, 3.2, 3.3, 3.4, 3.5, 3.6, 3.7}$.
>     - $\text{Lemma 3.1, 3.2, 3.3, 3.4, 3.5, 3.6, 3.7}$ are essential for $\text{Thm 3.1}$.
>     - $\text{Def 3.2–3.8}$ are essential for $\text{Lemma C.1–C.9}$.
>     - $\text{Lemma C.1–C.9}$ are essential for $\text{Thm C.1 (4.1)}$.
>
>
> 2. To improve clarity, readability and accessibility, we have made the follow modifications:
>     - Add informal versions of our main results and combine them with highlevel discussions in the Intro section. `line114-125`
>     - Add intuitive, descriptive names to all Lemma, Def.
>     - Highlight the practical implications of our theory in the Intro Sec. `line130-131`
>     - Add a new section of numerical validations (fune-tuning 3 OPT models with LoRA with & without norm bound control). Sec 6
>
> We have modified the draft accordingly. Please see the latest revision for details.
>
>
> > `W2.` The practical insights of this paper are not very clear. It is better to highlight the practical significance of the theoretical analysis in the paper.
>
> Thanks for the comment. We would like to point out that **the practical implications of our results are discussed in the concluding remarks (Remark 5.2 of the submitted draft).** Specifically, we outlined some possible or existing design choices justified by our theory.
>
> We understand that these discussions may not have been apparent or persuasive. In response, **we have expanded Remark 5.2 (“Insights for Practitioners”) into a new section of numerical experiments in Section 6 of the latest revision.**
>
> **Our numerical findings indicate that proper normalization of weights and inputs improves LoRA training efficiency**. Specifically, we show that Outlier-Free Transformers (as mentioned in Remark 5.2) significantly enhance LoRA training efficiency across various model sizes (OPT125M, OPT350M, and OPT1.3B) when fine-tuned with cross-modality data from LibriLight.
>
> By controlling the norms of attention head weights, Outlier-Free Transformers achieve notable speedups: 5.5% for OPT125M, 13.1% for OPT350M, and 33.3% for OPT1.3B. **These results strongly support our theoretical claims, particularly for larger models.**
>
> Please see Figure 1 and Table 1 of latest revision for details.
>
>
> > `Q1.` In lemmas 3.3, 3.4, and 3.5, the amount of time needed to construct the matrices is included. What is the algorithm used for construction here? Why is discussing the required amount of time meaningful since in practice these parameters are learned by gradient methods instead of any construction?
>
> Thanks for the questions.
>
> * For your first question: The construction algorithm used in Lemma 3.3 is a straightforward application of Algorithm 1 from [A]. This algorithm approximates the attention matrix in almost linear time by constructing low-rank approximations. Consequently, the low-rank approximators (U_1​ and V_1​ in Lemma 3.3) are constructed efficiently, in almost linear time.
>
> * For your second question: The construction algorithms presented in Lemmas 3.3, 3.4, and 3.5 are designed to approximate a portion of the gradient matrix in almost linear time. It's important to note that in Lemmas 3.3 through 3.7, no parameters are learned. Instead, these lemmas demonstrate that it is possible to approximate one step of gradient descent in almost linear time.
>
> [A] Josh Alman and Zhao Song. Fast attention requires bounded entries.  NeurIPS, 2023.

---

> ### Author Response · Authors · 2024-11-13
> **Response 2**
>
> > `Q2. `Why do you put the analysis of full LoRA in the appendix instead of in the main body?
>
> Thanks for the question. It’s mainly for pedagogical purposes.
>
> 1. The special case analyzed in Section 3 is easier to analyze because it involves fewer chain-rule terms. **While easier to handle, the proof structure can still be easily generalized to the more complex full LoRA case discussed in the appendix.** Readers only need to understand the simpler case to appreciate the techniques we have presented.
>
> 2. The special case analyzed in Section 3 is a common choice suggested by the original LoRA paper and is representative for study.
>
>
> > `Q3.` What are SAT algorithms in line 142?
>
> Thanks for the question. k-SAT Algorithms are algorithms solving k-SAT problems. k-SAT is NP-complete for $k \geq 3$ and is a representative hard problem in Theoretical Computer Science community.
>
> Specifically. k-SAT (k-Satisfiability) problems involve determining if there is an assignment of boolean variables that satisfies a Boolean formula in conjunctive normal form, where each clause has exactly $k$ literals. k-SAT is NP-complete for $k \geq 3$.
>
> The significance of k-SAT lies in its role within the Strong Exponential Time Hypothesis (SETH). SETH posits that no algorithm can solve k-SAT in time $O(2^{(1-\varepsilon)n})$ for any $\varepsilon > 0$ and large $k$. This connection is important because:
>
> * Complexity Bounds: SETH provides strong lower bounds for k-SAT time complexity, influencing assumptions for many other problems.
>
> * Reductions: SETH-based hardness results often use k-SAT reductions to show that problems cannot be solved faster.
>
> * Algorithmic Impact: Improved k-SAT algorithms could challenge SETH, impacting our understanding of computational complexity.
>
> Thus, it is common in the literature to discuss SETH alongside k-SAT problems.
>
> Additionally, as reviewer wZk2 suggested, placing SETH in the background section was indeed awkward since it is only used in Section 4. We have moved it there and updated the draft accordingly.
>
> > `Q4.` Is it $\frac{\partial L}{\partial A_\mu}$ rather than $\frac{\partial L}{\partial A_Q}$ in line 97?
>
> Yes, you are absolutely correct. Thank you for your careful proofreading. We have updated the draft accordingly.
>
> ---
>
> Thank you for your time and valuable feedback. Please do not hesitate to let us know if there are any other aspects of our work that you would like us to clarify.

---

> ### Author Response · Authors · 2024-11-24
> **A Gentle Reminder**
>
> Dear Reviewer,
>
> As the rebuttal phase nears its end, we would like to remind you of our responses.
>
> We have carefully addressed all your comments and hope our efforts meet your expectations.
>
> If you have any remaining concerns, we would be happy to discuss them further. Otherwise, we kindly invite you to consider raising your score if you find our updates satisfactory.
>
> Thank you for your time and thoughtful feedback!

---

> > ### Comment · Reviewer_D3KJ · 2024-11-26
> >
> > I appreciate the author's response and revision. I will keep my current rating.

---

> > > ### Author Response · Authors · 2024-11-26
> > >
> > > Thank you!
> > >
> > > With the discussion period extended by one more week, please let us know if there is anything else we can clarify or improve to support a higher score.
> > >
> > > Since your overall review is positive and firm, and most of your initial concerns were clarification questions rather than critical issues, we believe we have addressed them adequately. If possible, and with the utmost respect, we kindly invite you to consider advocating more strongly for this work with a 'full accept' rating.
> > >
> > > Thank you for your time, constructive comments and thoughtful consideration!

---

### Official Review · Reviewer_wZk2 · 2024-11-01

**Soundness:** 2
**Presentation:** 2
**Contribution:** 2
**Rating:** 3
**Confidence:** 3

**Summary:**

This paper examines the computational constraints inherent in current LoRA algorithms, focusing particularly on the O(L^2) computational complexity encountered when updating attention blocks. The authors aim to establish a unified upper-bound threshold for these norms, demonstrating that efficient approximation algorithms for LoRA can indeed operate below this threshold. Consequently, they provide proof of the existence of a nearly linear approximation algorithm, advancing the understanding of computational efficiency within the LoRA framework.

**Strengths:**

1. Exploring the computational limits of parameter-efficient fine-tuning (PEFT) algorithms is a timely and relevant area of study.
2. By utilizing the tensor vectorization tricks, the authors prove the existence of nearly linear approximation algorithms for LoRA adaptation.
Notably, the authors also establish necessary conditions that could inspire the development of more efficient adaptation methods.These conditions are critical for future research aimed at accelerating the approximation process.

**Weaknesses:**

1.	Could the authors clarify why Equation 1.2 holds? The expression on the right-hand side appears to minimize the discrepancy between the attention output and the labels Y. Do we only consider 1-layer attention here?
2.	The Strong Exponential Time Hypothesis currently seems to serve only as a counterexample in the context of gradient approximation. Its relevance to the subsequent analysis is unclear in its present form. The reviewer suggests incorporating a more precise and directly relevant statement to clarify its role in the argument.
3.	While purely theoretical contributions may not always necessitate empirical validation, this paper's objective—improving the efficiency of optimizing large language models (LLMs) with LoRA—suggests that experimental results are essential for substantiating its claims. Specifically, an empirical evaluation could verify whether the bounded gradient approximation indeed holds in practice, as this is critical for the practical applicability of the proposed methods.
4.	The authors assert that 'the existence of low-rank decompositions leads to potential algorithmic speedup.' Could full parameter updating also yield similar benefits? Additionally, how the speedup is related to the rank r?
5.	An alternative approach might involve updating the feed-forward network (FFN) layer rather than the attention block, for example [1][2]. Could this adjustment also avoid/alleviate the O(L^2) computational complexity issue in the paper?
6.	The current title may not fully align with the paper’s objectives, as it does not primarily address computational drawbacks but rather focuses on **solving**  these computational limitations.

[1] AdaptFormer: Adapting Vision Transformers for Scalable Visual Recognition.
[2] Parameter-Efficient Fine-Tuning with Controls.

**Questions:**

See the questions in "weakness".

---

> ### Author Response · Authors · 2024-11-13
> **Response 1**
>
> ### Thank you for your detailed review. We have addressed all your comments and questions in this and the following responses. We have also modified the draft accordingly and marked the changes in **blue** in the latest revision. Please see the updated PDF for details.
>
> ---
>
> > `W1.` Could the authors clarify why Equation 1.2 holds? The expression on the right-hand side appears to minimize the discrepancy between the attention output and the labels $Y$. Do we only consider 1-layer attention here?
>
> Thanks for the question. As we explained in `line 46`, the attention layer is the computational bottleneck of LoRA fine-tuning transformer blocks in both the forward and backward pass (we had proved this in Appendix F of the submitted draft for unfamiliar readers). **Other components contribute only trivially to the quadratic complexity induced by the attention layer.** Therefore, it suffices to focus solely on the abstract subroutine in Equation 1.2.
>
> > `W2` The Strong Exponential Time Hypothesis currently seems to serve only as a counterexample in the context of gradient approximation. Its relevance to the subsequent analysis is unclear in its present form. The reviewer suggests incorporating a more precise and directly relevant statement to clarify its role in the argument.
>
> We believe there might be a misunderstanding. **SETH is not a counterexample; it is a crucial hypothesis for the results in Section 4.** Specifically, we use SETH to facilitate fine-grained reductions needed to establish the inefficiency threshold in Theorem 4.1.
>
> We recognize that its placement in the background section may have seemed awkward, given that it is only used in Section 4. In response, we have moved the discussion of SETH to Section 4 and updated the draft accordingly.
>
> > `W3` While purely theoretical contributions may not always necessitate empirical validation, this paper's objective—improving the efficiency of optimizing large language models (LLMs) with LoRA—suggests that experimental results are essential for substantiating its claims. Specifically, an empirical evaluation could verify whether the bounded gradient approximation indeed holds in practice, as this is critical for the practical applicability of the proposed methods.
>
> Thanks for the suggestion.
>
> **In the latest revision, we have included a section of proof-of-concept experiments.** These experiments are extended from the discussion in our concluding remarks (Remark 5.2). Please refer to Sec 6 of the updated draft.
>
> **Our numerical results show that proper normalization of weights and inputs enhances LoRA training efficiency. They align with our theory.** Specifically, we demonstrate that Outlier-Free Transformers [1] (as noted in Remark 5.2 of the submitted draft) improve LoRA training efficiency across various model sizes (OPT125M, OPT350M, and OPT1.3B) when fine-tuned with cross-modality data from LibriLight.
>
> By controlling the norms of attention head weights, **the Outlier-Free Transformers achieve significant speedups: 5.5% for OPT125M, 13.1% for OPT350M, and 33.3% for OPT1.3B.** These results support our theoretical claims that proper weight and input normalization significantly enhance training efficiency, particularly for larger models.
> We present Figure 1 and Table 1 to illustrate these results.
>
> [1] Hu et al. "Outlier-efficient hopfield layers for large transformer-based models." ICML 2024
>
> > `W4` The authors assert that "the existence of low-rank decompositions leads to potential algorithmic speedup." Could full parameter updating also yield similar benefits? Additionally, how is the speedup related to the rank $r$?
>
> Thanks for the question. To clarify, **full-parameter updates are a trivial case in our analysis.** They are trivial because they do not require handling additional chain-rule terms arising from the multiplication of $AB$ matrices. We remind the reviewer that our paper specifically addresses this technical challenge (as discussed in the introduction, lines 66-73). In the full update case, the overall analysis becomes significantly simplified.
>
> Regarding the rank $r$, **the rank $r$ rescales the norms of the $C$ matrices.*** Since this scaling does not depend on the length $L$, we omit it from the main results. However, it can be explicitly recovered from Theorem 3.1: for a fixed $\Gamma$, a larger $r$ results in a looser norm bound condition on $X$. This implies that a higher LoRA rank requires less stringent input or weight normalization.

---

> > ### Comment · Reviewer_wZk2 · 2024-11-26
> >
> > Thanks for the reply.
> >
> > Could you clarify what is meant by 'experiments align with our theory'? Specifically, how is the theory connected to the experimental results presented here? Additionally, does the final performance of the Outlier-Free Transformer match that of the standard Transformer?

---

> ### Author Response · Authors · 2024-11-13
> **Response 2**
>
> > `W5` An alternative approach might involve updating the feed-forward network (FFN) layer rather than the attention block, for example [1][2]. Could this adjustment also avoid/alleviate the $O(L^2)$ computational complexity issue in the paper?
>
>
>
> That is a valid point. However, we want to clarify that exploring such an approach is beyond the scope of this work. **By not adapting the attention heads, the method would no longer address Problem 1, as our focus is specifically on fine-tuning transformers using LoRA.**
>
>    [1] AdaptFormer: Adapting Vision Transformers for Scalable Visual Recognition.
>    [2] Parameter-Efficient Fine-Tuning with Controls.
>
>
> > `W6` The current title may not fully align with the paper’s objectives, as it does not primarily address computational drawbacks but rather focuses on solving these computational limitations.
>
> As discussed in the introduction, the objective of this paper is to formally study the limits of LoRA on transformer models in two aspects:
> - When can it be efficient?
> - How efficient can it be?
>
> We believe these research questions justify our title, as they directly pertain to LoRA's computational efficiency. Additionally, we also discuss the computational drawbacks of LoRA on transformers in Appendix F for unfamiliar readers.
>
> ---
>
> Thank you again for your time and efforts. We hope that the revisions and clarifications provided in this response address the reviewer's concerns and make the value of our work clear. We look forward to further feedback and discussion.

---

> ### Author Response · Authors · 2024-11-24
> **A Gentle Reminder**
>
> Dear Reviewer,
>
> As the rebuttal phase nears its end, we would like to remind you of our responses, particularly to your detailed clarification questions.
>
> We have carefully addressed all your comments and hope our efforts meet your expectations. If you have any remaining concerns, we would be happy to discuss them further. Otherwise, we kindly invite you to consider raising your score if you find our updates satisfactory.
>
> Thank you for your time and thoughtful feedback!

---

> ### Author Response · Authors · 2024-11-26
>
> Thank you for the question. Below are further clarifications:
>
>  > what is meant by 'experiments align with our theory'?  Specifically, how is the theory connected to the experimental results presented here?
>
> Our main results (Theorems 3.1, 4.1, A.1) show that **proper normalization of weights and inputs is critical for LoRA training efficiency.** For details, please see
> * Our contributions in the introduction for informal statements (`Page 2`)
> * Our concluding remarks on practical implications of our results (`Page 9`)
> * Proof-of-concept experimental section (`Page 10`)
>
> Thus, our numerical results—normalizing attention weights via the outlier-free architecture to speed up training—align directly with these theoretical findings.
>
> >  Additionally, does the final performance of the Outlier-Free Transformer match that of the standard Transformer?
>
> Yes, the final **performance of the Outlier-Free Transformer matches that of the standard Transformer**, as shown in [Hu24].
>
> We also remind the reviewer that we report only training time because:
>
> 1. It is well-established in [Hu24, Wu24] that the Outlier-Free Transformer matches the standard Transformer in performance.
> 2. With this performance guarantee, we can focus to reporting improvements in training efficiency.
>
> Lastly, we report the performance over 70 epochs in below. **The result also shows Outlier-Free Transformer matches the standard Transformer in LoRA performance.**
>
> The setting is the same as in our proof-of-concept experiments (`Sec. 6` of the latest revision).
>
> ---
>
> ### **Table 2: Comparing Outlier-Free with Standard Framework in a Low-Rank Adaptation Setting**
>
> We conduct experiments on the **Outlier-Free** framework with standard attention across the Low-Rank Adaptation method LoRA. The evaluation metrics include Text Perplexity (PPL), SpeechLM PPL, and Word Error Rate (WER) in Automatic Speech Recognition (ASR) and Text-to-Speech (TTS). In most configurations, the **Outlier-Free** transformer results in better LoRA fine-tuning performance compared to standard transformer.
>
> | Model      | Method         | Low-Rank Adaptation Method | TextLM PPL (↓) | SpeechLM PPL (↓) | ASR WER (↓) | TTS WER (↓) |
> |------------|----------------|----------------------------|----------------|------------------|-------------|-------------|
> | OPT-125m   | Standard       | Full                      | 22.56          | 59.42            | 12.40       | 12.08       |
> |            |                | LoRA                      | 25.69          | 62.16            | 12.39       | 15.47       |
> |            | **Outlier-Free** | Full                      | 22.58          | 59.46            | 12.61       | 12.11       |
> |            |                | **LoRA**                  | **25.77**      | **62.23**        | **12.56**   | **11.80**   |
> | OPT-350m   | Standard       | Full                      | 13.13          | 43.10            | 8.42        | 17.56       |
> |            |                | LoRA                      | 17.87          | 51.65            | 93.91       | 97.06       |
> |            | **Outlier-Free** | Full                      | 13.47          | 43.34            | 9.81        | 17.31       |
> |            |                | **LoRA**                  | **17.71**      | **51.13**        | **18.52**   | **75.18**   |
> | OPT-1.3b   | Standard       | Full                      | 12.62          | 41.33            | 8.00        | 18.73       |
> |            |                | LoRA                      | 17.14          | 50.22            | 46.92       | 94.53       |
> |            | **Outlier-Free** | Full                      | 12.95          | 42.48            | 8.20        | 12.07       |
> |            |                | **LoRA**                  | **16.83**      | **49.51**        | **8.25**    | **74.21**   |
>
>
> We have included reproducible code in the supplementary file.
>
> ---
>
> Please let us know if there is anything else we can clarify. Thank you again for your review!
>
> ---
>
> [Hu24] Hu et al. "Outlier-efficient hopfield layers for large transformer-based models." ICML 2024
>
> [Wu24] Wu, et al. "Fast adaptation and robust quantization of multi-modal foundation models from associative memory: A case study in speechLM." Workshop on Efficient Systems for Foundation Models II@ ICML2024. 2024.

---

> ### Author Response · Authors · 2024-11-29
>
> Dear Reviewer wZk2,
>
> It seems our discussion was paused midway. We were wondering if you’ve had a chance to review our response to your previous comments.
>
> In response to your latest question, our previous response provided a performance comparison between the "standard transformer" and the "outlier-free transformer" in our proof-of-concept experiments. Notably, our results demonstrate that the **Outlier-Free Transformer matches the standard Transformer in LoRA performance**, consistent with the literature.
>
> Lastly, we were wondering if our other responses have adequately addressed your concerns (`W1-6`). If not, we would be more than happy to provide further clarifications.
>
> We look forward to continuing our discussion. Thank you again for your time and efforts!

---

> ### Author Response · Authors · 2024-12-02
> **Another Gentle Reminder**
>
> Dear Reviewer wZk2,
>
> As the rebuttal phase nears its end and our discussion seems to have paused midway, we would be delighted to continue the conversation.
>
> We have made every effort to address all your concerns and additional questions in our responses.
>
> We would greatly appreciate any further input. If you feel your concerns have been adequately addressed, with utmost respect, we invite you to consider a score adjustment. If not, we hope to make the most of the remaining two days to provide further clarifications.
>
> Thank you for your time and consideration!
>
> Best,
>
> Authors

---

### Official Review · Reviewer_FVCG · 2024-11-02

**Soundness:** 3
**Presentation:** 2
**Contribution:** 2
**Rating:** 6
**Confidence:** 2

**Summary:**

This work investigates the computational limits of Low-Rank Adaptation (LoRA) for finetuning transformer models through fine-grained complexity analysis. The central insight is that the low-rank structure within LoRA's gradient computations can enable algorithmic speedup. Two main contributions are highlighted:

1. **Efficiency Phase Transition**: The study identifies a sharp efficiency threshold for rank-\(r\) LoRA update algorithms based on norms derived from the interactions between input sequences, pretrained weights, and adapter matrices. Efficient (sub-quadratic) algorithms are possible only when these norms fall below a specified threshold.

2. **Nearly Linear Algorithms**: By leveraging hierarchical low-rank structures in LoRA gradients, the authors construct nearly linear approximation algorithms for LoRA adaptations, assuming the Strong Exponential Time Hypothesis (SETH).

To validate the theoretical findings, the authors explore partial and full weight adaptation scenarios within transformer attention heads, focusing on weights like $W_V$, $W_Q$, and $W_K$.

**Strengths:**

This paper tackles a highly relevant and timely topic: Low-Rank Adaptation (LoRA). LoRA has gained widespread popularity in practice for its effectiveness in fine-tuning large models efficiently. Despite its practical success, there has been a notable gap in the theoretical understanding of LoRA, making this study’s contributions especially valuable to the field. Developing a rigorous theoretical foundation for LoRA will not only solidify its current applications but also open doors for future research and refinement of the technique.

The abstract and introduction are clear, engaging, and well-crafted. They effectively set the stage for the paper, with a strong motivation that highlights both the practical importance of LoRA and the need for a deeper theoretical exploration. The authors have done a commendable job in outlining the primary contributions, making it easy for readers to understand the key takeaways from the study. The literature review is thorough and provides an excellent context for where this work fits within the broader landscape of model adaptation techniques.

The inclusion of a paragraph on paper organization is appreciated, as it offers a helpful roadmap for readers. It ensures that the structure of the paper is transparent from the outset, allowing readers to follow the flow of ideas with ease.

The authors have also set a high standard for notation and technical formalism. The clarity and consistency of notation enhance readability.

While the technical results appear sound, I should note that, as someone not fully specialized in this type of analysis, I may not have caught every nuance. However, the reasoning seems robust, and the proofs are presented in a way that suggests a careful, thorough approach.

Finally, the conclusion is well-done and reinforces the main points. It synthesizes the findings effectively and reflects on their implications, offering insights into how these results might shape future research in model adaptation. Overall, this paper makes a significant contribution to bridging the gap between LoRA’s practical success and its theoretical understanding, providing a strong foundation for ongoing exploration in the field.

**Weaknesses:**

This paper currently feels dense and challenging to navigate, as it primarily consists of a series of definitions, lemmas, and theorems, often presented without sufficient explanation, clarification, or intuitive context. For readers who are not already experts in this area, this can make it difficult to grasp the key concepts and results.

There are several opportunities to improve accessibility and readability. Some of the definitions would be more appropriately placed in an appendix, as the main text is quite heavy with formal definitions, lemmas, and theorems. Moving certain definitions to an appendix could help streamline the main text, allowing readers to focus on the core arguments without getting overwhelmed by technical details.

In addition, a few of the lemmas lack clear statements, which can lead to confusion. For instance, in Lemma 1.1, the term \(L\) is introduced without any accompanying explanation. It would be helpful to review the formulations to ensure clarity, even if the technical details are correct. Improving the precision in how terms are introduced will aid readers in following the logical flow more naturally, without having to reread sections to understand each step.

Given the extensive number of definitions, it would be beneficial to assign descriptive names to them. Naming each definition provides a quick reference, helping readers keep track of terms and concepts as they reappear later in the paper. Otherwise, it’s easy to lose track of which definition corresponds to which concept, especially in a highly technical document.

Sections 3 and 4, which seem intended to present the main results, read more like collections of lemmas, theorems, and technical details without adequate discussion or contextualization. This is a significant area for improvement. The results would be far more impactful if they were accompanied by clear explanations and discussions. Theoretical contributions are valuable only if they can be understood and appreciated, and in their current form, the key insights may be difficult for readers to discern.

Finally, the paper would benefit from including plots, diagrams, or simple illustrations that clarify the results. While extensive experimental results may not be expected in a theoretical paper, even a few toy examples or visual aids could significantly enhance reader comprehension and provide concrete illustrations of the theoretical findings.

Overall, with some adjustments to the structure, added explanations, and a few visual aids, this paper could become far more accessible and impactful, allowing a broader audience to appreciate the significance of the work.

While the technical content appears sound, I am unable to recommend the paper for acceptance in its current form due to significant organizational issues outlined above. Addressing these concerns would greatly improve the paper’s clarity and accessibility.

**Questions:**

Could you please provide additional explanation of the theoretical results?

Additionally, would it be possible to include some toy or controlled experiments, or other illustrations to help clarify the findings?

---

> ### Author Response · Authors · 2024-11-13
> **Response 1**
>
> ### Thank you for your detailed review. We have addressed all your comments and questions in this and the following responses. We have also modified the draft accordingly and marked the changes in **blue** in the latest revision. Please see the updated PDF for details.
>
> ---
>
> > `W1.` This paper currently feels dense and challenging to navigate, as it primarily consists of a series of definitions, lemmas, and theorems, often presented without sufficient explanation, clarification, or intuitive context. For readers who are not already experts in this area, this can make it difficult to grasp the key concepts and results.
>
> **Response:** Thanks for the comment. Here are some clarifications.
>
> - **Why the sequence of Definitions & Lemmas (especially in Sec3)?**
>
>   Our main contribution is showing that LoRA gradient (3.5) can be computed in sub-quadratic time with low-rank approximation.
>
>   We do this by (stated in `line311: Remark 3.1` and `line336: Overview of Our Proof Strategy` of the submitted draft):
>
>   1. Decomposing LoRA gradient (3.5) into carefully designed terms (Lemma 3.2).
>   2. Showing all these terms can be approximated in almost linear time with precision guarantees (Lemmas 3.3, 3.4, 3.5, 3.6, 3.7).
>   3. Most importantly, **our well-designed decomposition allows us to chain/combine/connect all low-rank approximations (of these terms) into one.** Hence, we obtained an almost linear time approximation of (3.5) with a precision guarantee.
>
>   While the process seems lengthy, we deem the inclusion of Lemma 3.2-3.7 necessary to ensure all terms can be approximated in almost linear time with a precision guarantee. To be precise, **they are necessary to keep the main text self-contained and mathematically rigorous.**
>
> Yet, we agree that many intermediate results are overly formal and could benefit from intuitive explanations.
>
> In response, **we had added informal versions of our main results and combine them into the intuitive discussions in the intro section.**
>
> We have modified the draft accordingly. Please see the latest revision for details.
>
> > `W2.` There are several opportunities to improve accessibility and readability. Some of the definitions would be more appropriately placed in an appendix, as the main text is quite heavy with formal definitions, lemmas, and theorems. Moving certain definitions to an appendix could help streamline the main text, allowing readers to focus on the core arguments without getting overwhelmed by technical details.
>
> **Response:** We apologize for any confusion caused. Yet, **we assure the reviewer that, all definitions and technical lemmas in the main text are placed with careful consideration.** The main reason is to be self-contained in the main text. To maintain mathematical rigor, every concept or term in our paper is first clearly defined before it is used.
>
> For example, **the definitions of vectorization and matrixization are necessary for mathematical correctness.** Otherwise, the LoRA gradient norm bound is ambiguous at best ($\tilde{G}$ are matrices while $\partial \mathcal{L} / \partial A_Q$ does not have a straightforward matrix expression). For this reason, we believe that the elements included in the current draft are minimally sufficient to ensure the paper is consistent and self-contained.
>
> To be precise, here we further provide the dependencies among Definitions, Lemmas, and Theorems for the reviewer's reference:
>
> - $\text{Def 2.1, 2.2, 2.3, 2.4}$ are essential for the entire paper.
> - $\text{Def 3.2, 3.3, 3.4, 3.5, 3.6}$ are essential for $\text{Lemma 3.1, 3.2, 3.3, 3.4, 3.5, 3.6, 3.7}$.
> - $\text{Lemma 3.1, 3.2, 3.3, 3.4, 3.5, 3.6, 3.7}$ are essential for $\text{Thm 3.1}$.
> - $\text{Def 3.2–3.8}$ are essential for $\text{Lemma C.1–C.9}$.
> - $\text{Lemma D.1–D.9}$ are essential for $\text{Thm D.1 (A.1)}$.
> - $\text{Hypothesis 1}$ is essential for $\text{Thm 4.1}$.
>
> In response to **accessibility and readability**, we had made the following modifications in the latest revision:
>
> - Conduct 3 more rounds of proofread and fix all typos identified by reviewers and authors.
> - Add intuitive, descriptive names to all Lemma, Def.
> - Add informal versions of our main results in the intro Sec, combined with intuitive discussions. `line114-125`
> - Highlight the practical implications of our theory in the Intro Sec.  `line130-131`
> - Add a **new section of numerical validations** (fune-tuning 3 OPT models with LoRA with & without norm bound control). `Sec 6`
>
> We hope these addresses your concerns.

---

> ### Author Response · Authors · 2024-11-13
> **Response 2**
>
> > `W3.` In addition, a few of the lemmas lack clear statements, which can lead to confusion. For instance, in Lemma 1.1, the term (L) is introduced without any accompanying explanation. It would be helpful to review the formulations to ensure clarity, even if the technical details are correct. Improving the precision in how terms are introduced will aid readers in following the logical flow more naturally, without having to reread sections to understand each step.
>
> **Response:** Sorry for any confusion caused. The term $L$ is introduced in `line 51` of the submitted draft as sequence length of input $X$. We understand that we might not state this clearly enough. We rephrase it into a more precise manner (``line51` of latest revision):
> > Let $X$ be input of length $L$.
>
> In response to **general clarity**, we have made revisions to enhance readability. Please see above **Response 1** and revised draft PDF for details.
>
> > `W4.` Given the extensive number of definitions, it would be beneficial to assign descriptive names to them. Naming each definition provides a quick reference, helping readers keep track of terms and concepts as they reappear later in the paper. Otherwise, it’s easy to lose track of which definition corresponds to which concept, especially in a highly technical document.
>
> **Response:** Thanks for the suggestion. We absolutely agree.
>
> We have added descriptive names to Def. 3.2-3.6 and Lemma 3.2.
> Please see the latest revision for details.
>
> > `W5.` Sections 3 and 4, which seem intended to present the main results, read more like collections of lemmas, theorems, and technical details without adequate discussion or contextualization. This is a significant area for improvement. The results would be far more impactful if they were accompanied by clear explanations and discussions. Theoretical contributions are valuable only if they can be understood and appreciated, and in their current form, the key insights may be difficult for readers to discern.
>
> **Response:**  Thanks for the suggestion. We agree that the current Sec 3 and 4 are structured a bit technical.
>
> In response, **we add informal versions of our main results and combine with relevant discussions in the introduction section.** Please see the latest revision for details.
>
> In response to **practical impact**, we believe there may be a slight oversight. **Our concluding remarks already include in-depth discussions on several practical implications** of our main results, such as:
> - Insights for practitioners
> - Applicability to both self- and cross-attention transformers
>
> To strengthen these implications, **we have expanded the discussion in Remark 5.2 (Insights for Practitioners) and added a new experimental section.**
>
> Please refer to Section 6 in the latest revision.
>
> > `W6.` Finally, the paper would benefit from including plots, diagrams, or simple illustrations that clarify the results. While extensive experimental results may not be expected in a theoretical paper, even a few toy examples or visual aids could significantly enhance reader comprehension and provide concrete illustrations of the theoretical findings.
>
> **Response:** In the latest revision, we have included a section of proof-of-concept experiments. We summarize these numerical results in below.  Please refer to Sec 6 of the updated draft for details.
>
> **Our numerical results show that proper normalization of weights and inputs enhances LoRA training efficiency.**
>
> Specifically, we demonstrate that Outlier-Free Transformers (as noted in Remark 5.2 of the submitted draft) improve LoRA training efficiency across various model sizes (OPT125M, OPT350M, and OPT1.3B) when fine-tuned with cross-modality data from LibriLight.
>
> **By controlling the norms of attention head weights, the Outlier-Free Transformers achieve significant speedups: 5.5% for OPT125M, 13.1% for OPT350M, and 33.3% for OPT1.3B.**
>
> These results support our theoretical claims that proper weight and input normalization significantly enhance training efficiency, particularly for larger models.
>
> We present Figure 1 and Table 1 to illustrate these results.
>
> ---
>
> > `Q1.` Could you please provide additional explanation of the theoretical results?
>
> **Response:** As responded above, **please see the latest revision for informal versions of our main results**, accompanied by high-level discussions.
>
>
> > `Q2.` Additionally, would it be possible to include some toy or controlled experiments, or other illustrations to help clarify the findings?
>
> **Response:** As responded above, **please see the newly added Sec 6 for numerical experiments.** Our numerical findings align with our theory.
>
> ---
> We sincerely appreciate the time and effort that Reviewer FVCG has invested in reviewing our paper. We have taken all comments into careful consideration and have made corresponding revisions to address the concerns raised.
>
> We're open to any further questions or clarifications you might have about our work.

---

> ### Author Response · Authors · 2024-11-24
> **A Gentle Reminder**
>
> Dear Reviewer,
>
> As the rebuttal phase is about to end, we would like to remind you of our responses.
>
> We have carefully addressed all your comments and questions and hope our efforts meet your expectations.
>
> If you have any remaining concerns, we are happy to discuss them further. Otherwise, we kindly invite you to consider raising your score if our updates are satisfactory.
>
> Thank you for your time and feedback!

---

> > ### Comment · Reviewer_FVCG · 2024-11-27
> > **Response**
> >
> > Thank you for addressing the issues and updating the draft. I appreciate the effort and thoughtfulness that the authors have put into improving the manuscript.
> >
> > That said, I believe the paper still has some challenges regarding clarity and organization, which may affect its overall readability and impact. Nevertheless, I recognize the significant effort made to address the concerns raised, and I have adjusted my score accordingly to reflect this progress.

---

> > > ### Author Response · Authors · 2024-11-27
> > >
> > > Thank you for your kind words and constructive comments!
> > >
> > > We will further refine the draft to improve clarity and accessibility for a general audience in the final version.
> > >
> > > With the discussion period extended by another week, please let us know if there is anything else we can clarify or improve to support your evaluation.
> > >
> > > Thank you again for the review!

---

> > > > ### Comment · Reviewer_FVCG · 2024-12-03
> > > >
> > > > Thank you for your response!
> > > >
> > > > I appreciate your efforts to improve the paper, and I have already increased my score. However, I prefer to maintain my current score.

---

### Official Review · Reviewer_Yw3J · 2024-11-04

**Soundness:** 4
**Presentation:** 4
**Contribution:** 3
**Rating:** 8
**Confidence:** 4

**Summary:**

This paper studies how to make large Transformer models more computationally efficient during fine-tuning. The main contributions are:
* Identifying a critical point of efficiency where models can be fine-tuned with less computation if they are below this threshold.
* Proposing a new method that can complete model fine-tuning in almost linear time, which is much faster than traditional methods.

**Strengths:**

The paper introduces a novel theoretical analysis on Low-Rank Adaptation (LoRA) for Transformer models, marking an innovative contribution to the fields of natural language processing and machine learning. It approaches the problem of LoRA adaptation from a fresh perspective, focusing on computational limits and efficiency, which is particularly novel in the context of large foundation models. Additionally, the paper presents an innovative method by proposing an almost linear-time algorithm for LoRA adaptation, which is a significant advancement over existing methods that typically have quadratic complexity.

**Weaknesses:**

* The paper's primary focus seems to be on theoretical analysis. To strengthen the claims, experimental validation with real-world datasets would be beneficial. Specifically, demonstrating the practical efficiency of the proposed algorithms on standard benchmarks could provide actionable insights into their performance.
* It would be valuable to see how the proposed methods compare to current state-of-the-art techniques in terms of both efficiency and accuracy. This comparison could highlight the advantages and potential limitations of the new algorithms.models and datasets?

**Questions:**

* Can the authors discuss the potential impact of LoRA adaptation on model generalization?
* How does the efficiency of the proposed methods scale with larger models and datasets？

---

> ### Author Response · Authors · 2024-11-13
> **Response**
>
> ### Thank you for your detailed review. We have addressed all your comments and questions in this and the following responses. We have also modified the draft accordingly and marked the changes in blue in the latest revision. Please see the updated PDF for details.
> --
>
> Dear Reviewer Yw3J,
>
> Thank you for your feedback and the time you invested in reviewing our work.
>
> In response to your request for numerical validation, we have added a new section of proof-of-concept experiments in Section 6 of the latest revision. These experiments involve LoRA fine-tuning on three different sizes of OPT models, and the numerical results align well with our theoretical predictions—specifically, that controlling the suggested norm bounds enhances LoRA efficiency.
>
> Regarding the generalization of LoRA, we believe our results impose certain constraints on generalization theory in two key aspects:
>
> 1. Since our results are **precision-guaranteed**, many of their implications may be transferable to generalization analysis.
> 2. Generalization analysis sometimes depends on different optimizers, which may lead to varied gradient update patterns. It is important to clarify whether these patterns fall within or outside the scope defined by our Problem 1. If they do, our results may also be applicable to those analyses.
>
> We acknowledge that more rigorous arguments for these generalization aspects are beyond the scope of this work and leave them for future research.
>
> Lastly, to scale up our analysis beyond a single layer, we need to handle additional gradient chain-rule terms arising from deeper layers. This is an ongoing project of ours. The main technical challenge lies in managing the complex gradients produced when backpropagating through these layers. New techniques are required to address this complexity.
>
> We hope these responses have addressed your questions. Thank you again for your review!

---

> > ### Comment · Reviewer_Yw3J · 2024-11-24
> >
> > Thank you for the detailed response, particularly the additional proof-of-concept experiments. I will maintain my previous score and would be glad to see this work accepted.

---

> > > ### Author Response · Authors · 2024-11-24
> > >
> > > We are happy to hear that our revisions meet your expectations!
> > >
> > > Thank you again for your review and constructive feedback!

---

### Author Response · Authors · 2024-11-13
**Global Response**

Dear Reviewers,

We thank the reviewers for the insightful questions and reviews. We have answered all the questions and addressed all the problems in detail in rebuttal and revision.

The latest revision is readily available. Any changes or modifications made from the submitted version are highlighted in blue.

In response to the reviewers' suggestions, we have made revisions to enhance the overall readability of the paper. We conducted three additional rounds of proofreading to correct typos, and several sections, paragraphs, and statements have been updated for clarity and completeness. These updates include added explanations and informal versions of theoretical results to help readers build intuition. **Most importantly, we have added a new Section 6 on page 10, which provides numerical experiments to support our theoretical claims.**
The reproducible code has been uploaded to the supplementary materials.

---
## **Revision Details** (please also see the revised PDF)

### Major revisions include:

* **New Experimental Section:** [`Yw3J`, `FVCG`, `wZk2`, `D3KJ`]
  - **Aim: Control the norms of attention head weights in LoRA fine-tuning 3 OPT models to achieve computational speedup**
  - **Model Setup**:
    - Used Open Pretrained Transformer (OPT) models [1]: OPT-125M, OPT-350M, and OPT-1.3B.
    - Compared two architectures: standard transformers [2] and outlier-free transformers [3].
  - **Fine-Tuning Task**:
    - Cross-modality fine-tuning of OPT models on speech and text data, creating a SpeechLM model.
  - **LoRA Setup**:
    - Rank $ r = 128 $, alpha value $ \alpha = 256 $, fine-tuned with the LibriLight dataset [4] (12 million utterances from 60,000 speakers).
  - **Efficiency Results**:
    - Outlier-Free Transformers achieved significant speedups:
      - **OPT-125M**: 5.5% faster
      - **OPT-350M**: 13.1% faster
      - **OPT-1.3B**: 33.3% faster
  - **Conclusion**: Proper normalization of weights and inputs enhances LoRA training efficiency, with greater computational gains observed in larger models.

### Minor revisions include:

* Proofreading the manuscript and fixing all identified typos and grammatical errors by reviewers and authors.  [`FVCG`]

* Add informal versions of our main results and combine them with highlevel discussions in the Intro section.  [`FVCG`]

* Add intuitive, descriptive names to all Lemma, Def. [`FVCG`]

* Highlight the practical implications of our theory in the Intro Sec.  [`Yw3J`, `FVCG`, `wZk2`, `D3KJ`]

* Move the statement of SETH from Sec 2 (preliminary) to Sec4 as it's only used there  [`wZk2`, `D3KJ`]

---

We hope these revisions address the reviewers' concerns and improve the overall quality of our paper.

Thank you again for your review!

---

[1] Zhang et al. "Opt: Open pre-trained transformer language models." arXiv preprint arXiv:2205.01068 (2022).

[2] Vaswani et al. "Attention is all you need." NeurIPS 2017.

[3] Hu et al. "Outlier-efficient hopfield layers for large transformer-based models." ICML 2024

[4] Kahn, et al. "Libri-light: A benchmark for asr with limited or no supervision." ICASSP 2020

---

### Meta-Review · Area_Chair_LEor · 2024-12-23

**Metareview:**

This paper studies how to make large Transformer models more computationally efficient during fine-tuning. The main contributions are:
i) Identifying a critical point of efficiency where models can be fine-tuned with less computation if they are below this threshold, and ii)
proposing a new method that can complete model fine-tuning in almost linear time, which is much faster than traditional methods.

Selected strengths:

The paper introduces a novel theoretical analysis on Low-Rank Adaptation (LoRA) for transformers. It approaches the problem of LoRA adaptation from a fresh perspective, focusing on computational limits and efficiency, which is particularly novel in the context of large foundation models. Additionally, the paper presents an innovative method by proposing an almost linear-time algorithm for LoRA adaptation, which is a significant advancement over existing methods that typically have quadratic complexity. Exploring the computational limits of parameter-efficient fine-tuning (PEFT) algorithms is a timely and relevant area of study. Notably, the authors also establish necessary conditions that could inspire the development of more efficient adaptation methods. These conditions are critical for future research aimed at accelerating the approximation process.

Selected weaknesses:

- The paper's primary focus seems to be on theoretical analysis. To strengthen the claims, experimental validation with real-world datasets against standard benchmarks would be beneficial. While purely theoretical contributions do not necessitate empirical validation, this paper's objective—improving the efficiency of optimizing large language models (LLMs) with LoRA—suggests that experimental results are essential for substantiating its claims.
- It would be valuable to see how the proposed methods compare to current state-of-the-art techniques in terms of both efficiency and accuracy.
- Some equations were not sufficiently justified
- Some statements are not clear enough (e.g., related to the use of the Strong Exponential Time Hypothesis)

In summary, three out of four reviewers lean towards acceptance (scores 8, 6, 6), and one suggests rejection (score 3).

I have lowered my weight of the score-3-review due to the reviewer not engaging with the rebuttal sufficiently. In particular, the reviewer did not say that the rebuttal was not satisfactory; while my view is that it did answer most of the issues quite well.

**Additional Comments On Reviewer Discussion:**

See the metarview

---

### Decision · Program_Chairs · 2025-01-22

Accept (Poster)